# Fatty acid oxidation fuels glioblastoma radioresistance with CD47-mediated immune evasion

Nian Jiang[1,2], Bowen Xie[1,3], Wenwu Xiao[4], Ming Fan[1], Shanxiu Xu[5], Yixin Duan[1], Yamah Hamsafar[6], Angela C. Evans[1], Jie Huang[1], Weibing Zhou[1,7], Xuelei Lin[2], Ningrong Ye[2], Siyi Wanggou[2], Wen Chen[1,7], Di Jing[4,7], Ruben C. Fragoso[1,8], Brittany N. Dugger[6], Paul F. Wilson[1,8], Matthew A. Coleman[1,8], Shuli Xia[9], Xuejun Li[2,10], Lun-Quan Sun [11], Arta M. Monjazeb[1,8], Aijun Wang [5], William J. Murphy [8,12], Hsing-Jien Kung[4,13], Kit S. Lam [4,8], Hong-Wu Chen [4,8,14] & Jian Jian Li [1,8✉]

Glioblastoma multiforme (GBM) remains the top challenge to radiotherapy with only 25% one-year survival after diagnosis. Here, we reveal that co-enhancement of mitochondrial fatty acid oxidation (FAO) enzymes (CPT1A, CPT2 and ACAD9) and immune checkpoint CD47 is dominant in recurrent GBM patients with poor prognosis. A glycolysis-to-FAO metabolic rewiring is associated with CD47 anti-phagocytosis in radioresistant GBM cells and regrown GBM after radiation in syngeneic mice. Inhibition of FAO by CPT1 inhibitor etomoxir or CRISPR-generated *CPT1A$^{-/-}$*, *CPT2$^{-/-}$*, *ACAD9$^{-/-}$* cells demonstrate that FAO-derived acetyl-CoA upregulates CD47 transcription via NF-κB/RelA acetylation. Blocking FAO impairs tumor growth and reduces CD47 anti-phagocytosis. Etomoxir combined with anti-CD47 antibody synergizes radiation control of regrown tumors with boosted macrophage phagocytosis. These results demonstrate that enhanced fat acid metabolism promotes aggressive growth of GBM with CD47-mediated immune evasion. The FAO-CD47 axis may be targeted to improve GBM control by eliminating the radioresistant phagocytosis-proofing tumor cells in GBM radioimmunotherapy.

[1] Department of Radiation Oncology, University of California Davis School of Medicine, Sacramento, CA 95817, USA. [2] Department of Neurosurgery, National Clinical Research Center for Geriatric Disorders, Xiangya Hospital, Central South University, Changsha, Hunan 410008, PR China. [3] Institute for Immunology and School of Medicine, Tsinghua University, Beijing 100084, PR China. [4] Department of Biochemistry and Molecular Medicine, University of California Davis, Sacramento, CA 95817, USA. [5] Department of Surgery, School of Medicine, University of California Davis, Sacramento, CA 95817, USA. [6] Department of Pathology and Laboratory Medicine, University of California Davis, Sacramento, CA 95817, USA. [7] Department of Radiation Oncology, Xiangya Hospital, Central South University, Changsha, Hunan 410008, PR China. [8] NCI-Designated Comprehensive Cancer Center, University of California Davis, Sacramento, CA 95817, USA. [9] Department of Neurology, Johns Hopkins School of Medicine, Baltimore, MD 21205, USA. [10] Hunan International Scientific and Technological Cooperation Base of Brain Tumor Research, Xiangya Hospital, Central South University, Changsha, Hunan 410008, PR China. [11] Center for Molecular Medicine, Xiangya Hospital, Central South University, Changsha, Hunan 410008, PR China. [12] Departments of Dermatology and Internal Medicine, UC Davis School of Medicine, Sacramento, CA 95817, USA. [13] TMU Research Center of Cancer Translational Medicine, Taipei Medical University, Taipei 110, Taiwan. [14] Veterans Affairs Northern California Health Care System, Mather CA95655, USA. ✉email: jijli@ucdavis.edu

Glioblastoma multiforme (GBM) is the most aggressive brain cancer in adults with the worst prognosis. Radiotherapy (RT) and/or chemotherapy with surgery are the major therapeutic modalities in control of GBM growth. However, the overall survival of GBM patients remains very low as 14.6 months with only about 5.6% 5-year survival rate[1–3]. Encouragingly, clinical benefits of combined radiation/immunotherapy (radioimmunotherapy) are evidenced in an array of solid tumors including GBM[4–6]. Promisingly, the tumor immunogenicity potentially enhanced by RT (the Abscopal Effect) and/ or radiation-induced molecules with latent immunomodulatory functions are increasingly identified in radiation-treated tumors[7,8], both of which are believed to contribute to the synergetic cancer control by radioimmunotherapy. However, in addition to such radiation-associated tumor response to immunotherapy, tumor microenvironment may also acquire resistance to immunotherapy under radiation[9], compromising the effectiveness of immune cell facilitated attacks on tumor cells[10,11]. Thus, for further validating and improving the efficacy of the combined modality, in addition to continuing on the elucidation of the radiation-induced immune attackable molecules, potential radiation-associate immune-cold status, especially in the recurrent and resistant tumors, is to be identified.

Metabolic reprogramming is one of the fundamental hallmarks in carcinogenesis and tumor progression featured by increased glycolysis in solid tumors including GBM[12]. However, flexible metabolic dynamics is demonstrated in mammalian cells under genotoxic stresses[13,14]. The oxidative respiration in mitochondria can be instantaneously adjusted in mammalian cells to meet the energy consumption for fueling cell cycle progression and DNA repair[15,16]. Recently we found that burning the saturated fat (i.e., palmitates in diet) by mitochondrial fatty acid oxidation (FAO) improves mitochondrial homeostasis[17]. Accumulating new evidence also indicates that in addition to the rudimentary glycolytic pathway, tumor cells are capable of reactivating oxidative phosphorylation (OXPHOS) to meet the increasing cellular fuel demand for repairing and surviving the genotoxic anticancer treatments[18,19]. Such bioenergetic flexibility appears to fit well into the increased capacity of tumors for metastasis[20–22], in which the fatty acid (FA) is an alternative critical energy resource to meet the high-fuel consumption in aggressively growing cancer cells[23–25]. FAO is shown to play a critical step for mitochondrial lipid digestion that enhances GBM metabolism[26,27], and targeting purine metabolism or NADPH biosynthesis increases the efficacy of GBM control[28,29]. Recently, two groups further reveal that FA metabolism accelerates the incidence of breast cancer brain metastasis[30] and that glycolysis is a less-essential uptake for GBM metabolism[31]. However, it is unknown whether the enhanced FA metabolism in tumor cells including the recurrent and radioresistant GBM can generate or boost an immunosuppressive status leading to the aggressive behavior with immune evasion.

CD47 is a well-defined immune checkpoint receptor protecting cells from the phagocytotic elimination by immune cells including macrophages via engagement of SIRPα on cell surface[32,33]. Therapeutic benefits by targeting CD47 are demonstrated in a series of pre-clinical and clinical studies in an array of human cancers[34–37] including GBM treated with radiation and anti-CD47 antibody[38], and CD47 is overexpressed in radioresistant breast cancer cells[37]. Using radioresistant GBM cells, regrown syngeneic mouse GBM, and recurrent tumors of GBM patients, this study reveals an immune evasion function of radioresistant GBM cells through FAO metabolism boosted CD47 antiphagocytosis. CD47 transcription is activated via FAO-derived acetyl-CoA that acetylates RelA K310 to upregulate CD47 transcription. Blocking FAO not only inhibits aggressive growth of radioresistant GBM cells but diminishes the orthotopically regrown mouse tumors by RT combined with anti-CD47 antibody. Blocking the acetyl-CoA-CD47 pathway is a potential therapeutic approach to sensitize radioresistant tumors to CD47 immunotherapy.

## Results

**Co-enhancement of FAO and CD47 in recurrent GBM with poor prognosis.** Glycolysis is documented to be the major metabolic pathway in cancer cells, increasing evidence suggests that metabolic rewiring with enhanced mitochondrial respiration including FAO is involved in tumor adaptive resistance and metastasis[39–43]. In this study, we aim to detect that FAO driving cell growth is conjugated with immunosuppressive function via FAO metabolite regulated CD47 expression. We first studied the history of 46 high-grade gliomas (HGG) with primary tumor paired recurrent disease following treatment of radio- and/or chemotherapy after initial surgery (Supplementary Table 1). IHC analysis revealed that the three essential FAO enzymes CPT1A, CPT2, and ACAD9 were increased 86.96, 84.78, and 76.09% respectively in the recurrent gliomas, companied with 82.6% CD47 enhancement (Fig. 1a). The expression of CD47 was positively correlated with CPT1A, CPT2, and ACAD9, respectively (Fig. 1b–d). In agreement, GBM database analysis (http://www.cgga.org.cn/index.jsp) of 504 HGG containing 220 recurrent and 284 primary GBM (not matched) also revealed the elevated expression of CPT1A, CPT2, ACAD9, and CD47 in the recurrent tumors (Fig. 1e). Furthermore, hierarchical clustering of CGGA RNA-seq heat map data revealed that the majority of FAO-related genes were positively correlated with *CD47* expression (Fig. 1f). Enrichment analysis further revealed that 157 FAO-related genes were enriched in tumors expressing a high level of CD47 (NES = 1.338, $P = 0.002$, Fig. 1g). Additionally, Pearson correlation analysis confirmed the positive correlations of CPT1A, CPT2, ACAD9, and CD47, respectively (Fig. 1h–j). Elevated expression of CD47, CPT1A, and CPT2 (except ACAD9) were negatively contributed to the overall survival (OS) (Fig. 1k–n). In agreement with the results above, the association of CD47 with CPT1A or CPT2 were shown as a significant clinical and pathological feature in the HGG patients (Supplementary Table 2). Intriguingly, although both CD47 and PD-L1 play key roles in immune checkpoint regulation, PD-L1, CPT1A isoforms (CPT1B and CPT1C) and ACAD9 isoforms (ACADVL and ACADS) did not show a detectable difference in expression between the primary and recurrent GBM tumors (Supplementary Fig. 1a), whereas CPT1B, ACADL, and ACADM (except CPT1C and ACADVL) were associated with lowered OS (Supplementary Fig. 1b–f). In consistence, Cox regression analysis of HGG patient data further identified a poor survival linked with elevated levels of ACADM, CPT2, CPT1A, CD47, and other factors including 1p19q codeletion, IDH mutation, and age (Supplementary Table 3). In addition, multivariate Cox proportional hazards model suggest that CPT2 (HR = 1.43, $P = 0.029$), ACADM (HR = 0.707, $P = 0.049$), 1p19q codeletion (HR = 0.38, $P < 0.0001$), and IDH mutation (HR = 0.43, $P < 0.0001$) were applied as independently predicters when considering age, expression of CD47 and CPT1A ($P < 0.05$) by univariate Cox regression. Based on these clinical data analysis and the well-established rate-limiting FAO cascade of CPT1, CPT2, and ACAD9 in mitochondrial FA transportation and digestion (Supplementary Fig. 1g)[44–46], we hypothesized that CPT1A, CPT2, and ACAD9 are the key FAO enzymes for rewiring metabolism by enhancing FA digestion that not only can meet the enhanced cellular fuel demand for tumor cell proliferation but also upregulate CD47 expression to protect tumor cells from macrophage-mediated immune surveillance, leading to the

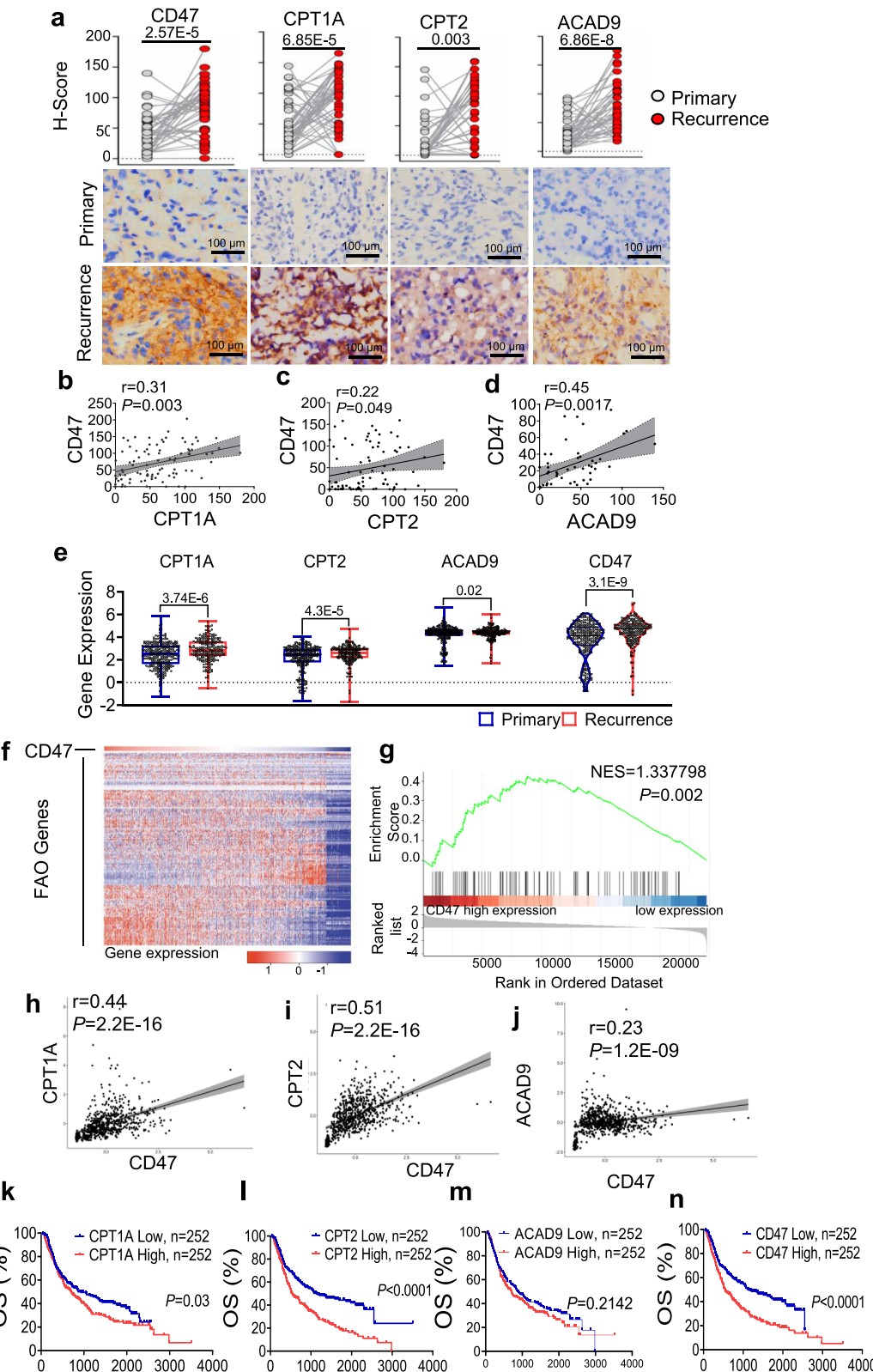

aggressive growth with immunosuppression in the radioresistant GBM.

**Mitochondrial FAO activity is dominant in radioresistant GBM cells.** Based on the above assumption, we tested that the basal glycolysis-dominant metabolism in the untreated tumor could be rewired to the FAO-centered metabolism in the radioresistant GBM cells. An array of radioresistant (RR) GBM cells were thus generated from human (U251, U87, and A172) and mouse (GL261) GBM cells following the established in vitro clinic-mimic radiation schedule (Supplementary Fig. 2a). The RR GBM cells were enriched with glioblastoma stem cells (GSCs) confirmed by specific marks of CD133, OCT4, SOX2, NANOG, and HER2 by Western blot (Supplementary Fig. 2b) with elevated

**Fig. 1 Co-enhancement of FAO enzymes and CD47 links to the recurrent GBM patients and poor prognosis. a** Scored expression of mitochondrial FAO enzymes (CPT1A, CPT2, ACAD9) and CD47 in a group of 46 recurrent high-grade gliomas (HGG) paired with their primary tumor biopsies (n = 46; paired two-tailed t-test). Representative IHC panels shown with scored FAO enzymes and CD47; scale bar = 100 μm). Pearson correlation coefficient analysis of CD47 expression levels with CPT1A (**b**), CPT2 (**c**), and ACAD9 (**d**) in the paired HGG tumors shown in **a**. Pearson correlation analysis was applied. Pearson correlation coefficient (r) and P values from Pearson correlation tests were shown. **e** Expression of CPT1A, CPT2, ACAD9, and CD47 in primary versus recurrent tumors from the CGGA database (n = 504; 504 HGG containing 220 recurrent and 284 primary GBM; the recurrent tumors were not individually matched with primary tumors). The box represents the 25th and 75th percentile, lines show medians, and error bars depict 1.5X IQR. An unpaired two-tailed t-test was applied. **f** Hierarchical clustering of RNA-seq of CD47 with FAO genes in a group of 504 GBM patients from the CGGA database. **g** Gene set enrichment analysis for a cluster of 157 FAO-related genes in CD47 high or low expressing tumors with green curve indicating a coordinative enrichment score and normalized enrichment score (NES) generated with GSEA software (NES = 1.337798 and P = 0.002). Pearson correlation coefficient analysis of CD47 expression levels with CPT1A (**h**), CPT2 (**i**), and ACAD9 (**j**) in HGG tumors from the CGGA database. Pearson correlation analysis was applied. Pearson correlation coefficient (r) and P values from Pearson correlation tests were shown. Kaplan–Meier overall survival (OS) of patients CGGA database categorized by high (red, n = 252) or low (blue, n = 252) expression of CPT1A (**k**, P = 0.03), CPT2 (**l**, P = 8.7E-7), ACAD9 (**m**, P = 0.2142), and CD47 (**n**, P = 6.09E-6). Kaplan–Meier survival analysis was applied. Results represent means ± SD. Source data are provided as a Source Data file.

CD133/HER2 population identified by flow cytometry analysis (Supplementary Fig. 2c, d). The enhanced aggressive behavior was measured by clonogenicity, neurosphere formation, migration, and invasiveness (Supplementary Fig. 2e–h). The FA-enhanced feature was recapitulated by PA-accelerated syngeneic RR GL261 tumor growth compared to that with control solvent treatment (Fig. 2a). Compared to wildtype (WT; untreated parental cancer cells) cells, mouse RR GL261 cells demonstrated an increased ATP generation peaked in the range of 12.5–25 μM for 48 h in FA-enriched medium with palmitate (PA) (Fig. 2b). By detecting isotope tracers ([9,10-³H(N)]-palmitic acid [0.5 μCi (~9.3 pmol)]) and ¹⁴C- palmitoyl-carnitine[47] tracer CPT2 enzymatic activity, RR GBM cell lines rewired the metabolism by increasing FAO rate (Fig. 2c). The boosted mitochondrial FAO activity was further identified by the elevated enzymatic activity of CPT2 measured by the rate of incorporation of labeled ¹⁴C-palmitoyl-carnitine into a pool of palmitoyl (Supplementary Fig. 3a). Furthermore, FAO production acetyl-CoA was also found to be elevated in RR GBM cells (Fig. 2d), whereas the glycolytic activity was lowered with decreased glucose uptake (Fig. 2e) and lactate generation (Fig. 2f) compared to all of their WT untreated counterparts. In addition, along with the increased mitochondrial numbers (Fig. 2g), RR GBM cells demonstrated an enhanced mitochondrial spare OCR which represents the potential adaptive metabolic capacity in comparison to the WT counterparts (Fig. 2h). The accelerated mitochondrial bioenergetics in the RR cells was also demonstrated by maximizing respiration with the uncoupler FCCP (Fig. 2h and Supplementary Fig. 3b–e). Etomoxir (ET) that inhibits CPT1 that couples with CPT2 for mitochondrial transportation of long-chain FA[48,49], was applied to inhibit FAO in RR GBM cells. Aggreged with lowered glycolysis, a remarkably low level of cytoplasmic lipid was detected in RR GBM cells which could be increased by CPT1 inhibition, whereas lipid accumulation in the WT cells was not affected by CPT1 inhibition (Fig. 2i and Supplementary Fig. 3f). Consistently, a substantial reduction of OCR and ATP generation was generated by ET in the RR but not WT cells (Fig. 2j, k and Supplementary Fig. 3g–j). Together, these metabolic features in RR GBM cells contrasted with WT untreated tumor cells indicate a capacity of rewiring the basal glycolytic pathway to an FA-dominant metabolism in RR GBM cells surviving radiation and sustaining an aggressive growth pattern.

**FAO-associated CD47 expression causes anti-phagocytosis in regrown GBM.** With the observed co-enhancement of FAO and CD47 in GBM prognosis and FAO-associated CD47 expression and anti-phagocytosis in RR GBM cells, we then assumed that in addition to fueling the energy demand for tumor cell survival and proliferation, the boosted FAO metabolism may offer the RR GBM cells with an immune evasion ability by protecting macrophage-mediated phagocytosis via FAO-mediated CD47 expression. Indeed, CD47 protein level was enhanced by radiation (IR 5 Gy) in WT tumor cells and remarkedly elevated in RR GBM cells (Fig. 3a). In addition, the percentage of CD47-expressing cells was increased 1.48- and 2.48-fold in radiation-treated WT and RR U251 cells and 3.2- and 5.97-fold in radiation-treated WT and RR U87 cells, respectively (Fig. 3b and Supplementary Fig. 4a). Furthermore, increased FAO enzymes and CD47 were detected in three human RR GBM cell lines compared to the corresponding parental WT GBM cells (Fig. 3c) and immunofluorescence revealed that co-expression of CPT1A, CPT2, or ACAD9 with CD47 was observed in RR cells (Supplementary Fig. 4b–d). To access the immune-escaping ability, macrophage-mediated phagocytosis was tested on WT and RR GBM cells which demonstrated a lowered phagocytosis rate in all three RR GBM cells compared to the WT counterparts (Fig. 3d and Supplementary Fig. 4e).

Co-enhancement of FAO enzymes with CD47 was then accessed by in vivo U251 tumor model. When tumors reached about 200 mm³, local treatment with fractionated radiation was delivered (2 Gy/day for 5 days) which led to the regrown tumors collected at day 28 (Fig. 3e). The quantitation of IHC staining and western blots demonstrated a substantial number of proteins of CD47 and CPT1A, CPT2, ACAD9 in regrown tumors compared to the control tumors without radiation (Fig. 3e, f and Supplementary Fig. 4f). Then, using GL261 syngeneic orthotopic GBM model with double-labeled Luc/GFP, the regrown tumor can be detected by magnetic resonance imaging (MRI) (Fig. 3g) by observing a significant shrinkage with undetectable tumor images at Day 17 and regrown tumor image at Day 28 (Fig. 3g). The elevated expression of CD47, CPT1A, CPT2, and ACAD9 in the regrown tumors was verified by IHC staining (Fig. 3h and Supplementary Fig. 4g), and macrophage phagocytosis was analyzed and scored by calculating the cells with GFP interacted or engulfed by macrophages, indicating a significantly lowered macrophage phagocytosis in the regrown tumors (Fig. 3i).

**FAO upregulates CD47 transcription via acetyl-CoA mediated RelA acetylation.** The fact that citrates and acetyl-CoA in the cytoplasm are major FAO metabolites and CD47 can be regulated by NF-κB[50,51] promoted us to test that FAO-derived citrates/acetyl-CoA functions to bridge the FAO metabolism and the CD47-mediated anti-phagocytosis. In consistence with FAO-mediated NF-κB activation, NF-κB luciferase activity was enhanced by PA (25 μM, 48 h) or mitochondrial FAO activator L-carnitine (gradient concentrations for 48 h) (Supplementary

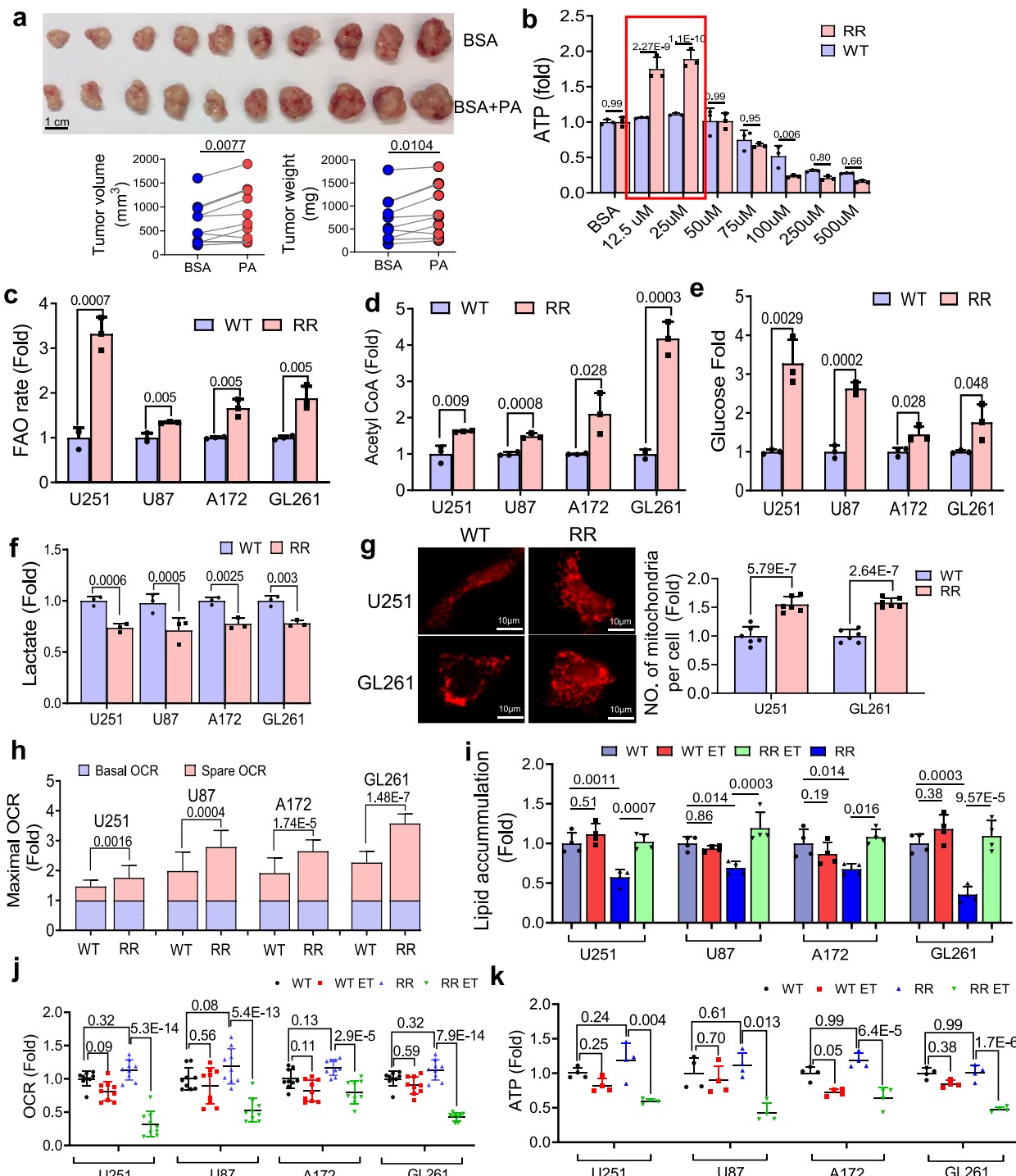

Fig. 5a, b) and CD47 mRNA level can be reduced by ET to 60 and 50% respectively in RR U251 and RR U87 cells (Fig. 4a). The NF-κB luciferase activity was reduced in the confirmed CRISPR KO $CPT1A^{-/-}$, $CPT2^{-/-}$, and $ACAD9^{-/-}$ cells (Fig. 4b and Supplementary Fig. 5c) that agreed with the reduced activity of CD47 promoter containing NF-κB motif (Fig. 4c) in the $CPT1A^{-/-}$, $CPT2^{-/-}$, or $ACAD9^{-/-}$ cells or the NF-κB motif was deleted (Fig. 4c) with CD47 mRNA level remarkably suppressed (Fig. 4d). NF-κB in coupling FAO and CD47 anti-phagocytosis was further supported by CGGA database analysis that identified a cluster of NF-κB subunits was tightly correlated with the mRNA levels of

CD47 in GBM patients, though the correlation was detected in both primary and recurrent GBM (Supplementary Fig. 6a, b), suggesting that FAO-mediated CD47 transcription is associated with FAO metabolites.

The citrates released from the FAO-enhanced TCA cycle is converted by ATP citrate lyase (ACLY) to acetyl-CoA that provides the acetyl group for acetylation in posttranslational modification. The citrate and acetyl levels were remarkably elevated in RR cells (Figs. 2d, 4e) and reduced by ET treatment or in $CPT1A^{-/-}$, $CPT2^{-/-}$, and $ACAD9^{-/-}$ cells (Fig. 4f and Supplementary Fig. 7a). CD47 promoter activity was enhanced by

**Fig. 2 Enhanced FAO drives the aggressive growth of radioresistant GBM cells. a** RR Gl261 cells ($5 \times 10^6$) were inoculated subcutaneously into both flanks of mice and when the tumor reached the sizes 200 mm³, palmitate (100 µl of 25 µM or 0.05% BSA as solvent control) were administrated twice 72 h apart via intratumoral injection. Tumor volumes ($P = 0.0077$) and weights ($P = 0.0104$) were measured at the end of experiment (Day 18; $n = 10$). A paired two-tailed $t$-test was applied. **b** ATP luminesce concentration of WT and RR GL261 cells treated by an indicated concentration of palmitate. $n = 3$. An unpaired two-tailed $t$-test was applied. **c** FAO activity quantified by the conversion of ³H palmitic acid to ³H₂O over 6 h using radioactive isotope tracers ([9,10-³H(N)]-palmitic acid [0.5 µCi (~9.3 pmol)]) ($n = 3$ per cell line) in parental WT and RR GBM cells (U251, U87, A172, and GL261). An unpaired two-tailed $t$-test was applied. The concentration of acetyl-CoA (**d**), glucose uptake (**e**), and L-lactate (**f**) was measured in WT and RR GBM cells (U251, U87, A172, and GL261; $n = 3$ per cell line). **g** Mitochondria number counts in parental WT and RR GBM cells (U251 and GL261; $n = 6$ per cell line; scale bar = 10 µm). **h** The basal and spared oxygen consumption rate (OCR) generated with FCCP, a potent uncoupler of mitochondrial oxidative phosphorylation in WT and RR U251 ($P = 0.0016$), U87 ($P = 0.0004$), A172 ($P = 1.74E-5$), and GL261 cells ($P = 1.48E-7$) ($n = 9$). An unpaired two-tailed $t$-test was applied in **b**–**h**. Lipid accumulation (**i**), OCR (**j**), and ATP generation (**k**) in WT and RR GBM cells treated with or without CPT1 inhibitor ET (in **i** and **k**, ET = 200 µM, 24 h, $n = 4$; in **j**, ET = 40 µM, 0.5 h, $n = 9$). Results represent the means ± SD; ANOVA two-way test was applied; WT wildtype, RR radioresistant, ET etomoxir. Source data are provided as a Source Data file.

citrate treatment whereas no obvious enhancement was observed with the promoter with deleted NF-κB binding motif (Fig. 4g). In addition, SB204990, a specific inhibitor of ACLY, reversed the citrate-enhanced NF-κB activation and CD47 mRNA level (Fig. 4h, i) and also reversed citrate-enhanced RelA acetylation and CD47 expression (Fig. 4j). However, in WT GBM cells, citrate cannot encourage *CD47* promoter activity (Supplementary Fig. 7b) although mildly upregulated *CD47* transcription which was also reversed by SB204990 (Supplementary Fig. 7c). Directly inhibition with A-485, an inhibitor for p300 regulated RelA acetylation[52] suppressed NF-κB luciferase reporter activity (Supplementary Fig. 7d) and CD47 promoter activity as well as gene expression which was recuperated in cells expressing RelA or mutant RelA K310R (Fig. 4k, l, and Supplementary Fig. 7e). Interestingly, KAT-specific acetylation sites in human CD47 proteins were predicted (Supplementary Fig. 8a) and also the acetylation of CD47 protein was enhanced by p300 and blocked by A-485 (Supplementary Fig. 8b). Thus, the FAO-mediated CD47 protein acetylation may help to stabilize protein which was indeed reversed by A-485 treatment (Supplementary Fig. 8c). Taking together, these results demonstrate that FAO-derived cytoplasmic acetyl-CoA is able to upregulate CD47 expression and protein stability via acetyl-CoA mediated acetylation on gene transcription and posttranslational modification.

**FAO inhibition suppresses radioresistant GBM cells aggressive growth with enhanced phagocytosis**. FAO-mediated CD47 anti-phagocytosis was further evidenced by reduction of macrophage phagocytosis in WT U251 and GL261 cells by boosting FAO activity with L-carnitine, PA, or radiation (Fig. 5a and Supplementary Fig. 9a). In addition, ET treatment on RR GBM cells can dose-dependently enhance macrophage-mediated phagocytosis (Fig. 5b, c and Supplementary Fig. 9b). However, a high dose of ET treatment on macrophages cannot increase phagocytosis (Supplementary Fig. 9c). Similar to ET-induced CD47 down-regulation, *CPT1A⁻/⁻*, *CPT2⁻/⁻*, and *ACAD9⁻/⁻* cells showed a strikingly reduced protein level of CD47 and CD47 positive cell populations (Fig. 5d and Supplementary Fig. 10a, b) and enhanced phagocytosis (Fig. 5e and Supplementary Fig. 10c, d) that can be enhanced by blocking ACLY or by inhibition of p300 with A-485 (Fig. 5f and Supplementary Fig. 10e). We assumed that such a pro-phagocytotic function by FAO suppression should also be accompanied by weakened aggressive growth of RR GBM cells due to reduced cellular fuel supply. Indeed, ET treatment or in *CPT1A⁻/⁻, CPT2⁻/⁻*, and *ACAD9⁻/⁻* cells, radiation-mediated elimination of the clonogenic tumor cells by 25–50% (Fig. 5g and Supplementary Fig. 11a–c). Similarly, apoptotic cell death was enhanced in CPT1A⁻/⁻, CPT2⁻/⁻, and *ACAD9⁻/⁻* cells or ET-treated cells (Fig. 5h and Supplementary Fig. 11d). Strikingly, the neurosphere formation and sphere sizes

which account for the critical aggressive behavior of GBM[53,54] enhanced in RR U251 cells were remarkably reduced by ET treatment or in *CPT1A⁻/⁻*, *CPT2⁻/⁻*, and *ACAD9⁻/⁻* cells (Fig. 5i, j and Supplementary Fig. 11e).

**Suppression of orthotopic regrown tumor by blocking of FAO and CD47**. In vitro analysis with RR U251 cells revealed substantial macrophage-mediated phagocytosis following treatment with ET (400 µM, 12 h) comparable to the level of anti-CD47 antibody (10 µg/ml, 2 h). However, dual treatment of ET and anti-CD47 antibody further enhanced the macrophagic attack (Fig. 6a and Supplementary Fig. 12). To evaluate the potential synergetic effects of inhibiting both FAO and CD47, the regrown model of mouse orthotopic GL261 tumors expressing with GFP/ Luciferase was generated allowing in vivo monitoring tumor growth after radiation (Fig. 6b). Animals were then grouped and fractionated radiation (3 Gy × 3) was delivered to the tumor region of all groups on Day 10 after inoculation when the tumor was visible. The shrinkages of tumor sizes were detected around Day 17 and observed regrowth by Day 24. Tumor volume in each mouse was monitored with the in vivo bioluminescence imaging (Fig. 6c and Supplementary Fig. 13). In the treatment groups, rat anti-mouse CD47 antibody generated from MIAP301 hybridoma cells (16 mg/kg), ET (30 mg/kg), or combination, was administrated via i.p. every other day for 14 days (total seven treatments) till Day 38 and tumor imaging and animal survival were monitored. We found that although tumor sizes were variable within each group, a noteworthy reduction of the regrown tumor volume was detected in mice treated by dual blockage with ET and CD47 antibody compared to ET or anti-CD47 alone although a significant inhibitory effect on the tumor regrowth was observed by each treatment (Fig. 6c, d). In consistence with the in vivo tumor imaging, the animal survival rate was remarkably enhanced in the group of dual blockades of FAO and CD47 compared to single treatment with one mouse actually surviving by Day 54 (Fig. 7a). These results encouraged us to detect if the macrophage-mediated phagocytotic attack contributes to a key role in such synergetic inhibition on the regrown tumors. Firstly, the potential capacity of anti-CD47 antibody crossing the blood–brain barrier (BBB) was identified by fluorescence analysis. Using the Fluorescent 14-bit images from Zeiss Axioscan.Z1 slide scanning system, the penetrated rat anti-CD47 antibody was detected by interacting to rabbit anti-rat IgG and rhodamine red conjugated goat anti-rabbit antibody on the surface of GFP-labeled tumor cells (Fig. 7b, c and Supplementary Fig. 14). Indeed, tumors treated by anti-CD47 antibody showed an increased antibody binding to tumor cells, whereas tumor binding antibody was less intensified in tumors treated with ET, indicating again that blocking FAO reduces CD47 expression. With similar reasoning, the tumor binding antibody in tumors with dual inhibition could not be

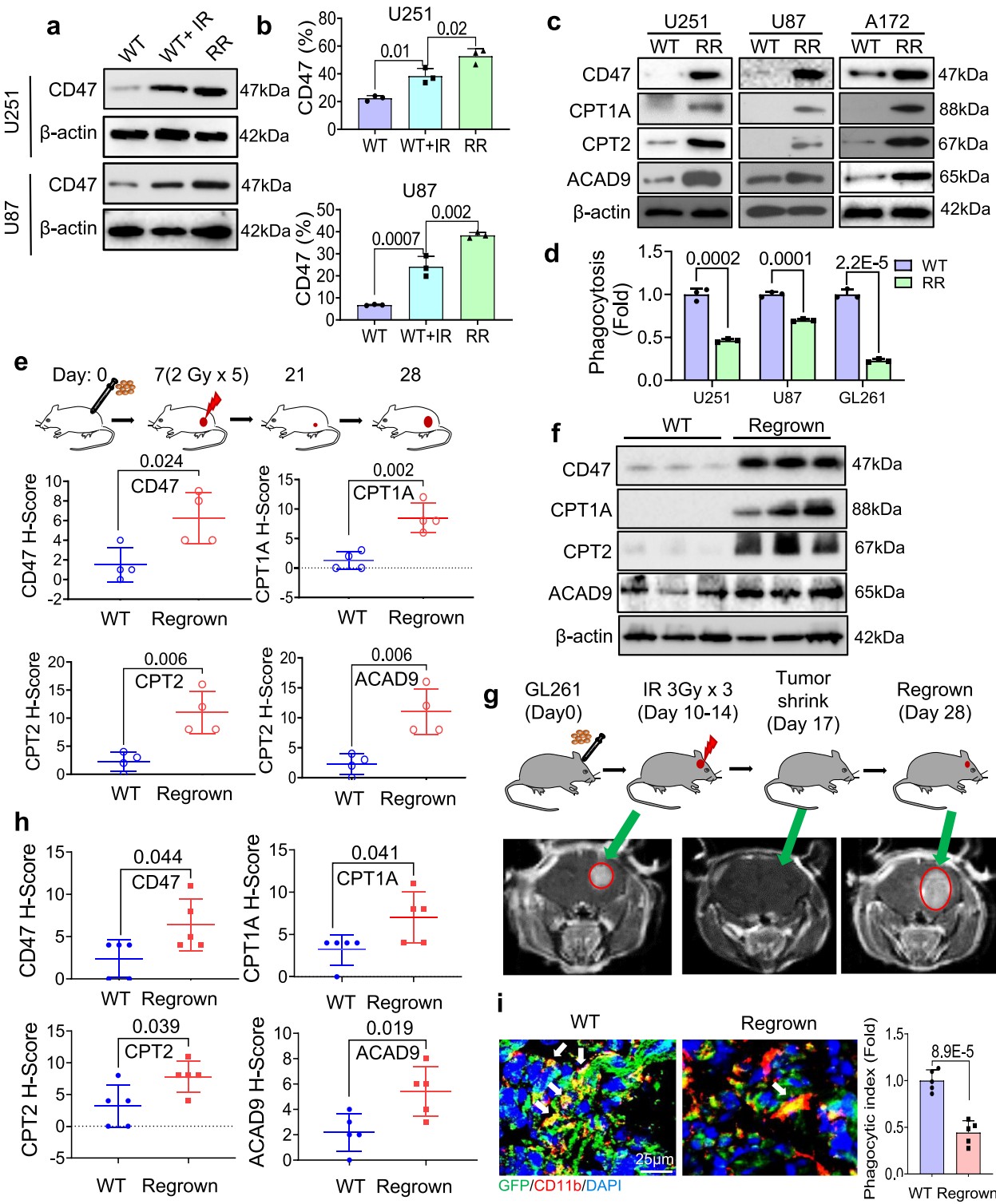

enhanced to the level of anti-CD47 treatment, suggesting an increased tumor cell death synergistically induced by blocking FAO and CD47. Consistent with inhibited tumor regrowth and antibody penetration in the combined treatments, increased macrophage-mediated phagocytosis on tumor cells was evidenced in combined treatment with ET and anti-CD47 antibody (Fig. 8a, b and Supplementary Fig. 15), although radiation with ET or anti-CD47 treatment also inspired a certain level of macrophage infiltration and phagocytosis on the tumor cells.

## Discussion

Fundamental mechanisms driving aggressive growth in the resistant and recurrent malignancy with life-threatening conditions remain as the major challenge in GBM treatment, although many milestones have been covered in improving the efficacy of treatment for other cancers. Using radioresistant cells and orthotopic regrown GBM, the present study uncovers an anti-phagocytic feature in recurrent GBM due to FAO-regulated CD47 expression. The glioblastoma stem cell (GSC)-enriched RR

**Fig. 3 Activation of FAO and CD47 with anti-phagocytosis in RR GBM cells and regrown tumors. a** Western blot of CD47 expression in WT, irradiated (5 Gy, 16 h) WT and RR GBM cells. ($n = 3$ experiments). **b** Increased CD47-expressing cell populations in **a** measured by flow cytometry ($n = 3$; ANOVA one-way test). **c** Western blot of CD47, CPT1A, CPT2, and ACAD9 in WT and RR GBM cells (U251, U87, and A172) ($n = 3$). **d** Macrophage phagocytosis of WT and RR GBM cells (U251, U87, and GL261) detected by flow cytometry ($n = 3$; unpaired two-tailed $t$-test). **e** Schematic for generating U251 xenograft tumors in nude mice with tumor radiation and IHC scoring for detecting the expression of CD47 ($P = 0.02$), CPT1A ($P = 0.002$), CPT2 ($P = 0.006$), and ACAD9 ($P = 0.04$) in untreated or regrown human GBM U251 tumors ($n = 4$; unpaired two-tailed $t$-test; *$P < 0.05$, **$P < 0.01$). **f** Enhanced expression of CD47 and CPT1A, CPT2, ACAD9 in three regrown U251 tumors after in vivo radiotherapy compared to three sham-irradiated control U251 tumors ($n = 3$ experiments). **g** Treatment protocol (upper panel) and MRI images (lower panel) of syngeneic mouse orthotopic GL261 tumors generated with inoculation of $2.5 \times 10^5$ GL261 cells double-labeled with Luc and GFP with in vivo radiotherapy of 3 Gy delivered daily for 3 days (total tumor radiation dose = 9 Gy). Representative images of regrown tumors at indicated time points were visualized by MRI. **h** IHC score of CD47 ($P = 0.044$) and CPT1A ($P = 0.041$), CPT2 ($P = 0.039$), and ACAD9 ($P = 0.019$) in regrown tumors (Day 28) compared to sham-irradiated controls ($n = 5$). An unpaired two-tailed $t$-test was applied. *$P < 0.05$. **i** Reduction of infiltrated macrophages and macrophage-mediated phagocytosis (indicated by arrows) in regrown tumors compared to WT tumors (green, tumor cells; red, infiltrated macrophages; quantitation of phagocytosis shown in the right; $n = 5$,). Significance was analyzed by a two-tailed $t$-test. Results represent means ± SD. Source data are provided as a Source Data file.

GBM cells are able to rewire the basal glycolysis-dominant metabolism to FAO-centered pathway which functions not only to fuel the cellular energy consumption demand for aggressive tumor growth but also, coordinatively, to defend the resistant cancer cells from macrophage-mediated phagocytosis. Such an alliance of FA-driven aggressive proliferation protected by CD47 anti-phagocytosis illustrates a unique survival advantage in the recurrent GBM cells. Thus, blocking the FAO-CD47 axis may eliminate the resistant cancer cells equipped with a boosted cellular energy fuel supply and CD47-mediated protection against macrophagic attack.

The glycolysis-dominant metabolism is a well-defined hallmark in malignant cells and glycolysis is shown to inactivate the immune cell activity causing an immune "cold" status[55,56]. However, it has been unclear if such a glycolysis-dominant pathway could be rewired or replaced by a high energy output metabolism in tumor cells that are capable of surviving genotoxic anticancer modalities including radiation. Indeed, a dynamic feature of tumor metabolism is recently identified in GBM radiosensitivity[29]. Such a programmatic metabolic ability is also detected in the present study using radioresistant GBM cells and a regrown mouse GBM model with enhanced lipid digestion for fueling the aggressive behavior of radioresistant GBM cells. The rewired metabolism is evidenced by reduced lactate generation with elevated FA metabolic capacity, accelerated cytoplasmic lipid turnover rates, increased mitochondrial OCR and ATP output. The FA-overriding is also illustrated by the results that inhibition of FAO remarkedly affected the radioresistant but not the parental untreated wild-type GBM cells, indicating that the rewired FA metabolism contributes to radiation-associated acquired tumor resistance. Shifting from glycolytic ascendency, the RR GBM cells show an increased spared respiratory capacity (SRC) that is related to the adaptive mitochondrial bioenergetics in cancer cell survival in radio- or chemotherapy[57,58]. In the light of FAO being able to promote SRC[59], we found that the elevated SRC is indeed involved in RR GBM cells following the glycolysis-FAO rewiring. Together, these findings support an adaptive FA-dominant metabolism in radioresistant GBM cells offering the survival advantage and aggressive behavior.

The FAO-mediated CD47 expression may contribute to the overall tumor adaptive (acquired) resistance with the feature of aggressive growth and anti-phagocytosis. A recent study demonstrates that CD47 knockout aggravates the lipid accumulation in normal liver cells[60] supporting tight coordination between FAO and CD47 expression. Among the potential key elements responsible for FAO-regulated CD47 expression, FAO-enhanced citrate level is shown to activate the CD47 promoter for gene transcription, thus functioning as the bridging factor between reprogrammed FA metabolism and CD47-mediated immune-escaping ability (Fig. 8c). The argument could be that the acetyl-CoA can also be generated by glycolysis in tumor cells. However, with the significantly lowered glycolysis in the RR GBM cells, it is likely that most cytoplasmic acetyl-CoA is resulted from FAO-derived citrates to form acetyl-CoA catalyzed by ACLY. In addition, acetyl-CoA is well defined to regulate fundamental cellular functions via protein lysine acetylation[61–63] including the prosurvival lipogenesis and DNA synthesis[64–66]. Indeed, NF-κB RelA K310 acetylation and CD47 promoter activity enhanced by acetyl-CoA were blocked by inhibition of ACYL with SB204990. Interestingly, predicted KAT-specific acetylation sites were also identified in the CD47 protein sequence and indeed, acetyl-CoA mediated CD47 acetylation demonstrated a reduced protein degradation (Supplementary Fig. 8). Therefore, the overall status of FAO-generated metabolic rewiring can activate multiple layers of signaling pathways for prosurvival and immune escaping in the GBM cells surviving radiotherapy. Further definition of the FAO metabolites that regulate other immune checkpoint and/or immune regulating factors will be a significant advance in revealing the dynamic immunosuppressive status in tumor microenvironment under radiotherapy. This study also suggests that tumors resided in a lipid-enriched microenvironment such as GBM and breast cancer brain metastasis, CD47-mediated immunosuppressive status could be readily adopted by the tumor cells via rewiring FA combustion with aggressive growth and immune evasion. However, in addition to citrate converted acetyl-CoA, other FAO-derived metabolic intermediates could also be involved in the checkpoint gene regulation including NADPH and reactive oxygen species (ROS), both have been identified in FAO-dominant cells and required for de-novo lipogenesis in promoting cancer growth[67,68].

The lipids released from the dead tumor cells in a tumor microenvironment treated by radiation and immunotherapy can also offer the survival advantage for cancer stem-like cells (CSCs) which are enriched in radioresistant tumor[57,59,69–71] via FAO-mediated self-renewal and metastasis[22,70]. Our present work further reveals that FAO-CD47-mediated anti-phagocytosis coordinatively contributes to the aggressive behavior of resistant tumor cells enriched with GSCs in the regrown tumors with the heterogeneity that is well-defined[72]. Thus, a metabolic reprogramming-associated immune-cold status in the GSC-enriched radioresistant/recurrent GBM holds a latent value in predicting tumor response to immunotherapy and in inventing new metabolic targets in the combined immunotherapy with radiation or chemotherapy. Furthermore, referred to as an Achilles' Heel of GSCs, the feature of DNA replication stress in radiation DNA damaging response[73,74] may also be tightly connected to the FAO dynamics since FAO serves the major cellular fuel for GSCs surviving radiation. Further deciphering the

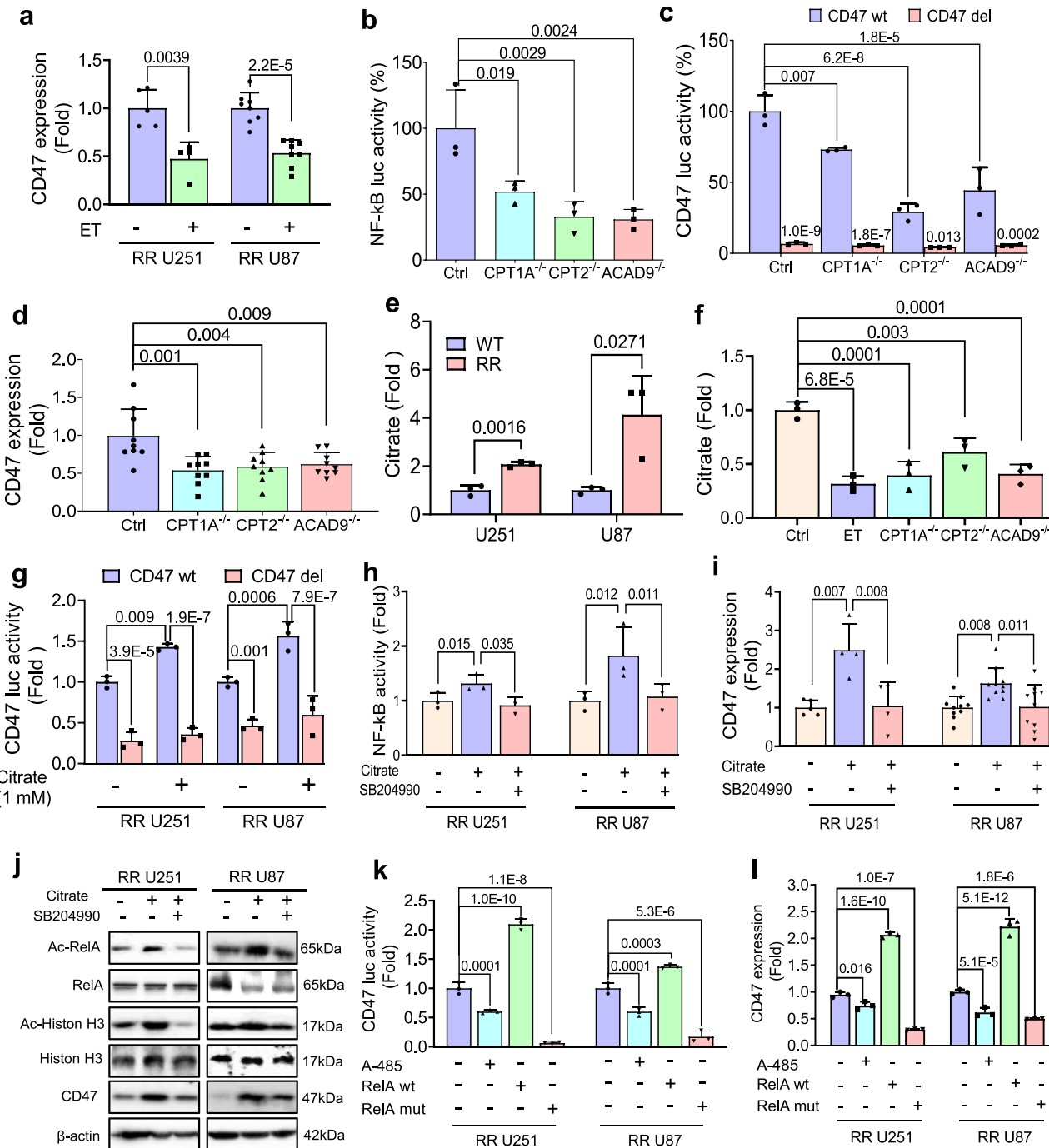

**Fig. 4 FAO regulates CD47 transcription via citrate/acetyl-CoA mediated RelA acetylation. a** CD47 mRNA levels in RR U251 ($n = 5$, $P = 0.0039$) and RR U87 ($n = 8$, $P = 2.2E-5$) cells treated with CPT1A inhibitor etomoxir (ET; 200 μM, 24 h). An unpaired two-tailed $t$-test was applied. NF-κB reporter **b** or CD47 promoter activity **c** in RR U251 cells compared to $CPT1A^{-/-}$, $CPT2^{-/-}$, and $ACAD9^{-/-}$ RR U251 cells (in **c**, CD47 promoter with NF-κB motif deleted as the negative control ($n = 3$). ANOVA one-way test was applied in **b** and ANOVA two-way test was applied in **c**. **d** CD47 mRNA levels in $CPT1A^{-/-}$, $CPT2^{-/-}$, and $ACAD9^{-/-}$ RR U251 cells. $n = 9$. ANOVA one-way test was applied. **e** Citrate concentrations measured in WT and RR GBM (U251 and U87) cells ($n = 3$). An unpaired two-tailed $t$-test was applied. **f** Citrate concentrations measured in $CPT1A^{-/-}$, $CPT2^{-/-}$, and $ACAD9^{-/-}$ or ET-treated RR U251 cells ($n = 3$). ANOVA one-way test was applied. **g** CD47 promoter-controlled luciferase activity with or without NF-κB motif deletion measured in RR U251 and RR U87 cells treated with citrate (1 mM, 6 h) ($n = 3$). ANOVA two-way test was applied. **h** NF-κB luciferase activity in RR U251 and RR U87 cells treated with citrate (1 mM, 6 h) in the presence or absence of ACLY inhibitor SB204990 (25 μM, 24 h) ($n = 3$). ANOVA one-way test was applied. **i** CD47 mRNA level in RR U251 ($n = 4$) and U87 ($n = 10$) cells treated with citrate (1 mM, 6 h) combined with or without ACLY inhibitor SB204990 (25 μM, 24 h). **j** Western blot of RelA K310 acetylation, Histone 3 acetylation in RR U251 and RR U87 cells treated with citrate (1 mM, 6 h) with or without SB204990 (25 μM, 24 h) ($n = 3$ experiments). **k** CD47 promoter-controlled luciferase activity either treated with A-485 (20 μM, 24 h) or with or without RelA mut measured in RR U251 and RR U87 cells. **l** CD47 mRNA level in RR U251 and U87 cells treated with A-485 (20 μM, 24 h) or transfected with WT or mutant RelA. In **k** and **l**, $n = 3$, ANOVA one-way test was applied. Results represent means ± SD. Source data are provided as a Source Data file.

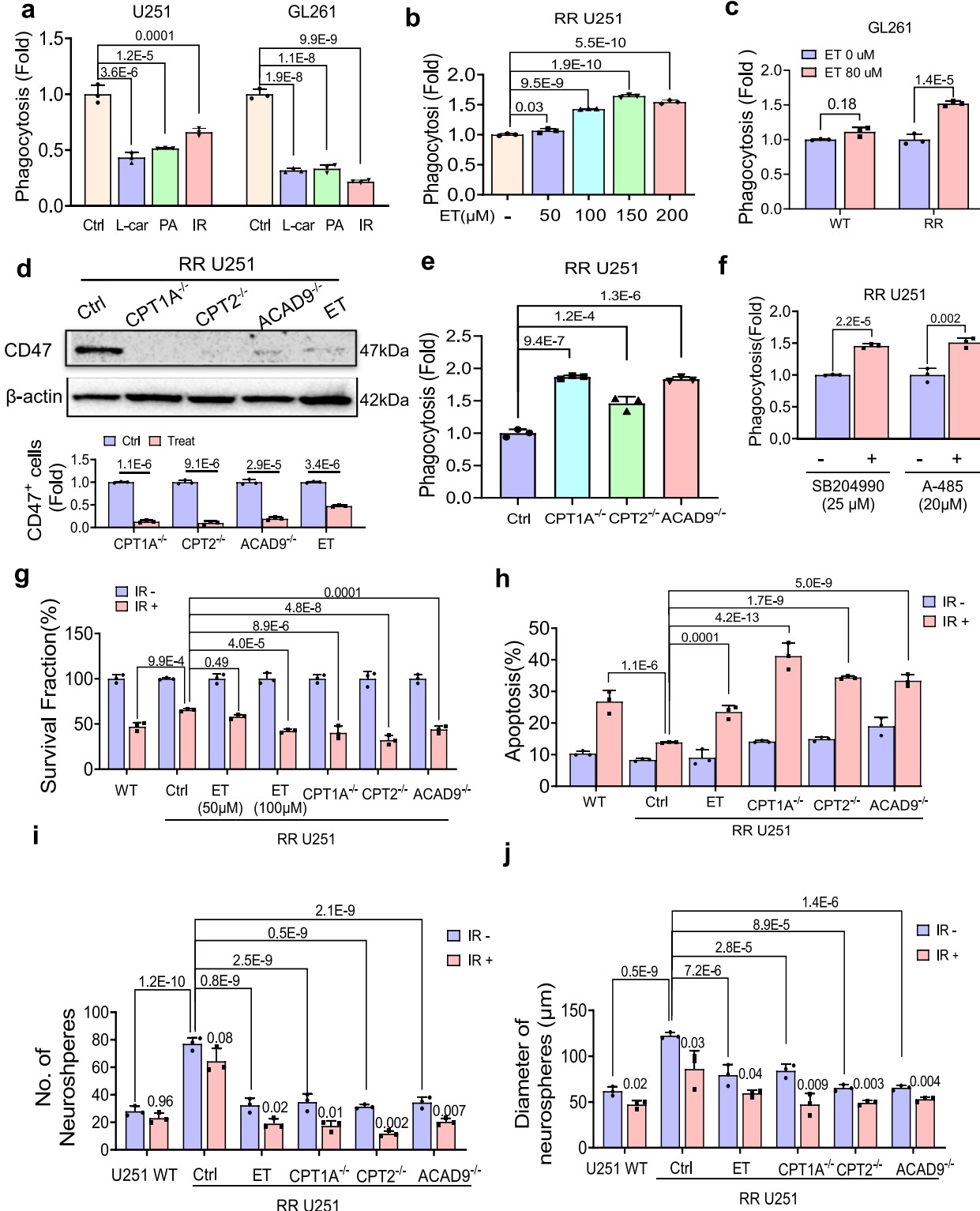

coordinative mechanism between a dynamic fatty acid metabolic rewiring and the anticancer immune enforcement in CSCs and GSCs will be of importance for discovering more effective CSC/GSC-targeting agents.

Ideally, an immune hot status favors the anti-tumor immunotherapy since immune cells can be activated by the cytotoxicity induced by anticancer therapies in the tumor microenvironment[75,76]. Like other critical immune checkpoints such as CTLA-4, PD-L1, and

PD-1, CD47 has been extensively studied with anticipated innate and adaptive immune system responses. An adaptive tumor immune resistance to CD47 immunotherapy is reported via suppressing radiation-associated tumor immunogenicity by hijacking caspase 9 signaling pathway[9]. The FAO-inducible CD47 promoter activity further supports a dynamic CD47 level in an irradiated tumor microenvironment. Therefore, such increased intensity of CD47 receptors on an RR GBM cell or a GSC will allow it to effactually

**Fig. 5 FAO deficiency boosts macrophage phagocytosis with diminished aggressive growth. a** Macrophage phagocytosis of WT U251 and GL261 cells treated with L-carnitine (L-car, 10 mM, 48 h), palmitate (PA, 25 μM, 48 h) or IR (3 Gy, 16 h). **b** Enhanced macrophage phagocytosis on RR U251 cells treated with indicated ET concentrations (24 h). In **a** and **b**, ANOVA one-way test was applied. **c** Macrophage phagocytosis of WT and RR GL261 cells treated with ET (80 μM, 24 h). **d** Upper panel: Western blot of CD47 in ET-treated (200 μM, 48 h) WT U251 cells compared to $CPT1A^{-/-}$, $CPT2^{-/-}$, and $ACAD9^{-/-}$ RR U251 cells. Lower panel: Percentage of CD47-expressing cells by flow cytometry in $CPT1A^{-/-}$, $CPT2^{-/-}$, and $ACAD9^{-/-}$ or ET-treated RR U251 cells (200 μM, 48 h). Analysis was conducted by two-tailed $t$-test in **c** and **d**. **e** Macrophage (THP1) phagocytosis of control and $CPT1A^{-/-}$, $CPT2^{-/-}$, $ACAD9^{-/-}$ RR U251 cells detected by flow cytometry. ANOVA one-way test was applied. **f** Macrophage (THP1) phagocytosis of RR U251 cells treated with ACLY inhibitor SB204990 (25 μM, 24 h) and A-485 (20 μM, 24 h) detected by flow cytometry. Clonogenic survival (**g**), apoptosis (**h**), neurosphere number (**i**), and neurosphere sizes (**j**) of WT and $CPT1A^{-/-}$, $CPT2^{-/-}$, and $ACAD9^{-/-}$ RR U251 cells or ET (200 μM, 48 h) treated RR U251 cells combined with or without IR (5 Gy) treatment. $n = 3$; Results represent means ± SD; Significance was analyzed with a two-tailed $t$-test in **g–j**. Source data are provided as a Source Data file.

escape the immunosurveillance of phagocytotic attack by immune cells. In addition, increased receptors on the RR GBM cells may efficiently consume anti-CD47 antibodies before it reaches the threshold concentration sufficient for tumor cell elimination, especially in the case of GBM with the challenge of antibody crossing the blood–brain barrier (BBB) or blood–tumor barrier (BTB). Our present study illustrates a variable tumor immune evasion ability due to metabolic rewiring status in the surviving cancer cells.

Checkpoint immunotherapy requires an efficient and consistent antibody concentration in the targeted tumor tissue which poses a challenge for GBM immunotherapy due to the limitation of BBB and/or BTB. It is reported that GBM breaks the integrity of BBB, making it possible to deliver a fraction of circulating drugs or IgG into the tumor parenchyma[77–81]. In addition to macrophages recruited from the periphery, microglia are found to be effector cells of brain tumor cell phagocytosis in response to anti-CD47 blockade[35]. Using the same mouse GL261 tumor, Gholamin et al.[82] demonstrate that the anti-CD47 antibody is able to penetrate the BBB, and blood concentration around 200 μg/ml can achieve 20 μg/ml in cerebrospinal fluid in the medulloblastoma bearing mouse. In consistent, the present study reveals the anti-CD47 antibody is indeed able to penetrate BBB and bind to tumor cells in irradiated orthotopic GL261 tumors treated with anti-CD47 antibody or combined with FAO inhibitor. These findings, in the line with radiation-enhanced BTB permeability of checkpoint immunotherapy[83], suggest a therapeutic advantage of radiation-mediated BTB permeability in antibody infiltration.

It is worthy of noticing that radiation can enhance GBM response to anti-CD47 antibody[38]. It is thus highly possible that in addition to damage-associated molecular patterns (DAMPs), the radiation-induced FAO-CD47 axis may at least in part lead to the synergistic efficacy enhancing the CD47-expressing cells for being targeted by the therapeutic antibodies. The effective tumor control by CD47 immunotherapy is being increasingly demonstrated. Untangling the FA-mediated tumor immune evasion function may help to elucidate the mechanistic insights between irradiated tumor cells and the infiltrated immune cells. However, the FAO-boosted CD47 overexpression although beneficial for enhancing tumor response to targeted antibody, may cause adverse effects including raising the threshold of antibody concentration for eliminating tumor cells and exhaustion of the infiltrated immune cells. We thus assume that a cluster of immunosuppressive genes including CD47 is involved in the adaptive immunotolerance responsive to genotoxic anti-tumor modalities, which is related to the immune evasion and metastasis due to adaptive response in treated tumors[84,85]. Radiation-induced metabolic rewiring may also be affected by radiation quality including the tumor-delivered dose and time[86]. Compared to high doses of radiation, lower doses of radiation are shown to induce high tumor immunogenicity and tumor response[87] or to increase the molecules of DAMPs required for enhancing

immune attack to tumor cells[88]. Therefore, elucidating the multiple communications between treated tumor cells and host immune surveillance will provide valuable information for inventing effective targets in the combined modality of radiation with immunotherapy.

In conclusion, this study reveals that glycolysis to FAO rewiring in radioresistant GBM cells can boost aggressive growth with CD47-mediated immune evasion via FAO-enhanced acetyl-CoA. Such FAO-driven intrinsic growth potential synchronized with the evasion to extrinsic immune cell attacks is identified in radioresistant GBM cells and recaptured in mouse regrown tumors and clinically treated tumors. This harmonic prosurvival coordination between lipid metabolic dynamics and the anti-phagocytotic function represents a highly flexible adaptive capacity of tumor cells under genotoxic conditions, also highlighting a vital role of metabolic rewiring in the immune-escaping ability of tumor cells. The FAO-CD47 axis is a potential target to eliminate the radioresistant and anti-phagocytotic tumor cells in GBM radioimmunotherapy.

## Methods

**Cell lines and other reagents**. Human monocytic THP1 cells and GBM U87 cells were purchased from American Type Culture Collection. U251 and GL261 cell lines were provided By Dr. Kit Lam's lab at the University of California Davis. A172 cells were kindly provided by Dr. Rajesh Khanna at the College of Medicine, University of Arizona. Three human cell lines U251, U87, and A172 were confirmed by Short Tandem Repeat (STR) analysis. The MIAP301 hybridomas were obtained from Dr. William Frazier (Washington University School of Medicine). Both human and mouse radioresistant GBM cell lines were generated by in vitro radiation using a Cabinet X-ray System Faxitron Series (dose rate: 0.028 Gy/min; Hewlett Packard, McMinnville, OR, USA) with fractionated doses (2 Gy × 20; total accumulated dose = 40 Gy). The surviving clones were selected based on western blot detection using CD133 antibody and maintained in the same medium formula as the parental cells. Experiments were conducted within ten passages after the termination of FIR treatment. U251 cells were cultured in MEM medium (CORNING Cellgro, Catalog #10-010-5 CV) supplemented with 10% FBS (CORNING, Catalog #35-010-CV), 0.1 mM NEAA (CORNING Cellgro, Catalog #25-025-CI), 1 mM sodium pyruvate (CORNING Cellgro, Catalog #25-000-CI), 10 mM Hepes (VWR, Catalog #97064-360) and 100 U/ml penicillin (Roche), and 100 μg/ml streptomycin (Roche). U87, A172, and GL261 cells were maintained in DMEM medium (CORNING Cellgro, Catalog #10-013-CV) containing 10% FBS, 100 U/ml penicillin, and 100 μg/ml streptomycin. THP1 cells were cultured in ATCC-formulated RPMI-1640 medium supplemented with 10% FBS, 15 mM Hepes, 4.5 g/L glucose, 2-mercaptoethanol to a final concentration of 0.05 mM, 100 U/ml penicillin, and 100 μg/ml streptomycin. Cells were cultured at 37 °C and 5% CO₂. The MIAP301 hybridomas were cultured in IMDM medium (GLBCO, Catalog #1240-053) supplemented with 20% FBS and 100 U/ml penicillin and 100 μg/ml streptomycin. Oil Red O (Catalog #O-0625), palmitic acid (Catalog #P5585)), A-485 (Catalog #SML2192), and etomoxir (Catalog #E1905) was obtained from Sigma-Aldrich (St. Louis, MO, USA). SB204990 was purchased from AdipoGen (Catalog #AG-CR1-3697). The anti-human CD47 antibody Hu5F9-G4 was kindly gifted from Dr. Irving Weissman.

**Human GBM studies**. Pathological slides of 46 GBM patients with primary biopsy matched with recurrent tumors with a history of radio- and/or chemotherapy after surgery were included in this study (Supplementary Tables 1–3). The patient's information on diagnosis and treatment was obtained with the approval of the Human Ethics Committee of Xiangya Hospital, Central South University, and

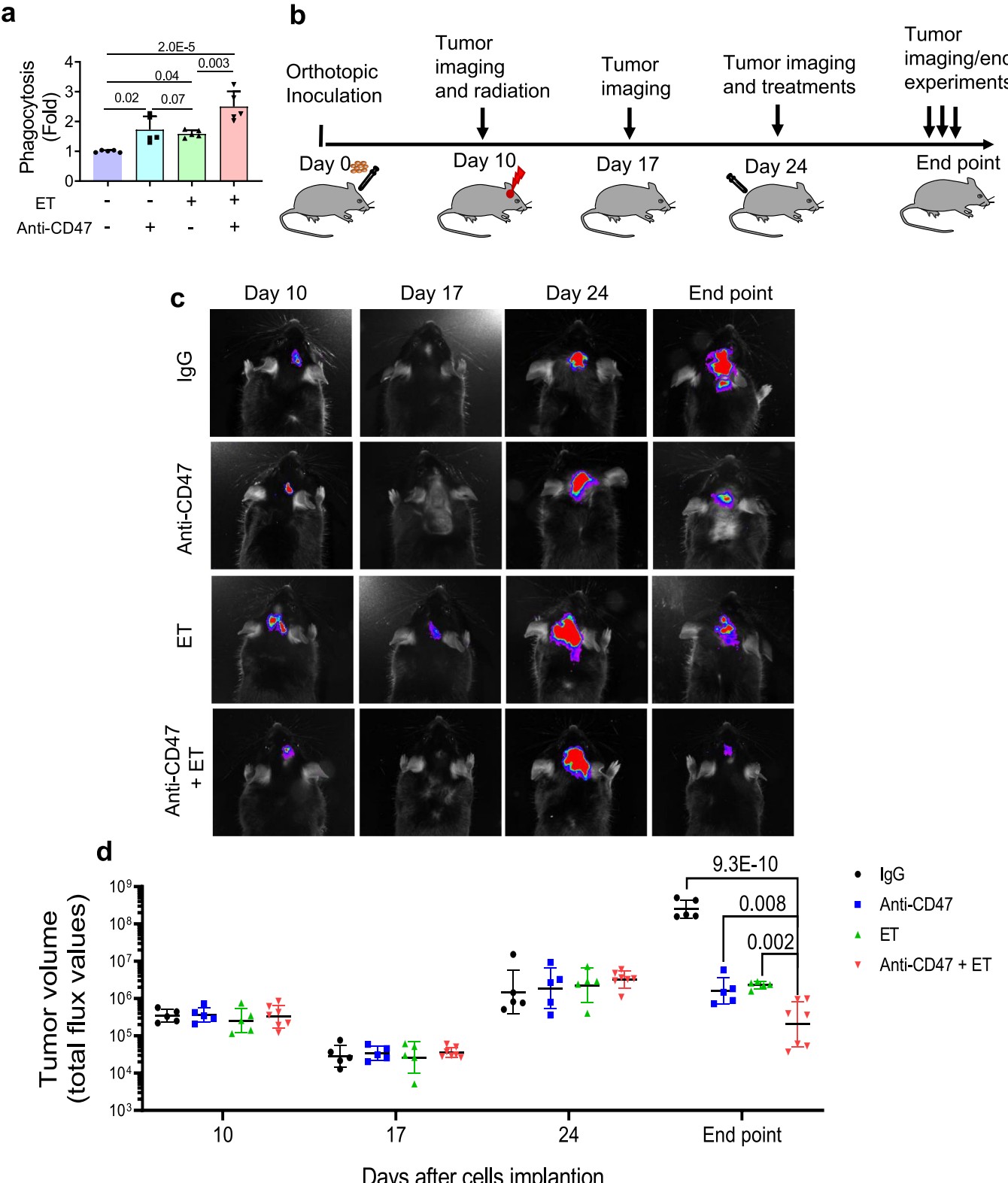

informed consent from all patients. All of the 46 patients underwent intracranial surgery during a period from March 2010 to January 2019, and 34 of them received radiotherapy after the first surgery. All pathological slides of primary matched with recurrent tumors were evaluated according to H&E staining.

**Animals**. Animal use and care protocol of in vivo radiation treatment were approved by the Institutional Animal Use and Care Committee of the University of California Davis (IACUC 15315). Animals were housed and cared for according to

standard guidelines with free access to food and water. All experiments were performed on 6 weeks old female C57BL/6 mice (Taconic) or 6 weeks old female nude mice (The Jackson Laboratory, Stock No:002019). Animals including littermates of the same sex were randomly assigned to control or treatment conditions.

**Immunohistochemistry (IHC)**. Tumor tissues were fixed with 10% formalin followed by an embedding in paraffin wax and sectioned at 5 μm thickness. After deparaffinization and rehydration, antigen retrieval was conducted with citrate

**Fig. 6 Radiation with blocking FAO and CD47 diminishes tumor regrowth. a** In vitro macrophage phagocytosis of RR U251 cells treated by ET (400 μM, 12 h), anti-CD47 antibody (10 μg/ml, 2 h), or ET followed by an anti-CD47 antibody; results were normalized with the cells treated with IgG ($n = 5$, ANOVA two-way test). **b** Schematic diagram of GL261 orthotopic tumors treated with fractionated irradiation (3 Gy × 3) followed by treatment with FAO inhibitor ET, rat anti-mouse CD47 antibody or combined of two by administration via i.p. Normal rat IgG (Cat. BE0094, BioXcell) used as non-reactive control (16 mg/kg; $n = 5$), and anti-CD47 antibody (16 mg/kg; $n = 5$), ET (30 mg/kg; $n = 5$), or combination of anti-CD47 antibody (16 mg/kg) and ET (30 mg/kg) ($n = 7$) were administrated every other day for seven treatments in 14 days. The tumor volume of each animal was monitored by luminescence flux values of IVIS (**c**, **d**) and tests were terminated based on the appearance of standard symptoms for termination. In **d**, $n = 7$ biologically independent animals in a combination of anti-CD47 antibody and ET group and $n = 5$ biologically independent animals in each of other groups. Significance was analyzed with ANOVA one-way test. Results represent means ± SD. Source data are provided as a Source Data file.

buffer (pH 6.1) at 95 °C for 40 min and endogenous peroxidase activity was blocked with 10% horse serum before incubation with primary and secondary antibody using Vectastain ABC Kit (Catalog #PK-6100, Vector Laboratories). The slides were then reacted with a DAB peroxidase substrate kit (Catalog #SK4100, Vector Laboratories) and counterstained with hematoxylin (Catalog #H-3401-500, Vector Laboratories). The target protein expression levels were estimated by grading the staining intensity and calculating the positive cells as the following: 0: negative staining; 1: weak; weak staining intensity in more than 25% of tumor cells; 2: moderate; moderate staining intensity in more than 25% of tumor cells; 3: strong; strong staining intensity in more than 25% of tumor cells. For multispectral imaging and analysis, the InForm 2.4 software (PerkinElmer) was used to batch analysis of multispectral images from the experiment. To build an algorithm for tissues and cells segmentation, a few representative multispectral images from the experiment were loaded into InForm software. Areas with non-tumor tissues were excluded before cells segmentation, the segmented tissues and cells were trimmed according to Hematoxylin or DAB signals. For scoring, three to six representative regions of interest from each case were chosen by the pathologist for high-powered (200×) imaging. After the detected tissues compartments were selected and quantified for each stained protein in slides, thresholds for "positive" staining and the accuracy of phenotypic algorithms were optimized and confirmed by the pathologist. 0–3 + (4-bin) scoring system was used to quantify proteins levels. The objective images were then loaded into InForm and the established algorithm was used for IHC scoring. The score system can be used to calculate H-score with cell stains. It included four levels (0/1+, 1+/2+, 2+/3+, 3+), score results were shown by the percentage positivity of cells with each bin. H-score was calculated using the percentages in each bin and ranges from 0 to 300. Batch process all remaining multispectral images from the experiment.

**CGGA cohort**. *CD47* expression was correlated with expression of FAO enzymes *CPT1A CPT2*, and *ACAD9* in primary versus recurrent tumors from the CGGA database (504 HGG containing 220 recurrent and 284 primary GBM; the recurrent tumors were not individually matched with primary tumors). Additionally, gene set enrichment analysis was conducted with a cluster of 157 FAO-related genes in CD47 high or low expressing tumors to generate a coordinative enrichment score and normalized enrichment score (NES) generated with GSEA software. Kaplan–Meier overall survival (OS) analysis was derived from the CGGA database categorized by high ($n = 252$) or low ($n = 252$) expression of FAO enzymes and CD47 expression.

**Bioinformatics analysis**. The information on Glioma samples was obtained from the Chinese Glioma Genome Atlas organization (http://www.cgga.org.cn). The RNA-seq data were normalized for analyzing relative expression, and the overall survival rate of GBM patients was evaluated using the Kaplan–Meier method. The statistical differences in survival length were determined using the log-rank test. Heatmaps, clustering analyses, and correlation analyses were performed using the ggplots, fgesa, and limma R language packages, respectively.

**ATP generation**. Cellular ATP generation was measured by luciferase ATP assay. Cells were seeded in 96-well plates and cultured for different time intervals according to the experiments. Cells were washed with cold phosphate-buffered saline (PBS) twice followed by extracting cellular ATP using 80 μl of cold 0.5% (w/v) trichloroacetic acid (TCA) and shaking on ice for 20 min. The extraction was then neutralized by adding140 μl of 250 mM Tris-acetate buffer (pH 7.75) and 10 μl of neutralized ATP extracts was mixed with 40 μl of fresh luciferin-luciferase reagent (20× reaction buffer, 0.1 M DTT, 10 mM Luciferin, 0.25 μl luciferase; Invitrogen), and the ATP concentrations were detected by Turner Biosystems 20/20 Luminometer (Promega) and the results were normalized to the protein concentration using the BCA method (Pierce, Rockford, IL).

**Radioactive FAO rate measurement**. The cellular FAO was determined following the published literature[89]. Briefly, $3 \times 10^5$ cells were seeded into each well of a 12-well plate. After 12 h incubation, [9,10-$^3$H(N)]-palmitic acid [0.5 μCi (~9.3 pmol)] (PerkinElmer, Catalog# NET043001MC) in 15 μl carrier solution (1 mM sodium palmitate, 0.17 mM BSA, 150 mM NaCl) was added to each well and incubated for

6 h. The [9,10-$^3$H(N)]-palmitic acid was converted to $^3$H$_2$O via mitochondria FAO. To control the background, each experiment contained cell-free control. After labeling, 0.5 ml out of 1 ml culture medium was collected into a 15 ml tube with 100 μl of 1.2 N HCl, and an uncapped 0.5 ml tube containing 0.25 ml of sterile distilled water was carefully inserted into the tube and further incubated for 3 days, and the radioactivity present in the water and medium were determined by scintillation counting. The FAO rate of each sample was calculated and normalized to protein concentration.

**Acetyl-CoA assay**. Commercially available fluorometry-based assay (MAK039, Sigma-Aldrich, Ontario, Canada) was used. Cells were collected from 100 mm dish plate after reaching 80% confluency and lysed with lysis buffer (20 mM Tris pH 7.5, 150 mM NaCl, 1 mM EDTA, 1 mM EGTA, 1%Triton X-100, 2.5 mM sodium pyrophosphate, 1 mM β-glycerophosphate, 1 mM Na$_3$VO$_4$, 1 μg/ml leupeptin, and 1 mM PMSF). Cell lysates were deproteinized by 1 M perchloric acid and neutralized by 3 M potassium bicarbonate solution to make the final pH in the range of 6–8. About 50 μl of the diluted sample (~50 μg total protein) was mixed with an equal amount of the reaction mix (41.8 μl Acetyl-CoA Assay buffer, 2 μl Acetyl-CoA substrate mix, 1 μl conversion enzyme, 5 μl Acetyl-CoA enzyme mix, 0.2 μl fluorescent probe). Each experiment contained a blank control to omit the conversion enzyme in the reaction mix. Samples were prepared in triplicate in a 96-well plate at 37 °C for 10 min in dark. Fluorescence intensity was measured by using the fluorescence Microplate spectrophotometer (Molecular Devices) at $\lambda_{ex} = 535/\lambda_{em} = 587$ nm. The results were then normalized to input protein content.

**Glucose detection**. The levels of glucose in the cell culture medium of four paired wildtype and RR GBM U251, U87, A172, and GL261 cell lines were measured by glucose assay kit (Catalog #A-114, Biomedical Research Service Center, Buffalo, NY, USA). According to the manufacturer's protocol, $2 \times 10^4$ cells were seeded per well in 96-well plates, incubated for 24 h at 37 °C with 5% CO$_2$. About 25 μl cell culture medium was deproteinized by precipitation with an equal amount of PEG Solution, vortex and incubated on ice for 30 min and spin down at 13, 15,800×$g$ for 5 min at 4 °C. Supernatants were harvested and diluted 20- to 40-fold with distilled H$_2$O. The reaction was initiated by adding 50 μl of Glucose Assay Solution and incubated at 37 °C for 30 min. Absorbance was measured using the fluorescence microplate spectrophotometer (Molecular Devices) at 492 nm. Glucose concentrations were calculated by derived equation from standards performed at the same time times dilution factor and normalized based on cell number.

**L-Lactate assay**. To determine the energy shift in RR GBM cells from the status of glycolysis to oxidative phosphorylation, L-Lactate concentration in four paired wildtype and RR GBM U251, U87, A172, and GL261 cell lines were tested using the L-Lactate Assay kit (Catalog #A-108L, Biomedical Research Service Center, Buffalo, NY, USA). According to the manufacturer's protocol, $3 \times 10^4$ cells were seeded per well in 96-well plates, incubated for 24 h at 37 °C with 5% CO$_2$. About 50 μl cell culture medium was mixed with an equal amount of PEG Solution, vortexed and incubated on ice for 30 min followed by centrifuging at 15,800×$g$ for 5 min at 4 °C, and diluted 20 μl supernatant by fivefold ice-cold dH$_2$O (80 μl) prior to assay. The reaction was initiated by adding 50 μl of lactate assay solution and stopped by the addition of 50 μl of 3% acetic acid. The L-Lactate level was measured using the fluorescence microplate spectrophotometer (Molecular Devices) at 492 nm. Plot lactate standards vs. samples respective OD (492 nm). Due to the ten times dilution of samples by PEG and dH$_2$O, the result needs to be multiplied by the dilution factor.

**Mitochondria mass**. pERFP-N1-MTS plasmids were constructed in frame insertion of an 87 base MTS (mitochondria targeting sequence) fragment[90]. Cells were cultured in pre-coated 15 mm round cover Glass with 0.01% poly-L-lysine. TurboFact transfection reagent (Thermo Scientific. Catalog #R0534) was used according to manufacture instructions. After 48 h transfection, cells were washed two times with 1X PBS and followed by fixation in 40% paraformaldehyde about 20 min and two times washes with 1X PBS. High-resolution fluorescence images were acquired using a 63× oil lens of Zeiss Observer.Z1 microscope. Fluorescence

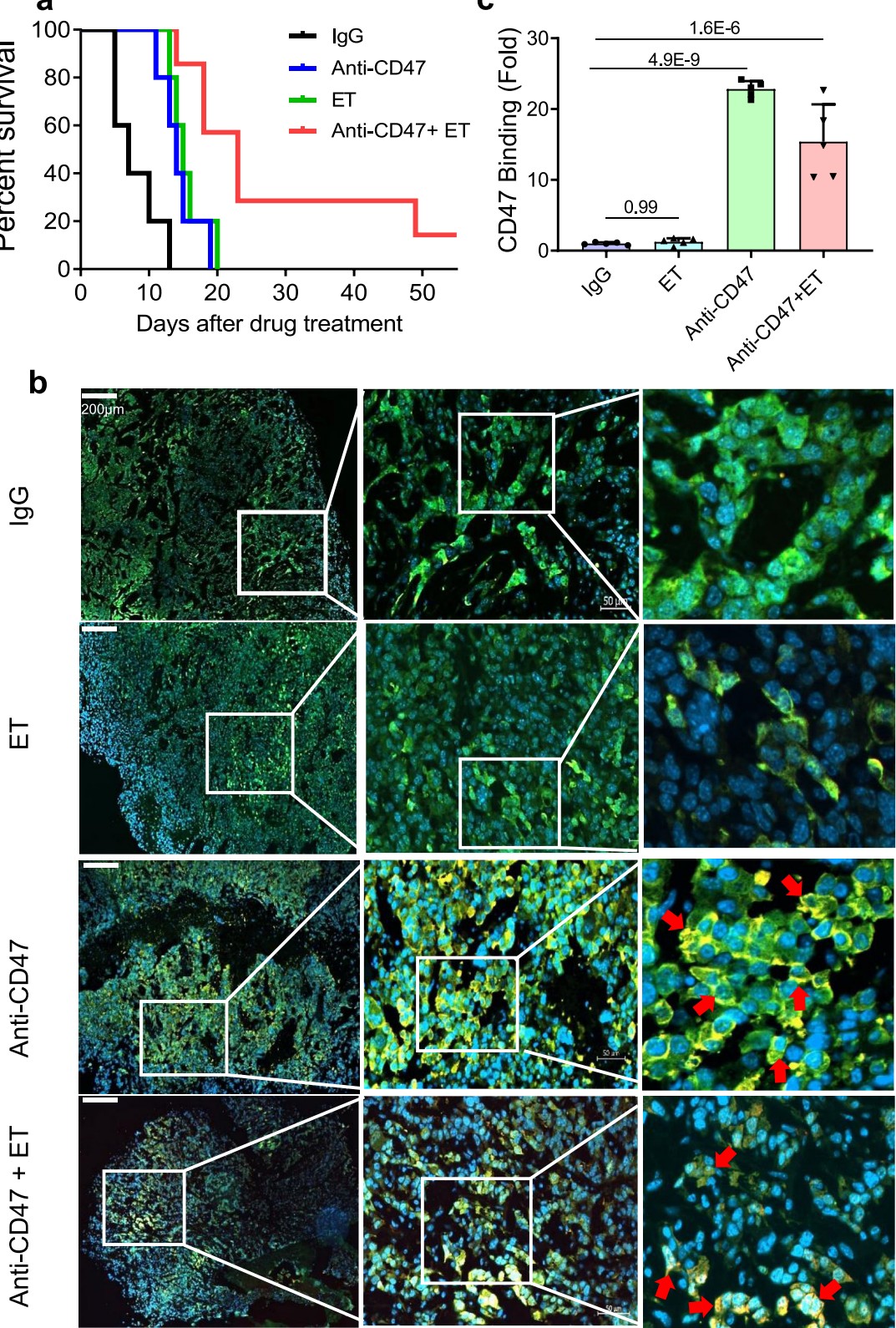

**Fig. 7 Animal survival and anti-CD47 antibody tumor penetration in combined treatment. a** Kaplan–Meier survival of mice of Fig. 6b. **b** Representative immunofluorescence images of infiltrated anti-CD47 antibody in GFP- labeled GL261 orthotopic tumors. Antibody tumor penetration and binding to the targeted receptor were identified by rabbit anti-rat IgG as primary antibody and Rhodamine Red-X-Conjugated goat anti-rabbit as the secondary antibody. The mouse anti-GFP antibody and Alexa 488 conjugated goat anti-mouse secondary antibody recognized GFP expressing GL261 tumor cells. The binding of the anti-CD47 antibody on tumor cells is indicated with red arrows (green, tumor cells; yellow, rabbit anti-rat CD47 antibody bind to CD47 presenting tumor cells; blue, nucleus stained with DAPI). **c** Antibody binding density of control and treated tumors was estimated by Image-Pro Plus 6.0 ($n = 5$). Results represent means ± SD; ANOVA one-way test was applied. Source data are provided as a Source Data file.

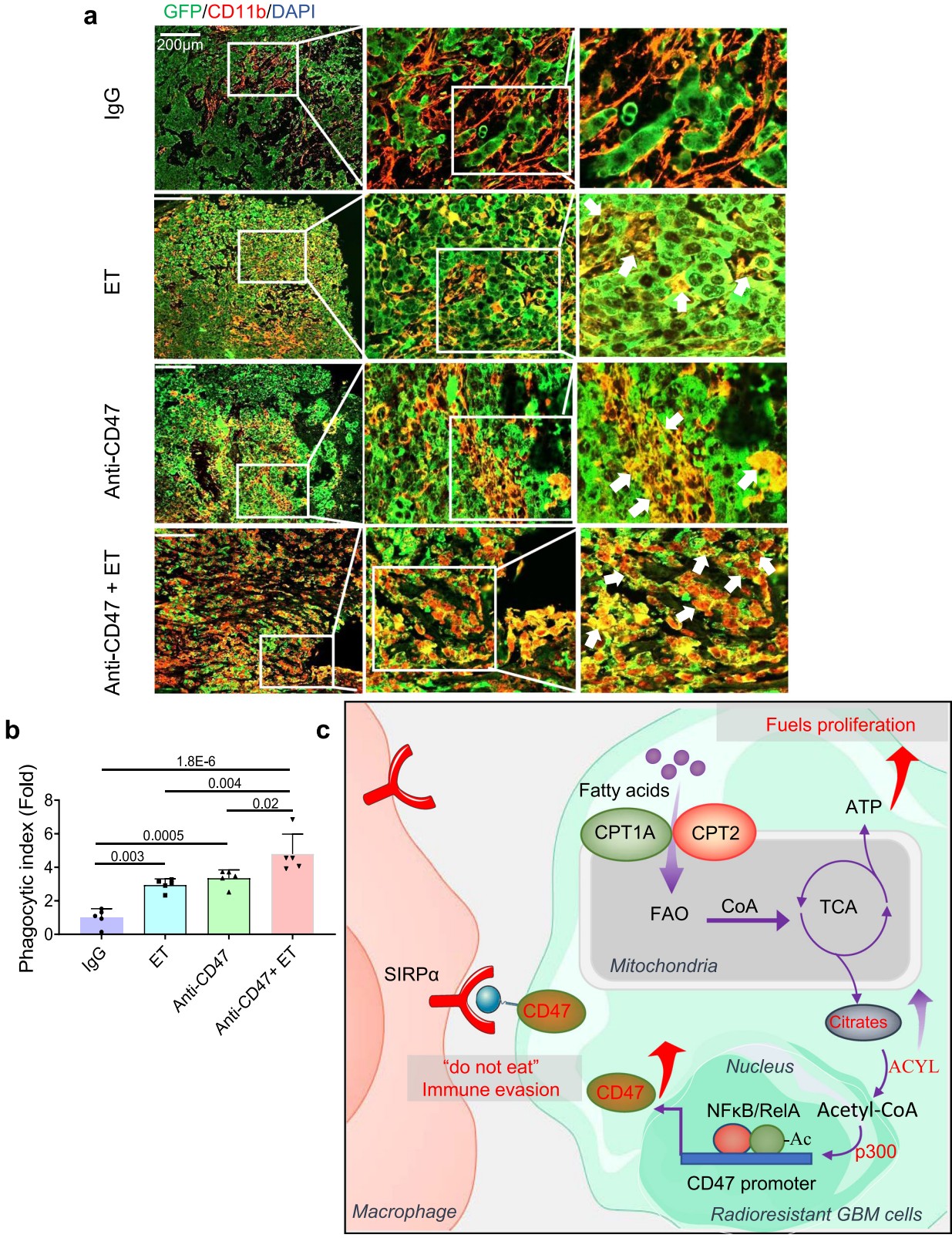

**a** GFP/CD11b/DAPI

200μm

IgG

ET

Anti-CD47

Anti-CD47 + ET

**b**

Phagocytic index (Fold)

1.8E-6
0.004
0.02
0.0005
0.003

IgG  ET  Anti-CD47  Anti-CD47+ ET

**c**

Fuels proliferation

Fatty acids

CPT1A  CPT2

FAO  CoA  TCA  ATP

Mitochondria

Citrates

SIRPα  CD47

"do not eat"
Immune evasion

CD47

ACYL

Nucleus

NFκB/RelA  Acetyl-CoA
-Ac
CD47 promoter  p300

Macrophage  Radioresistant GBM cells

intensity was quantified using Axiovision software (Zeiss, Germany). Mitochondria mass was quantified by Image J software (NIH Image).

**OCR**. The extracellular oxygen consumption was measured using the Extracellular Oxygen Consumption Assay kit (ab197242, Abcam). Cells were plated in 96-well plates ($3 \times 10^4$ cells/well) and cultured overnight at 37 °C, then replaced with 90 μl of fresh culture media and added 10 μl of extracellular $O_2$ consumption reagent into each well. The wells without $O_2$ consumption reagent were served as blank

controls. All the wells were then sealed with 100 μl of prewarmed high-sensitivity mineral oil. The real-time kinetic analysis of OCR was read at 1.5 min intervals for 180 min at 37 °C using the fluorescence microplate reader (Molecular Devices, Sunnyvale, CA, USA) with the ratio EX/EM = 380/650 nm. The OCR was analyzed by calculating the rate of signal for each assay profile.

**FAO assay**. FA oxidation, the key step in mitochondrial fatty acid metabolism, was detected using the Assay kit (ab217602) in combination with the Extracellular

**Fig. 8 Enhanced macrophage phagocytosis of tumor cells in combined treatment. a** Immunofluorescence images of macrophage phagocytosis in GL261 tumors treated with ET, anti-CD47 antibody, or anti-CD47 antibody + ET as in Fig. 6b. Macrophage phagocytosis of tumor cells are indicated with white arrows (green, tumor cells; red, infiltrated macrophages, blue, nucleus stained with DAPI). **b** Tumor macrophage phagocytosis was estimated by quantitation of phagocytic index by Image-Pro Plus 6.0 (n = 5). Results represent means ± SD; ANOVA two-way test. **c** A schematic pathway illustrating the FAO-CD47 axis-mediated aggressive phenotype and anti-phagocytosis in radioresistant GBM cells. The adaptive rewiring in metabolism increases the mitochondrial lipid combustion which is companied with the immunosuppressive function of CD47 anti-phagocytosis protecting the radioresistant GBM cells from macrophagic attack. FAO upregulates CD47 transcription via NF-κB RelA acetylation by elevating cytoplasmic citrate concentration. Thus, the coordinative mechanism between FA metabolic dynamics and immune-escaping capacity in tumor cells features the involvement of multiple pathways contributing to an immunosuppressive (cold) tumor. Inhibition of mitochondrial FAO is suggested to enhance the control of GBM by radiation with anti-CD47 immunotherapy. Source data are provided as a Source Data file.

Oxygen Consumption Assay kit. Cells were cultured overnight and were washed with 100 µl of prewarmed FA-free Medium three times followed by adding 90 µl of FA medium containing FAO conjugate (150 µM oleate-BSA). The wells without cells were included as the basal signal controls. The wells with FA medium only were used as blank control. For the FA-Free control wells, 85 µl of FA-free medium was added plus 5 µl of BSA to keep the consistent BSA concentration in samples containing FAO conjugate. To inhibit FAO, CPT1 inhibitor etomoxir was added into the testing wells 0.5 h before detection. To obtain the maximal mitochondrial OCR, a mitochondrial uncoupler carbonyl cyanide 4-(trifluoromethoxy) phenylhydrazone (FCCP; 0.625 µM) was applied immediately prior to the assay and 10 µl of extracellular O$_2$ consumption reagent was supplemented to all samples. For the blank control wells, 10 µl of FA measurement media was added to replace the extracellular O$_2$ consumption reagent. The spare OCR was calculated by retraction of basal OCR from maximal OCR. FAO-driven OCR was calculated by the disparities of signal change rates between the ET-treated group and shamed control group.

**Western blotting**. The total protein was extracted from cells using RIPA lysis buffer, separated in SDS-PAGE, and transferred onto PVDF membranes. After blocking with 5% milk for 1 h, the membranes were probed with primary antibodies shaking at 4 °C overnight, followed by secondary antibody conjugation at room temperature for 1 h. The membranes were visualized using the ECL Western Blotting Detection system (GE Healthcare) or chemiluminescence western blot detection system (BioRad, Hercules, CA, USA). The following primary antibodies were used: anti-HER2 (C-18, SC-284) and anti-CPT2 (G5, SC-377294) antibodies were bought from Santa Cruze Biotechnology (Santa Cruze, CA, USA). anti-CPT1A (D3B3, Catalog# 12252), anti-OCT4 (C30A3, Catalog# 2840), anti-SOX2 (D6D9, Catalog# 3579) anti-NANOG (D73G4, Catalog# 4903), anti-Histone H3(D1H2, Catalog#4499), and anti-acetylated-Lysine (Catalog#9441) antibodies were from Cell Signaling Technology (Beverly, MA, USA). Anti-ACAD9 antibody was kindly gifted from Dr. Vockley Lab. Anti-CD47 (Catalog# PA2223) antibody were from Boster (Pleasanton, CA, USA). Anti-β-actin (Catalog# A2066) was from Sigma-Aldrich, anti-NF-κB p65 (acetyl K310) (Catalog# ab19870), and anti-Histone H3 (acetyl pan, Catalog# ab47916) antibodies were bought from Abcam (Cambridge, MA, USA).

**Flow cytometry analysis of CD47 or CD133/HER2 positive cells**. Cells that reached 80% confluency were exposed to a single dose of 5 Gy irradiation and were collected after further culture for 15–16 h and the pellets were rinsed with washing buffer (PBS containing 0.5% BSA). Cell pellets were resuspended in pre-diluted FITC mouse anti-human CD47 antibody (BD Pharmingen Catalog #556045) and incubated in room temperature for 1 h in dark, followed by three times washing with washing buffer. For detection of the co-expression of stem cell markers on RR U87 and RR U251, the double staining of PE-CD133 (Miltenyi Biotec, Catalog #130-080-801), and APC-HER2 (R&D Catalog #FAB1129A) were performed. All antibodies were titrated to optimized conditions. The FACS Canto II cytometer (BD) and FlowJo (Three Star, Ashland, OR, USA) were used for analysis. The gating strategies were based on FSC and SSC. The appropriate channels of cell populations, such as unstained and positive control, were gated as standard.

**Clonogenic survival**. The radiosensitivity of RR GBM cells with or without CRISPR knockout of FAO genes or treated with CPT1A inhibitor ET were estimated using the clonogenicity assay. Cells were constantly passaged every two days until reaching 80% confluency. Cells were exposed to irradiation (5 Gy) and 500 cells were following seeded into six-well plates. Both irradiated and control cells were cultured for 10 to 14 days. Colonies were fixed and stained with Coomassie blue. The colony with more than 50 cells were counted. The colony images were obtained by Nikon microscope (Eclipse, E1000M, Japan). The clonogenicity were calculated by the percentages of colonies formed from seeded cells in each group.

**Neurosphere formation**. Parental wild-type and RR GBM cells with or without CRISPR knockouts of FAO genes were subjected to neurosphere formation assessment. For the RR GBM cells without FAO knockout, cells were treated with

CPT1A inhibitor ET (200 µM, 24 h). The ET-treated cells and FAO KO cells were then irradiated with 5 Gy and non-radiated control and irradiated cells were collected and filtered through a 40-µm cell strainer, and 1000 cells suspension were added into 35 mm Petri dish in triplicates. The DMEM/F12 neurosphere medium is supplemented with 1×B27 (Life Technology, Carlsbad, CA, USA), 20 ng/ml EGF (Biovision, Mountain View, CA, USA), 10 ng/ml FGF (Biovison, Milpitas, CA. USA), 5 µg/ml insulin, 1% FBS. Cells were maintained in sphere formation medium for 7–10 days and fresh medium were supplemented every 2 days. The spheres with size exceeding 40 µm were counted and calculated in triplicates.

**Gap filling assay**. Gap filling assay was used for revealing cell migration capacity. In total, $1 \times 10^6$ cells were seeded into six-well plates and when reached 100% confluence, the medium was replaced with a 1% FBS starvation medium and cells were further cultured for 24 h before the gap was created by scraping the dish diagonally with a sterile tip. The cell migration capacity was monitored at 0, 12, 24, and 48 h post scraping. Images were obtained by phase-contrast microscopy, and the gap filling ability of cells was estimated by measuring the gap distance and quantitation using Image J (NIH Image).

**Transwell invasion assay**. Matrigel (Cat #356231, BD Biosciences) was diluted with the coating buffer: 0.01 M Tris (pH 8.0), 0.7% NaCl at a final concentration of 200–300 µg/ml (1:40–45 dilution from stock), and 100 µl of the diluted matrigel were added into the upper chamber of 24-well transwell (Costar catalog #3422, Corning, NY) and incubated at 37 °C for 2 h for gelling. The coating buffer was then removed from the permeable support membrane and the cells to be tested were resuspended in a medium containing 1% FBS at a density of $2.5 \times 10^4$ cells/ 300 µl before adding onto the upper chamber. The lower chamber was then filled with 800 µl of fresh cell culture medium containing 5 µg/ml fibronectin (Catalog #SC-29011, Santa Cruz). Cells were incubated for 48 h and the cell-penetrating capacity was measured with the membrane stained with Diff-Quick Stain kit (K7128, IMEB INC).

**CPT2 enzymatic activity**. CPT2 activity was measured with parental wildtype and RR U251 cells collected in homogenization buffer (150 mM KCL, 5 mM Tris, pH 7.2) followed by 3 s sonication on ice. Then, 200 µg proteins were prepared in 200 µl final volume of homogenization buffer and 20 µl of 20% n-octyl-β-D-glucopyranoside was added into each sample and incubated on ice for 30 min with vortex every 5 min. To initiate reaction, 100 ml of homogenate was transferred into a mixture containing 100 µl of substrate mixture (0.2 mM L-carnitine, 0.013 µCi [$^{14}$C]-carnitine, 0.05 mM palmitoyl-CoA), 700 µl assay cocktail (1.3% BSA, 2 mM KCN, 4.73 mM ATP, 4.2 mM MgCl2, 244 µM glutathione, 100 mM rotenone, and 0.15 M Tris buffer, pH 7.4) and 100 µM malonyl coenzyme (CPT1 fraction bioinhibitor). The blank was prepared with 200 µl of 0.15 M KCL, 700 µl of assay cocktail, and 1 µl of 1.2 M HCl. The reaction mixtures were incubated at 30 °C for 30 min and followed with vortex for 15 sec and terminated by 1 M HCl. 1 mL of 1-butanol was then added, and the sample was centrifuged ($1000 \times g$, 5 min at room temperature) to extract palmitoyl-carnitine (the top layer). About 500 µl of the top layer solution was transferred into a new tube along with 100 µl of H$_2$O followed by centrifugation ($1000 \times g$, 5 min, room temperature), 200 µl of the top layer solution was transferred into scintillation vials with 4 ml of scintillation cocktail and counted in a Multi-Purpose Scintillation Counter (Beckman). Each sample was measured for 5 min, and the results were normalized by the corresponding protein concentrations.

**Oil red O staining and quantitation**. To evaluate the level of lipid accumulation in RR GBM cells, the oil red staining method was applied to WT versus RR U87 and U251 cells. In total, $3 \times 10^4$ cells were seeded in each well of 96-well plates. After incubation for 12 h, cells were treated with 250 µM free FA oleate (oleic acid: palmitate acid = 2:1) for 24 h. Cells were rinsed with PBS twice and were fixed with 4% paraformaldehyde for 30 min at room temperature followed by washing with dH$_2$O twice. About 50 µl of Oil red working solution (oil red o stock solution: dH$_2$O at ratio 6:4) was added in dark for 15 min incubation at room temperature. Then the Oil Red O solution was removed and 50 µl of 60% isopropanol was added

immediately in each well for 20 s at room temperature. Cells were washed 2–5 times with dH2O until excess stains were moved completely. The oil red dye was eluted with 100 µl DMSO and incubated for 10 min with gently shaking. The lipid accumulation results were detected using the fluorescence microplate spectrophotometer (Molecular Devices) at 510 nm. Images were obtained under a Nikon microscope (Eclipse, E1000M, Japan).

**THP1 phagocytosis assay**. In vitro phagocytosis was conducted using the human monocyte THP1 cells that were differentiated into macrophages by incubation with 40 nM Phorbol 12-myristate 13-acetate (PMA) for 48 h. The human GBM cell lines were stably transfected with GFP. GBM-GFP cells ($1$–$1.5 \times 10^6$) were cocultured with macrophage ($1 \times 10^6$) per well in six wells plate for 2–4 h at 37 °C. Cocultured cells were collected and washed with 0.5% BSA-PBS. Cell pellets were incubated with fluorescence APC/Cyanine 7 labeled CD11b primary antibody (Catalog #101225, BioLegend) in dark for 30 min, followed by three times washing with 0.5% BSA-PBS. The phagocytic activity was analyzed by flow cytometry (Becton Dickinson canto II, BD, NJ, USA) and FlowJo (Three Star, Inc., Ashland, OR, USA).

**Mouse bone marrow-derived macrophages**. Bone marrow cells were collected from the femur of mice and passed through a 70 µm cells strainer. The cell suspension was then centrifuged at 1500 rpm for 5 min. The supernatant was discarded, and cells were then seeded into a six-well plate cultured for 7 days in a macrophage complete medium (DMEM/F12 growth medium, 10%FBS, 10mM L-glutamine, and 100 U/ml M-CSF). On day 3, cells were fed with another 1 ml macrophage complete medium. On day 7, cells were washed with PBS and stimulated with a macrophage complete medium containing lipopolysaccharide (LPS, 20 ng/ml) for 48 h.

**Mouse bone marrow-derived macrophage phagocytosis**. In vitro phagocytosis of mouse GBM cells was conducted using primary mouse bone marrow-derived macrophages that were differentiated into macrophages by incubation with 20 ng/ml LPS for 48 h. GFP-labeled tumor cells (GL261 cells, $1$–$1.5 \times 10^6$) were cocultured with macrophage ($1 \times 10^6$) for 2–4 h at 37 °C. Cocultured cells were collected and washed with 0.5% BSA-PBS. Cell pellets were incubated with anti-CD11b (APC-Cy7) antibody in dark for 30 min, followed by three times washing with 0.5% BSA-PBS. The phagocytic activity was analyzed by flow cytometry (Becton Dickinson canto II, BD, NJ, USA) and FlowJo (Three Star, Inc., Ashland, OR, USA).

**Immunocytochemistry (ICC)**. To identify the co-expression of CD47 and CPT1A, CPT2, or ACAD9 in RR U251 cells, cells grown on round coverslips with 60–80% confluence were washed one time with PBS and fixed in 4% paraformaldehyde (pH 7. 2) for 20 min. After three times washing with PBS, cells were permeabilized with 0.2% Triton X-100 in PBS incubation at room temperature for 5 min. Samples were then blocked for 30 min with 1% BSA and co-incubated with primary antibodies derived from different species (anti-mouse CD47 or anti-rabbit CPT1A, CPT2, or ACAD9) at 1:50–250 dilutions overnight at 4 °C. Followed by three times washing with PBS, secondary antibodies (anti-mouse conjugated with Alex 488 or anti-Rabbit conjugated with APC) were added for incubation. Appropriated dilution (1:500 in 5% BSA) was selected and cells were incubated for 1 h at room temperature. Images were obtained, and the fluorescence intensity was quantified using fluorescent microscopy using Axiovision software (Zeiss, Germany). Positive cells were quantified by Image-Pro Plus 6.0 (Media Cybernetics, MD, USA).

**RT-qPCR**. Total RNA was extracted from cells using the Trizo method (Invitrogen). cDNA was synthesized by Superscript III first-strand cDNA synthesis kit (Invitrogen) according to the manufacturer's instruction. RT-qPCR was performed with the SYBR Green PCR master mix (Thermo Fisher Scientific) on a 7500 Fast Real-Time PCR machine (Applied Biosystems/Thermo Fisher Scientific). The following primers were used:

h*CD47* forward: 5′AGAAGGTGAAACGATCATCGAGC3′
h*CD47* reverse: 5′CTCATCCATACCACCGGATCT3′
*GAPDH* forward: 5′ GGACTCATGACCACAGTCCAT 3′
*GAPDH* reverse: 5′ GTTCAGCTCAGGGATGACCTT 3′
*β-actin* forward: 5′CATGTACGTTGCTATCCAGGC3′
*β-actin* reverse: 5′CTCCTTAATGTCACGCACGAT3′.

Each sample was repeated in triplicate settings independent three times. Relative gene expression was calculated after correction for GAPDH and β-actin expression using the $2^{-\Delta\Delta Ct}$ method.

**Luciferase reporter assay**. Luciferase reporter (pGL2-basic-*CD47*) driven by human *CD47* promotor and the control reporter with NF-κB binding sequence (769-757: GTGGAAGCTCCCT) deleted (pGL2-basic-*CD47*-ΔNF-κB) were constructed. The sequence deletion was conducted with Pfu Turbo DNA polymerase (Stratagene, La Jolla, CA) with the following PCT primers: *CD47* NF-κB Delta F: 5′GTGGTCGGGTACCTGCCCGCTCGCCCCTCGCGGGCTCTGCG3′, *CD47* NF-κB Delta R: 5′CGCAGAGCCCGCGAGGGGCGAGCGGGCAGGTACCCGACCAC3′. For transactivation analysis, GBM cells with or without FAO gene knockout

(WT, $CPT1A^{-/-}$ $CPT2^{-/-}$ $ACAD9^{-/-}$) were seeded in triplicate 96-well plate (8000 cells/well) and co-transfected PolIII-Renilla control reporter with control NF-κB luciferase reporter, pGL2-basic-*CD47* or pGL2-basic-*CD47*-ΔNF-κB for 48 h using transfection reagent TurboFect (Thermo Scientific, USA, catalog #R0533). Different treatments (citrate or SB204990) were conducted with the reporter-transfected WT GBM cells for varied time periods and cell lysates were prepared by adding 100 µl of lysis buffer (DTT 1 mM, MgCl2 8 mM, EGTA 4 mM, PMSF 100 nM) and incubated for 15 min at room temperature followed by adding 10 µl of lysates with 50 µl of luciferase buffer (5 mM DTT, 100 nM ATP, 150 µg/ml CoA) and 500 nM luciferin. The luciferase activity was measured using the Turner TD20/20 with Renilla luciferase for normalization transfection efficiency.

**CRISPR/Cas9-mediated FAO gene edition**. The Lenti-CRISPRv2 vector was purchased from the Addgene plasmid repository (Addgene cat #52961) (https://www.addgene.org). Single guide RNAs were designed using CRISPR design software (http://crispr.mit.edu) from Zhang's lab. Oligos corresponding to the sgRNAs were synthesized and cloned into a lenti-CRISPRv2 vector following the protocol from Addgene. The human sgRNA sequences are shown as follow:

h*CPT1A* F: CACCGCTCCGGACGGGATTGACCTG
h*CPT1A* R: AAACCAGGTCAATCCCGTCCGGAGC
h*CPT2* F: CACCGCGGGGCCCCGCGGTTGGTCC
h*CPT2* R: AAACGGACCAACCGCGGGGCCCCGC
h*ACAD9* F: CACCGCTTGCCTAAACTGGCGTCCG
h*ACAD9* R: AAACCGGACGCCAGTTTAGGCAAGC
sgRNA sequencing primer: GAGGGCCTATTTCCCATGATT

Lentiviral particles were packaged in HEK293T cells according to the protocol from Addgene. GBM cells were plated at $1.25 \times 10^5$ per well in 12-well plates. After 16 h, the medium was replaced with 0.5 ml lentiviruses and 0.5 ml fresh medium containing 10 ng polybrene (Sigma-Aldrich, Catalog #H9268-10G) for 6–8 h and then add an additional 0.5 ml fresh medium. After 12 h incubation, cells were exchanged to fresh medium and kept culturing for another 48 h. The infected cells were selected with 0.3 µg/ml puromycin for 1 week and confirmed by Western blot.

**Cycloheximide chase assay**. Chase assay was performed following the published protocols[91]. Briefly, $5 \times 10^6$/well RR U251 cells were seeded into a six-well plate and after overnight incubation, the cultured medium was replaced by a fresh cell culture medium with or without A-485 (20 µM, 24 h). Cells were then treated with 150 µg/ml cycloheximide and time-dependent changes in the level of CD47 were analyzed by Western blotting.

**Citrate detection**. Cellular citrate concentration was measured by using the EnzyChromTM citrate Assay kit (Cat #ECIT-100, BioAssay Systems, Hayward, CA, USA). Briefly, $2 \times 10^6$ cells were homogenized in 100 µl PBS and centrifuged at 14,000 rpm for 5 min and the clear supernatants were collected as the citrate sample. The standards of citrate were prepared by diluting citrate in dH2O with a concentration range from blank to 400 µM. For detection of citrate concentration, the working reagent mix containing 85 µl Developer, 1 µl CL Enzyme, 1 µl ODC Enzyme, and 1 µl Dye was added into 20 µl of each standard or sample well. The sample without CL Enzyme was served as blank control. Citrate concentration was measured by using the fluorescence microplate spectrophotometer (Molecular Devices) at 570 nm and results were calculated as [Citrate] = R(sample) – R(blank /Slope ($\mu M^{-1}$) × n (µM), n = dilution factor.

**Apoptosis analyses**. Apoptotic cell death was measured in cells with different treatments or with the CRISPR knockout of FAO genes (CPT1A, CPT2, or ACAD9). Briefly, cells ($2.5 \times 10^5$) were seeded in a six-well plate and cultured overnight followed by 5 Gy irradiation treatment. The irradiated cells were further cultured for 48 h, and the attached cells and cells suspended in the medium were all collected and washed twice with 1× binding buffer (10 mM HEPES/NaOH, pH 7.4, 150 mM NaCl, 5 mM KCl, 1 mM MgCl2, 1.8 mM CaCl2). Cells were stained using Annexin-V/PI kit (Biosource, Invitrogen, Carlsbad, CA, USA) and apoptotic cells were analyzed by flow cytometry (Becton Dickinson canto II, BD, NJ, USA). Data were analyzed using Flowjo software (Three Star, Inc., Ashland, OR, USA).

**Preparation for anti-mouse CD47 antibodies**. Protein G Resin from GenScript (L00209) was applied for the preparation and purification of mammalian monoclonal and polyclonal IgG. Following standard column purification procedures, rat anti-mouse CD47 antibody IgG2α was purified from the supernatants of hybridoma MIAP301. The purified IgG was then concentrated using Protein Concentrator PES (88516) from Pierce. The final antibody concentration was determined using the absorbance at OD280 by NanoDrop ND-1000 spectrophotometer.

**Syngeneic subcutaneous mouse GBM tumor model**. The animal use and care protocol applied in this test including local tumor radiation and ET and antibody treatment was approved by the Institutional Animal Use and Care Committee of the University of California Davis (IACUC 15315). For the test of palmitate (PA) consumption rate in RR GBM tumors, syngeneic mouse GBM tumors were

**Table 1 CD47 antibody binding and macrophage-mediated phagocytosis.**

| Dye/ channel name | Channel color | Light source intensity (%) | Exposure time (ms) |
|---|---|---|---|
| Rhodamine red | Red | 50.0 | 300 |
| Alexa Fluor 488 | Green | 50.0 | 35 |
| DAPI | Blue | 50.0 | 10 |

subcutaneously generated with RR GL261 cells ($5 \times 10^6$ 100 µl PBS) inoculated into 6 weeks old female C57BL/6 mice (Taconic) in both flanks ($n = 10$; total tumor number = 20). When tumor volumes reached ~200 mm³, PA was intratumorally administered (100 µl, 25 µM) for two treatments, one day apart, with 100 µl of 0.05% BSA-PBS injected into the other tumor of the same mouse used as the solvent control. Tumor volumes were calculated as V = Length × width²/2.

**Human GBM tumors regrown model**. To detect the enhancement of FAO enzymes and CD47 in human regrown tumors after radiation, 6 weeks old female nude mice ($n = 4$; The Jackson Laboratory, Stock No: 002019) were applied as the tumor-bearing animal with $5 \times 10^6$ WT U251 cells inoculated subcutaneously in the right flanks of mice. When tumor volumes reached ~200 mm³ radiation treatment was delivered (2 Gy/day for continuously 5 days; total dose = 10 Gy). Reduction of tumor volumes were detected after radiation and regrown to about 200 mm³ by day 28. The tumors without radiation treatment were applied as the control. Expression of FAO enzymes and CD47 in control and regrown tumors was detected by Western blot and IHC staining.

**Mouse orthotopic GBM regrown model**. Mouse orthotopic tumor was established using WT GL261 cells transduced with lentivirus vector pCCLc-MNDU3-LUC-PGK-EGFP-WPRE following the reported protocol. Briefly, C57BL/6 mice (female, 6 weeks) from Taconic were anesthetized with ketamine (70 mg/kg) and xylazine (12 mg/kg). An incision longitudinally in the middle of the scalp was made followed by drilling the parietal bone to generate a 4 µm diameter hole located 1.5 mm posterior to the bregma and 1.5 mm to the right of the sagittal suture. GL261 cells ($2.5 \times 10^5$) in 5 µl PBS were intracranially injected through an entry site to a depth of 2.5 mm by a 10 µl Hamilton syringe. On day 10 when tumor image was detected, all mice were divided into two groups with or without radiation ($n = 5$). Radiation was delivered under anesthesia with 3 Gy on days 10, 12, and 14 and the control group received anesthesia without radiation. The tumor growth was in vivo monitored with BioSpec 7 T MRI scanner (Bruker, German) on day 10, day 17, and day 28, respectively after cell inoculation. The irradiated group was terminated at day 28 and the control group was ended at day 17 when tumor sizes reached the maximal.

**Inhibition of orthotopic tumors regrown by blocking FAO and CD47**. The mouse GBM orthotopic tumors regrown after radiation was used to investigate the synergetic effect of radiation combined with anti-CD47 antibody, etomoxir (ET), or a combination. When tumors were detected by in vivo imaging on Day 10, all tumors were irradiated with 3 Gy/day, every other day, for three times, total radiation dose = 9 Gy. Then mice were divided into four groups for further treatment starting on Day 24 as Group 1, IgG 16 mg/kg ($n = 5$); Group 2, anti-CD47 antibody 16 mg/kg ($n = 5$), Group 3, ET 30 mg/kg ($n = 5$); Group 4, combination anti-CD47 antibody and ET ($n = 7$). All reagents were intraperitoneally administered every 2 days for a course of 2-week treatments with tumor monitored by in vivo imaging. Mice weight and clinical symptoms were observed and recorded every other day until the endpoint of the experiment. The whole-brain tissues were collected once euthanasia criteria were reached[92] for the analysis of antibody penetration and macrophage-mediated phagocytosis.

**CD47 antibody binding and macrophage-mediated phagocytosis**. For immunofluorescence staining, orthotopic GFP expressing GBM model-derived tumor tissues were fixed in 4% paraformaldehyde and FFPE sectioned at 5 µm thickness were prepared. Antigen retrieval was applied in citrate buffer (pH 6.1) at 95 °C for 40 min in microwave and then flush dH₂O. Slides were blocked in 10% normal horse serum for 1 h at room temperature and then incubated with a primary antibody cocktail of Rabbit anti-rat IgG (1:500) and Mouse anti-GFP IgG (1:1000) with 10% normal horse serum. After washing three times with 1% BSA in PBS, slides were incubated with a secondary antibody cocktail of Alexa Fluor 488 conjugated Goat anti-mouse IgG and Rhodamine Red-X-Conjugated goat anti-rabbit (1:500 diluted in 10% normal horse serum) followed by a quick 1% Sudan black staining. Cell nuclei were counterstained using Vectashield Antifade solution containing DAPI (Vector Laboratories, Burlingame, CA, USA). For detection of macrophage-mediated phagocytosis, the primary antibody cocktail contained anti-GFP (1:100; cell signaling) and anti-CD11b (1:100; Millipore) and the secondary antibodies were 1:250 diluted fluorescent conjugated (anti-mouse-APC for CD11b and anti-rabbit-Alexa Fluor 488 for GFP). Fluorescent 14-bit images were obtained

with the Zeiss Axioscan.Z1 slide scanning system (Carl Zeiss Microscopy, Oberkochen, Germany) using a Plan-Apochromat 20x/0.8 M27 objective with standard acquisition settings. The light source intensity and exposure time per dye were described as below (Table 1):

The positive cells were quantified by Image-Pro Plus 6.0. The phagocytic index was calculated by the following formula: Phagocytic index = (number of engulfed cells/total number of macrophages containing engulfed cells) × (total number of macrophages containing engulfed cells/total number of counted macrophages) × 100.

**Quantification and statistical analysis**. Data generated by in vitro and in vivo mouse tumor models in this study are presented as mean ± SD and analyzed using the two-tailed Student $t$-test for two groups or ANOVA for multiple groups. The statistical significance of Kaplan–Meier survival curves was assessed with a Mann–Whitney test. A value of $P$ less than 0.05 was considered statistically significant. *$P < 0.05$, **$P < 0.01$, ***$P < 0.001$, and **** $P < 0.0001$.

**Reporting summary**. Further information on research design is available in the Nature Research Reporting Summary linked to this article.

## Data availability
The RNA sequencing data referenced during the study are available in a public repository (Part B, DataSet ID: mRNAseq_693) from CGGA website (http://www.cgga.org.cn/download/20191128/download.jsp).

The data that support the findings of this study are within the Article, Supplementary Information, or available from the corresponding author upon reasonable request. Source data are provided with this paper. Requests for resources and reagents should be directed to Jian Jian Li (jijli@ucdavis.edu). Source data are provided with this paper.

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

## Acknowledgements

We acknowledge Dr. Irving Weissman for providing the anti-human CD47 antibody Hu5F9-G4, Dr. William Frazier for providing anti-mouse CD47 MIAP301 hybridoma, Dr. Watkins Vockley for providing anti-ACAD9 antibody, Dr. Orin Bloch for discussion of in vivo mouse GBM model, and Dr. Ji Ming Wang for reading the manuscript. The clinical and pathological studies conducted in Xiangya Hospital, Central South University, were supported by the National Foundation of Natural Science (81672509). Dr. Nian Jiang was supported by China Scholarship Council (201706370117). The radio-resistant GBM cell lines and in vitro and in vivo studies using radioresistant mouse GBM cells carried out at the Cancer Center of University of California Davis were supported by NCI RO1 CA213830 (J.J.L.), NCI R01 CA224900 (H.-W.C.), VA MERIT Award I01BX004271 (H.-W.C.), UC Davis Immunology Pilot Grant Support (J.J.L.), and UC Davis Cancer Center supported by the CCSG Grant awarded by the National Cancer Institute (Dr. Primo Lara; NCI P30CA093373). This work is dedicated to the memory of Dr. William C. Dewey for his mentoring and long-term friendship. "A mentor is someone who sees more talent and ability within you, than you see in yourself, and helps bring it out of you"—Bob Proctor.

## Author contributions

N.J., B.X., S.X., L.-Q.S., Xj.L. A.M.M, W.J.M., H.-J.K., K.S.L., H.-W.C., and J.J.L. conceived the project. N.J., B.X., W.X., M.F., J.H., W.Z., P.F.W., M.A.C., and A.W. conducted experiments on in vitro and in vivo investigation. N.J., M.F., Y.H., Y.D., A.C.E., Sh.X., and B.N.D. conducted on in vitro and in vivo imaging analysis. R.C.F., X.L., Xj.L., N.Y., S.W., W.C., and D.J. conducted clinical sampling and analysis. X.L, N.Y., and S.W conducted on bioinformatics. N.J., B.W., Sh.X., K.S.L., H.-W.C., and J.J.L. wrote the manuscript. All authors commented on the manuscript.

## Competing interests

The authors declare no competing interests.
