## [Peer Review File · Nature Communications]

Fatty acid oxidation fuels glioblastoma radioresistance with CD47-mediated immune evasionREVIEWER COMMENTS

Reviewer #1 (Remarks to the Author); expert on macrophage immunosurveillance:

Jiang and colleagues, in this manuscript, profiled the change in fatty acid metabolism of the RR GBM cells, and a possible role the rewired metabolism plays in supporting immune surveillance escape via upregulating CD47 expression. GBM is one of the hardest-to-treat malignancies with limited treatment choice and the one of worst prognosis expectancy. It is of great value, both scientifically and translationally, to illustrate the variances taken place in the refractory GBM tumors and to investigate the defensive mechanism.

The experiments regarding the impact of radiation towards the metabolic rewiring of GBM tumor cells are generally well designed and properly controlled. Radiation activates NFkB and subsequently upregulate CD47 expression is previously reported also by the same group, though in another cancer indication.

The rationale of combining anti-CD47 antibody and inhibition of FAO for GBM treatment, however, is not well justified. If inhibition of FAO has already dramatically decreased CD47 expression, why would a combination of CD47 blockade be necessary? The effectiveness of antibodies blocking CD47 would be attenuated on cells with low or no expression of CD47. The author needs to better interpret this. In addition, there are multiple discrepancies reflected from data collected from various assays or by different technical means from the very same single in vivo efficacy experiment. Together, the in vivo data are difficult to interpret.

Major:

1. There is a lack of rationale for the authors to investigate the expression of CD47 in association of FAO. All the data shown by authors in fig. 1 and fig. S1 combined supports the claim that fatty acid metabolism is rewired by RR GBM cells facilitating an aggressive growth. However, none of them connects fatty acid metabolism and macrophage phagocytosis checkpoint. It would be very helpful to describe what prompts the authors to investigate CD47 expression.
2. The authors proposed, via the data presented in fig.4a-4d, that the citrate-mediated NF-kB/RelA acetylation positively regulates the CD47 transcription. However, it would be helpful to further support this claim if the authors could directly prove acetylation-disabled RelA mutants are incapable to initiate the CD47 transcription in the corresponding assays, and transcription initiation is rescued upon restoration of wildtype RelA.
3. For the in vitro phagocytosis assay, PMA stimulated THP1 cells are used as a surrogate of primary macrophages. However, it would be stronger evidence if the authors could also examine the in vitro phagocytosis using primary human macrophages to co-culture with RR U251 and RR U87, as they are more sensitive to CD47 induced phagocytosis inhibition. Preferably, since the authors used murine syngeneic model in their in vivo efficacy experiment, it would be equally informative if the authors study the phagocytosis effect of bone-marrow derived macrophages and GL261 tumors. Either way, adopting primary macrophages in tumor cell phagocytosis assay is more persuasive than using THP-1 derived surrogates.
4. The pharmacological inhibition of FAO by Etomoxir is unlikely to selectively target GBM cells while sparing macrophage, and FAO is involved in mitochondria energy metabolism. It

is necessary to investigate the overall in vitro phagocytosis efficacy of primary human macrophages in the presence of etomoxir, instead of only etomoxir pre-treated tumor cells.

5. Both the Western Blot and flowcytometry data in fig.3 and fig. S4. suggested that wildtype glioblastoma cell lines expressed minimal level of CD47, which seems to be a contradictory to previously publication (eg, PMID: 29308321, PMID: 32198351, PMID: 33329583, PMID: 30602457). CD47 overexpression by tumor cells, including GBM, is critical for them escape innate immune surveillance, while disrupting CD47-SIRP α axis would agitate microglia/macrophage phagocytosis to destroy tumor cells. It is hard to understand that GBM cells with such low initial CD47 expression (as shown in fig.3) does not translate into better prognosis, comparing to the RR counterpart post radiation which is not only more progressive (fig. 5g and 5h) but also more immuno-inhibitory (much higher CD47 expression). In fact, as shown in Fig.2k, CD47 low GBM cohort shows a much higher OS rate against CD47 high cohort.

6. The data in Fig.6a are quite confusing. According to the authors, ET treatment downregulate CD47 expression to an almost undetectable level (eg. Fig5c). In this case, a reasonable expectation would be, on top of ET treatment, an addition of CD47-blocking antibody should have minor effects on phagocytosis given that these cells express CD47 at neglectable level. But an additive/synergistic phagocytosis was observed in the combination treatment – the authors need to interpret this observation. Can this experiment be performed with RR GL261 and mouse macrophages as well, given RR GL261 is the line used for in vivo experiment?

7. To confirm the anti-tumor effects observed in the in vivo experiments using ET and anti-CD47 are resulted from macrophage phagocytosis, a macrophage depletion assay should be included (eg. Treatment with clodronate or anti-CSF1R).

8. It is also to the reviewer's curiosity that whether the overall outcome of GL261 orthotopic xenograft with ET + CD47 mAb treatment would be superior to radiation followed by ET + CD47 mAb. Given that the GL261 tumor cells, as suggested by the authors, are of low CD47 expression, they should be susceptible to macrophages/microglia phagocytosis. With the ET and anti-CD47 combination treatment, the wt GL261 growth should be further suppressed. Given that the RR GL261 is 3-fold more aggressive, it will be very interesting to see the overall therapeutic outcome of ET and anti-CD47 combination on wildtype GL261, compared to the radiation treated group shown in fig.6b.

9. The description of experiments in fig.7b are not clear. Did the authors collect tumors from each group and stained with anti-rat IgG? If this was the case, then the staining from the ET group is certainly low because the mice didn't receive CD47 antibody treatment and thus there was no CD47 antibody in the tumor. The experiment procedure needs to be described in more details for the readers to better understand the results.

10. The CD11b staining intensity shown in fig.8a reflected the myeloid cell infiltration. Both the ET and CD47 mono-treatment seems to mildly, if any, improved the myeloid compartment infiltration. It is surprisingly that with the ET + anti-CD47 combination treatment, suddenly a multiple-fold increase of CD11b compartment infiltration is observed. An interpretation is needed for such much improved immune infiltration.

11. There seems to be a lack of consistency regarding the tumor volume and survival data, making the in vivo experiments difficult to interpret.

Bio-imaging was performed every 7 days before the initiation of the treatment (day 24) but stopped during the treatment period for 14 days, until the completion of the experiment (day 38). Can the authors explain why this experiment is designed this way?

In addition, according to the figure legend, the data of Fig. 6d and Fig. 7a are from the same experiment. However, according to fig. 6d, all the mice stayed alive till at least day 38 (day 14 after treatment), but according to fig. 7a, majority of the mice in the IgG group and some of the mice in the CD47/ ET groups died before day 14 after treatment. The authors should explain this discrepancy.

12. The images in fig. 8 are unclear and not convincing. It's difficult to recognize phagocytosis on these images. The authors should also interpret what criterion they have used to define phagocytosis based on these images.

Minor point:

1. It is misleading that while the first sentence (line 33-35) says radiation can boost the efficacy of immune checkpoint blockade, the immediate second sentence (line 35-37) says radioresistant GBM cells are more aggressive and less responsive to immune checkpoint blockade.
2. The CD47 expression intensity as shown in fig. 2e is not in line with that in fig.3 and fig. S4, in which a majority of the primary GBM specimen expresses high level of CD47.
3. Line 179 and line 180 seems to be redundant.
4. Fig.5e-h, it would be easier to interpretate if the x-axis label Ctrl could be replaced by RR or others.
5. Line 241 to 244. This sentence can be rewritten for better clarity.
6. For fig. 3i, the authors should give more details regarding how phagocytosis index was quantified. According to the images shown here, comparing to the sham radiated tumors in which more CD11b positive cells were detected but only two of them overlapped with GFP cells, the only red (CD11b) cell shown in regrow is overlapping with GFP cells, suggesting an actually highly phagocytosis intensity.

Reviewer #2 (Remarks to the Author); expert on metabolism and genetic regulation:

This study by Jiang and colleagues examines metabolic features of radioresistant glioblastoma cells, linking increased fatty acid oxidation to RelA acetylation and expression of CD47, thereby promoting immune evasion by suppressing macrophage phagocytosis. In general the observations are interesting, and promising in vivo effects of targeting FAO and CD47 are shown. Thus, the manuscript has potential for high significance. I have some concerns about the rigor of some of the mechanistic data that would need to be addressed, however.

1. The presentation of many of the metabolic data in Figure 1 is unclear. More details are needed to understand what is being shown.
 - a. Lactate data in Figure 1a. How is the data normalized? To cell #? Protein content? Additionally, lactate abundance in cells on its own is not sufficient to draw conclusions about glycolysis. Ideally, glucose uptake and lactate production in culture medium over time should be assessed.
 - b. Why does palmitate increase ATP levels at lower concentrations and suppress ATP at higher concentrations? This may be due to toxicity due to exposing cells to high amounts of

a saturated fatty acid. Authors should consider repeating experiments using a mix of palmitic and oleic acids to avoid toxicity.

c. What is the assay shown in Figure 1e? What lipid is measured?

d. In figures f and g, some of the statistics reported look questionable to me. For example, visually, in panel f, it does not look like A172 FIR and FIR ET conditions are significantly different, but $p < 0.01$ is reported. I recommend showing individual data points and double checking the statistical analysis.

2. Authors never directly demonstrate that RR cells are conducting more FAO, which is a core conclusion of the paper. This should be tested using radioactive or stable isotope tracers.

3. The pathway from fatty acids to regulation of RelA acetylation and CD47 expression needs to be more thoroughly tested. Does fatty acid supplementation promote and inhibition of FAO suppress RelA acetylation? Can ChIP experiments be used to examine NF- κ B binding to CD47 locus- is this impacted by fatty acid availability? Does NF- κ B inhibition block CD47 upregulation? Does acetate supplementation rescue RelA acetylation and CD47 expression upon ACLY inhibition or FAO inhibition?

4. It should also be tested whether the effects of FAO on downstream phenotypes are dependent on ACLY-dependent acetyl-CoA production and NF κ B. Does inhibition of NF κ B or ACLY impact phenotypes such as phagocytosis or neurosphere formation?

Minor:

- Positive and negative controls should be included in the NF- κ B luciferase assay
- Fig. 4j: when reporting an increase in RelA and H3 acetylation, total RelA and histone H3 should be shown as controls.
- Manuscript should be edited for grammar/ English language.

Reviewer #3 (Remarks to the Author); expert on glioblastoma:

In this manuscript, Jiang et al present an interesting story linking the re-wired metabolism of radiation resistant GBM cells to decreased immunogenicity. Here, they show that RR cells have an increase of FAO enzymes which allow for the acetylation of NF κ B subunits, increasing CD47 expression – and that increases of these proteins give rise to a worse OS in GBM patients. Inhibition or genetic manipulation of the FAO enzymes reduced cell growth, and increased phagocytosis – where dual targeting of FAO and CD47 greatly increased mouse OS and phagocytosis. Overall, this is an intriguing story that may assist in developing better immunotherapy combinations in GBM.

Other comments:

1) Figure 1f should be read “RR” and “RR ET” instead of FIR, and Figure 1d tumor volume is graphed as if they are matched tumors. However, it wasn’t stated in the text that these were primary/recurrent tumors, and may be better graphed more clearly. Sup 2h is missing the “RR+ET label”.

2) In Sup Figure 5d, the correlation between CD47 and the NF κ B targets is greater in primary, as compared to recurrent tumors. However, in the primary group there seems to be two sub-groups with lo/lo and hi/hi expression, it would be interesting to see if there was a

survival difference between these two subgroups, strengthening the idea that increased CD47/NFkB targets leads to worse OS.

3) ET treatments over 200 μ M have been shown to also have an inhibitory effect on Complex I of the ETC. Do you think this may be an explanation to ET having a greater effect than a CPT1A $-/-$ model (ex. Fig 4f).

4) Figure 4g and 4i should also be done in the WT models to show that this pathway is not being affected in the WT endogenously, and further show that the RR cells have been remodeled and this is why they are able to respond to excess citrate.

5) Line 241 has a fragmented sentence.

6) I think line 259 is supposed to reference Fig. 8a, b – instead of 7.

7) Since mitochondrial function is being studied in this manuscript, it would be advantageous to determine if there is a difference between the overall numbers of mitochondria between the WT and RR cells.

8) CPT1A has also been shown to complex with VDAC1, which may also be playing a role in the CPT1A $-/-$ lines, as well as ET treated cells.

9) Backgrounds on IB blotting images need to be adjusted. Many of them are too dark and some of these blots were adjusted too much, in particular contrast and whiteness. Thickness of the frame of some of these blots are inconsistent. Some of these frames thickness need to be reduce to 0.75 or 0.5 mm.

Dear Referees,

We thank you for granting us an extended period of time to complete the suggested experiments and to all reviewers for their thoughtful comments. We have thoroughly revised the manuscript with new data generated from the suggested experiments, which has significantly raised the overall scientific quality of our findings. Among the revised portions, new data confirmed the potential function of etomoxir on anti-CD47 function using primary macrophages as suggested by Reviewer #1, specificity of CD47 upregulation to radioresistant cells compared to the wild type potential counterparts (Reviewer #1, #3), and the metabolic insights of FAO mediated CD47 expression (Reviewers #1, #2). The revised manuscript (new data and described marked in red and all other rewritten parts are marked in blue) now contains a total of 89 data panels and 78 in the revised Supplementary Figures and Supplementary Tables. For your convenience, a portion of the newly added data presented in the Response Letter is attached at the end of Response Letter as Response Figures and Response Tables.

Reviewer #1 has raised an important question of the rationale of how this work was proposed. In the revised manuscript, as suggested by Reviewer #1, we have added the reports for proposing the current study and previous evidence which prompted us to propose the hypothesis that the metabolic shifting in radioresistant cancer cells links FAO metabolism with enhanced immune evasion due to FAO-mediated CD47 expression. During the revision period, two new observations have been reported that are supportive to our conclusions. One reported that fatty acid metabolism accelerates the incidence of breast cancer brain metastasis (*Nat. Cancer*, 2021)[1] and the other described that glycolysis is a less-essential uptake for GBM metabolism (*Nature*, 2021)[2]. CD47 is one of major immune checkpoint receptors expressed in many human cancers including GBM [17, 18] and anti-CD47 and radiation demonstrated synergistic anti-tumor responses. Our group also reported CD47 overexpression in radioresistant breast cancer cells (*Nat Comm*, 2020)[3]. These findings motivated us to hypothesize that enhanced FAO in radioresistant cancer cells not only boosts ATP output for cell proliferation but also activates the anti-macrophage phagocytotic response of tumor cells, which coordinatively contribute to the aggressive behavior GBM cells. Thus, the data presented in our revised manuscript reveals a previously unknown mechanism that tumor anti-phagocytotic function can be enhanced by mitochondrial FAO metabolism, and the crosstalk between CD47 and FAO serves as a new approach to eliminating radioresistant-antiphagocytotic GBM. These statements have been added into the revised Introduction.

Another concern by Reviewer #1 was related on the in vitro phagocytosis assay with PMA stimulated THP1 cells which has the disadvantage of malignant cell phenotype. We have repeated the phagocytotic experiments using human peripheral blood monocyte derived macrophages and mouse bone marrow derived macrophages. Due to pandemic, we could only obtain a limited amount of human blood and human macrophages not enough for testing phagocytosis. An alternative approach, as suggested by Reviewer #1, we successfully obtained a sufficient amount of macrophage from mouse bone marrow monocytes. Using mouse macrophages, we confirmed a ~1.7 fold enhancement of phagocytosis in RR GL261 cells with CRISPR CPT2 KO cells compared to the counterpart CPT2 WT RR GL261 cells. Using the same mouse macrophages on GL261 cells, phagocytosis was enhanced by ET treatment in RR GL261 cells but not in WT GL261 cells. Thus, these new data support our findings that radioresistant GBM cells enhanced FAO with CD47 mediated anti-phagocytotic function. These new data have been added into the revised data and manuscript.

A rigor in deep analysis of the rewired metabolism in radioresistant cancer cells, *i.e.*, glycolysis-to-FAO metabolism, a potential critical metabolic dynamic in acquired tumor tolerance to anticancer therapy, was suggested by Reviewer # 2. The measured lactate data and enhanced mitochondrial energy outputs may not directly reflect the nature of fatty acid metabolic rewiring in the radioresistant GBM cells. In response to this concern, we have conducted the experiment of glucose uptake in radioresistant U251, U87, A172, GL261 cells compared to their counterpart WT cell lines by application of the glucose assay kit (A-114, BRSC). We further measured the acetyl CoA level presented in WT and RR U251, U87 and compared with ET treated (200 μ M, 48h) and CPT1A^{-/-} status. In addition, directly FAO metabolic measurement using radioactive FAO assay, suggested by Reviewer #2, was conducted using radioactive isotope tracers ([9,10-³H(N)]-palmitic acid [0.5 μ Ci (~9.3 pmol)]) and cellular FAO activity was determined by quantifying the conversion of ³H palmitic acid to ³H₂O over 6 hours, which demonstrated an enhanced FAO activity in four wild type GBM cells (U251, U87, A172, GL261).

Reviewer #3 suggested to determine a potent adverse effect of etomoxir (ET) treatments with the high concentration on potential normal tissue injuries or effects on macrophage polarization that may weaken the anti-tumor phagocytotic function. This concern also mentioned by Reviewer #1 is potential critical in future clinical trial of ET/antiCD47 therapy. Our current study with relatively a high increased ET concentration may involve FAO unrelated effects but mouse *in vivo* systematically administration of ET showed an increased macrophage infiltration and phagocytosis, indicating that ET-mediated FAO inhibition suppresses CD47 expression and enhances immune cell attacks. We have repeated experiments presented in Figure 4g and 4i supporting that ET at low concentration increases macrophage activity whereas high ET concentration may weaken macrophage phagocytotic function. Additionally, we have added new data generated using mitochondria targeted GFP vector indicated an enhanced mitochondrial population in the RR GBM cells.

Among the other revised portions, we have added new data to confirm the potential function of etomoxir on anti-CD47 function using primary macrophages as suggested by Reviewer #1, specificity of CD47 upregulation to radioresistant cells compared to the wild type potential counterparts (Reviewer #1, #3), and the metabolic insights of FAO mediated CD47 expression (Reviewers #1, #2). Additionally, acetyl CoA mediated protein stabilization on CD47 is founded and included into the revised manuscript.

Thank you very much for your consideration.

Sincerely yours,

Jian Jian

Jian Jian Li, MD, PhD
Professor
Department of Radiation Oncology
University of California Davis School of Medicine
4501 X Street, Suite G0140
Sacramento, CA 95817, USA
jjli@ucdavis.edu

Reviewer #1 (Remarks to the Author); expert on macrophage immunosurveillance: Jiang and colleagues, in this manuscript, profiled the change in fatty acid metabolism of the RR GBM cells, and a possible role the rewired metabolism plays in supporting immune surveillance escape via upregulating CD47 expression. GBM is one of the hardest-to-treat malignancies with limited treatment choice and the one of worst prognosis expectancy. It is of great value, both scientifically and translationally, to illustrate the variances taken place in the refractory GBM tumors and to investigate the defensive mechanism.

The experiments regarding the impact of radiation towards the metabolic rewiring of GBM tumor cells are generally well designed and properly controlled. Radiation activates NF- κ B and subsequently upregulate CD47 expression is previously reported also by the same group, though in another cancer indication.

The rationale of combining anti-CD47 antibody and inhibition of FAO for GBM treatment, however, is not well justified. If inhibition of FAO has already dramatically decreased CD47 expression, why would a combination of CD47 blockade be necessary? The effectiveness of antibodies blocking CD47 would be attenuated on cells with low or no expression of CD47. The author needs to better interpret this. In addition, there are multiple discrepancies reflected from data collected from various assays or by different technical means from the very same single in vivo efficacy experiment. Together, the in vivo data are difficult to interpret.

Response: We thank the Reviewer for the careful review and the insightful comments. We greatly appreciate the positive comment that this work may reveal an important crosstalk between radiation-induced tumor metabolic rewiring and immunosuppressive gene expression in radioresistant GBM. We have added an array of new experimental results supplement the original findings.

Major:

1. There is a lack of rationale for the authors to investigate the expression of CD47 in association of FAO. All the data shown by authors in fig. 1 and fig. S1 combined supports the claim that fatty acid metabolism is rewired by RR GBM cells facilitating an aggressive growth. However, none of them connects fatty acid metabolism and macrophage phagocytosis checkpoint. It would be very helpful to describe what prompts the authors to investigate CD47 expression. I think change Fig2 with Fig1, use clinical data to Introduce the relationship between CD47 and FAO, and then show in RR cell with FAO relation. Therefore, RR cell through FAO metabolic changes influence CD47 expression alteration.

Response: The order of Fig. 1 and Fig. 2 has been exchanged and the Introduction has been revised to illustrate the background driving the idea of tumor FAO metabolism activating CD47-mediated immunosuppression. We have reported that oxidative respiration in mitochondria can be instantaneously adjusted in mammalian cells to meet the energy consumption for fueling cell cycle progression and DNA repair [4, 5]. Recently we demonstrated that burning saturated fat (i.e., palmitates, the bad fat in diet) by mitochondrial FAO may even improve mitochondrial homeostasis (*Dev. Cell*, 2020)[6]. These findings shed new lights on the current concept that glycolysis solely provides tumor energy metabolism which agrees the findings of FAO enhancement in cancer cells capable of surviving anti-cancer genotoxic conditions[7-11]. Recently, fatty acid metabolism has been shown to increase the incidence of breast cancer brain metastasis (*Nat. Cancer*, 2021)[1] and glycolysis is found to be a less-essential uptake for GBM metabolism (*Nature* 2021)[2], both are supportive to our current findings. We showed that CD47 is overexpressed in radioresistant breast cancer cells (*Nat Comm*,

2020)[3]. In the Introduction, we have added the rationales for the hypothesis that enhanced FAO in radioresistant cancer cells increases ATP output by GBM cells with anti-phagocytotic capacity.

2. The authors proposed, via the data presented in fig.4a-4d, that the citrate-mediated NF- κ B/RelA acetylation positively regulates the CD47 transcription. However, it would be helpful to further support this claim if the authors could directly prove acetylation-disabled RelA mutants are incapable to initiate the CD47 transcription in the corresponding assays, and transcription initiation is rescued upon restoration of wildtype RelA.

Response: We agree that an acetylation deficient RelA will further support our finding that FAO-mediated acetylation activates NF- κ B/RelA-regulated CD47 transcription. Via database analysis, we found Rel protein contains several acetalization domains which have been targeted to study NF- κ B transactivation [12, 13]. Following the reported procedures, we purchased the plasmid of RelA mutants with disabled KAT specific acetylation. The T7-RelA(K310R) plasmid was obtained from Addgene (cat. 23250). Acetylation at lysines 310 regulates RelA's DNA binding activity. The mutation of RelA at K310R directly affected its

transcription activity of CD47 promoter. After co-transfecting with CD47 promoter luciferase reporters in the radioresistant GBM cells, compared to KAT acetylation WT RelA, CD47 transcriptional activity was markedly suppressed in cells transfected with the KAT acetylation Mut RelA or treated with CBP/p300 acetalization inhibitor A-485 (**Response Figure 1a; Revised Figure 4k**). In agreement with the inhibited CD47 transcription, CD47 mRNA levels were also markedly (or remarkably) reduced by transfection with mut RelA or treated with A-485 (**Response Figure 1b; Revised Figure 4l**). These results together with the original findings support the conclusion that NF- κ B/RelA acetylation is enhanced by FAO-derived acetyl-CoA. In addition, we found that CD47 protein also contains the CBP/p300 acetalization domains which may be responsive to FAO-enhanced acetyl-CoA and prolongs CD47 protein stability further enhancing tumor cell anti-phagocytotic activity.

c (SI Fig 8a)

Human CD47 potential Acetylation sets			
KAT	Site	Sequence	P-value
CBP/p300	166	GQFGIKTLK ^R YRSGGMDE	0.0395
CBP/p300	163	LFWGQFGIKTLK ^R YRSGG	0.5121
CBP/p300	175	YRSGGMDEK ^R TIALLVAG	0.884
p300	290	QLLGLVYMK ^R FVE-----	1

Response Figure 1. (a) Radioresistant GBM cells transfected with CD47 promoter-controlled luciferase activity were either treated with A-485 (20 μ M, 24 hrs), CBP/p300 inhibitor, or co-transfected with KAT-specific acetylation WT or Mut RelA. (b) CD47 mRNA level in RR U251 and RR U87 cells treated with A-485 (20 μ M, 24 hrs) or transfected with KAT-specific acetylation WT or mutant RelA. (c) Predicted KAT-specific acetylation sites in human CD47 protein. (d) Co-immunoprecipitation assays with WT U251 and RR U251 cells transfected with P300 with or without A-485 (20 μ M, 24 hrs). (e) Degradation of CD47 protein in RR U251 cells with or without A-485 treatment on indicated time point. n = 3; **P < 0.01, ****P < 0.0001.

Indeed, co-immunoprecipitation assay demonstrate that acylated CD47 proteins were elevated in RR U251 cells which can be enhanced by transfection of p300 and inhibited by A-485 (**Response Figure 1c, 1d, 1e; SI Figure 8a,8b,8c**). FAO-mediated CD47 protein stability is an interesting finding which is now added to SI figure and will be further investigated.

3. For the in vitro phagocytosis assay, PMA stimulated THP1 cells are used as a surrogate of primary macrophages. However, it would be stronger evidence if the authors could also examine the in vitro phagocytosis using primary human macrophages to co-culture with RR U251 and RR U87, as they are more sensitive to CD47 induced phagocytosis inhibition. Preferably, since the authors used murine syngeneic model in their in vivo efficacy experiment, it would be equally informative if the authors study the phagocytosis effect of bone-marrow derived macrophages and GL261 tumors. Either way, adopting primary macrophages in tumor cell phagocytosis assay is more persuasive than using THP-1 derived surrogates.

Response: We have repeated the phagocytosis experiments using human peripheral blood monocyte-derived macrophages and mouse bone marrow-derived macrophages. Due to the SARS-COV2 pandemic, we only obtained a limited amount of human blood and macrophages insufficient for testing phagocytosis experiments. However, as suggested by the reviewer, a sufficient number of macrophages was generated from mouse bone marrow monocytes. Using these cells, we confirmed that phagocytosis

was increased by ~1.7 fold in RR GL261 cells with CRISPR-mediated CPT2 KO compared to the WT CPT2 control RR GL261 cells (**Response Figure 2a; SI Fig. 10d**). Consistently, the phagocytosis of RR GL261 was reduced by mouse macrophages and RR U251 and RR U87 cells by THP1 human phagocytis cells (**Response Figure 2b; Revised Fig. 3d**). Using the same mouse macrophages, phagocytosis was enhanced by ET treatment of RR GL261 cells but not of WT GL261 cells (**Response Figure 2c; Revised Fig. 5c**). This was recaptured in RR U251 cells with increasing gradients of ET concentrations (**Response Figure 2d; Revised Fig. 5b**) or in RR U251 cells with CRISPR KO FAO enzymes CPT1A, CPT2 and ACAD9 (**Response Figure 2e; Revised Fig. 5e**). However, phagocytosis of human (U251) or mouse (GL261) GBM cells were decreased by FAO activator L-carnitine, low-dose palmitate or IR treatment (**Response Figure 2f;**

Response Figure 2. Phagocytosis of mouse macrophage on mouse GBM GL261 cells. (a) Macrophage phagocytosis on CPT2^{-/-} RR GL261 cells compared with RR GL261 cells. (b) Macrophage phagocytosis on WT and RR GBM cells (U251, U87, GL261). (c) Macrophage phagocytosis on WT and RR GL261 cells treated with ET (80uM,24 h). (d)Enhanced macrophage phagocytosis on RR U251 cells treated with indicated ET concentrations (24 h). (e) Macrophage(THP1) phagocytosis on vector control and CRISPR-KO CPT1A^{-/-}, CPT2^{-/-}, ACAD9^{-/-} RR U251 cells detected by flow cytometry. (f) WT U251 and GL261 cells treated with L-carnitine (L-car, 10 mM, 48 h), palmitate 974 (PA, 25 μM, 48 h) or IR (5 Gy, 16 h). n = 3; *P < 0.05, **P < 0.01, *** P < 0.001, ****P < 0.0001

Revised Fig. 5a). These new data support our findings that radioresistant GBM cells enhanced FAO with CD47-mediated anti-phagocytotic function. The new data have been added to the revised manuscript (page 10, lines 212-215,218-220).

4. The pharmacological inhibition of FAO by Etomoxir is unlikely to selectively target GBM cells while sparing macrophage, and FAO is involved in mitochondria energy metabolism. It is necessary to investigate the overall in vitro phagocytosis efficacy of primary human macrophages in the presence of etomoxir, instead of only etomoxir pre-treated tumor cells.

Response: We appreciate this critical question. In fact, mice systematically administrated with ET demonstrated that ET treatment enhanced phagocytosis of GL261 tumor cells compared to cells not treated with ET (**Revised Figure 8a, b**). In the literature, ET has been shown to dose-dependently affect macrophage polarization which was enhanced with low concentration but reduced with high concentration of ET [14, 15]. To ascertain if macrophage-mediated phagocytosis is affected by ET, we tested a dose range of 1-12 μM . In agreement with the reported findings, we found that phagocytosis of parental WT GL261 cells was enhanced by ET-pretreated macrophages with ET concentrations at 1-3 μM (24 h) whereas reduced when macrophages were pretreated with 6 μM and 12 μM ET (**Response Figure 3; SI Fig. 9c**).

Response Figure 3. Phagocytosis on WT GL261 cells mediated by mice macrophage in presence of indicated ET concentrations for 24 hrs. n=3; * $P < 0.05$, **** $P < 0.0001$; ns= not significant

5. Both the Western Blot and flowcytometry data in fig.3 and fig. S4. suggested that wildtype glioblastoma cell lines expressed minimal level of CD47, which seems to be a contradictory to previously publication (eg, PMID: 29308321, PMID: 32198351, PMID: 33329583, PMID: 30602457). CD47 overexpression by tumor cells, including GBM, is critical for them escape innate immune surveillance, while disrupting CD47-SIRPa axis would agitate microglia/macrophage phagocytosis to destroy tumor cells. It is hard to understand that GBM cells with such low initial CD47 expression (as shown in fig.3) does not translate into better prognosis, comparing to the RR counterpart post radiation which is not only more progressive (fig. 5g and 5h) but also more immuno-inhibitory (much higher CD47 expression). In fact, as shown in Fig.2k, CD47 low GBM cohort shows a much higher OS rate against CD47 high cohort.

Response: Thanks for the insightful comments. We totally agree that CD47 expression level is increased in many cancers and is defined as a potential target in cancer immunotherapy. However, the expression levels are usually compared to the counterpart normal tissues or organs. The results shown in Fig. 3 is a comparison of CD47 levels in radioresistant GBM vs parental wild type GBM cells and can be markedly enhanced by radiation, indicating a potential inducible property of CD47 expression. Such a dynamic and inducible high level of CD47, although being beneficial to anti-CD47 antibody targeting, could be detrimental due to the potential exhaustion of macrophages and requirement of raising antibody concentrations to eliminate tumor cells, which may be similar to the reported resistance to PD-1 immunotherapy in PD-L1-expressing tumors. Thus, CD47 protein levels measured

by Western blot is a relative expression level compared to control cells and data from flowcytometry analysis of CD47 positive cells (**Revised Fig. 3b**) should be more accurate. The data shown in Fig. 3 and Fig. 5 together with the western blot are consistent indicating that radioresistant GBM cells overexpressed a higher level of CD47 with enhanced anti-phagocytosis and aggressive growth behavior. These results are also in agreement with metadata analysis shown in **Revised Fig. 1k** that reduced CD47 expression is linked to a better patient prognosis. Furthermore, it is consistent with clinical observation that recurrent GBM patients suffer from poor prognosis with shorter PFS and OS (median PFS of 1.5–6 months and median OS of 2–9 months[16-18]).

6. The data in Fig.6a are quite confusing. According to the authors, ET treatment downregulates CD47 expression to an almost undetectable level (e.g. Fig.5c). In this case, a reasonable expectation would be, on top of ET treatment, an addition of CD47-blocking antibody should have minor effects on phagocytosis given that these cells express CD47 at neglectable level. But an additive/synergistic phagocytosis was observed in the combination treatment – the authors need to interpret this observation. Can this experiment be performed with RR GL261 and mouse macrophages as well, given RR GL261 is the line used for in vivo experiment?

Response: Thanks for the constructive comments. As indicated in question 5 by the reviewer and our response above, we believe that although CD47 level was almost undetectable in ET treated RR U251 cells by Western blot (**Revised Fig. 5c**), ET could not totally block CD47 expression and nor effectively eliminate tumor cells by ET-mediated phagocytosis. The results are supported by our repeated experiments with ET treatment of RR U251 cells followed by detecting CD47-expression cells by flowcytometry. The results show that there are a small number of cells still expressing CD47 in the ET treated RR U251 cells (**Response Figure 4a; Revised Figure 5d**). We also noticed that there is still a small fraction of CD47-expressing cells in FAO KO U251 cells (**Revised Fig. 5d**). Similarly, even in FAO gene KO cells, there was still a certain level of CD47 promoter activity detected (**Revised Fig. 4c**) with a substantial CD47 mRNA (**Revised Fig. 4d**). Thus, it appears that although FAO markedly enhances CD47 expression and its anti-phagocytotic function, blocking FAO could not totally eliminate FAO-induced CD47 expression, which may explain the synergistic tumor inhibition by ET+anti-CD47. In addition, with the suggestion by the reviewer, we have conducted in vitro phagocytosis test with RR GL261 and mouse macrophages, which shows an increased phagocytosis rate by mouse macrophages of RR but not of WT GL261 cells (**Response Figure 4b; Revised Fig. 5c**).

7. To confirm the anti-tumor effects observed in the in vivo experiments using ET and anti-CD47 are resulted from macrophage phagocytosis, a macrophage depletion assay should be included (e.g. Treatment with clodronate or anti-CSF1R).

Response: We fully understand the suggested experiments. GBM regrowth model of in vivo radiotherapy in mice with systematic depletion of macrophages could be ideal to further strengthen our findings. However, it appears that, currently, our lab is able to establish such a mouse model due to the uncertainty in mouse survival rate with orthotopic GBM regrowth after local tumor irradiation. Additional difficulty is related to the fact that phagocytosis of GBM in brain tissue is mediated not only by myeloid cells but also by microglia [19] which could be hard completely deplete. However, this is a model we should pursue in future studies.

Response Figure 4. (a) Upper panel: Western blot of CD47 expression in control and ET treated (200 μ M, 48 h) WT U251 cells compared to CRISPR-mediated CPT1A^{-/-}, CPT2^{-/-}, ACAD9^{-/-} of RR U251 cells. Lower panel: Percentage of CD47 expressing cells detected by flow cytometry in CPT1A^{-/-}, CPT2^{-/-}, ACAD9^{-/-} RR U251 populations or ET treated RR U251 cells. **(b)** Macrophage phagocytosis on GL261 cells with and without ET treatment (80 μ M, 24 h). n=3; **** $P < 0.0001$; ns= not significant

8. It is also to the reviewer's curiosity that whether the overall outcome of GL261 orthotopic xenograft with ET + CD47 mAb treatment would be superior to radiation followed by ET + CD47 mAb. Given that the GL261 tumor cells, as suggested by the authors, are of low CD47 expression, they should be susceptible to macrophages/microglia phagocytosis. With the ET and anti-CD47 combination treatment, the wt GL261 growth should be further suppressed. Given that the RR GL261 is 3-fold more aggressive, it will be very interesting to see the overall therapeutic outcome of ET and anti-CD47 combination on wildtype GL261, compared to the radiation treated group shown in fig.6b.

Response: This is a very valid question. We proposed this work based on the clinical observations that many GBM are originally sensitive to radiation but gradually become radioresistant, leading to treatment relapse. Based on this, primary untreated GBM were not our focus of the study and ET may not be an effective agent to enhance CD47-mediated phagocytosis resistance if combined with RT to treat primary GBM. Other groups have also demonstrated that ET or anti-CD47 therapy has limited effects on inhibition of the progress of glioma [20, 21]. In agreement with this, results shown in **Revised Fig 2** indicate that FAO activity is relatively low in untreated wild type GBM cells compared to RR GBM cells. Thus, ET treatment of tumor cells may not be very effective to induce significant macrophage-mediated phagocytosis. Indeed, new data shown in above **Response Figure 4b (Revised Fig. 5c)** demonstrate that macrophage phagocytosis is enhanced by ET treatment of RR GL261 but not of WT GL261 cells. Based on these results, it appears that the current settings of ET/anti-CD47 fit the goal to treat the recurrent and radioresistant GBM more than targeting primary tumors that are glycolysis dominant.

9. The description of experiments in fig.7b are not clear. Did the authors collect tumors from each group and stained with anti-rat IgG? If this was the case, then the staining from the ET group is certainly low because the mice didn't receive CD47 antibody treatment and thus there was no CD47 antibody in the tumor. The experiment procedure needs to be described in more details for the readers to better understand the results.

Response: Thanks for the insightful comments. This experiment is conducted to prove that CD47 antibody systematically administrated is capable of penetrating the blood brain/tumor barrier and can effectively bind to CD47-expressing tumor cells. **Revised Fig. 7b** showed representative immunofluorescence images of penetrated anti-CD47 antibody in GFP-labeled GL261 orthotopic tumors in the groups of anti-CD47, anti-CD4+ET treatments (**Revised Fig.7b**). Therapeutic anti-CD47 antibody was rat anti-mouse CD47 antibody IgG2 α purified from the supernatants of hybridoma MIAP301. Tumor penetration and binding of the anti-CD47 antibody were identified by rabbit anti-rat IgG as primary antibody and Rhodamine Red-X-Conjugated goat anti-rabbit as secondary antibody which stained as red color. The mouse anti-GFP antibody and Alexa 488 conjugated goat anti-mouse secondary antibody recognized GFP expressing GL261 tumor cells which stained as green color. Thus, binding of the anti-CD47 antibody to tumor cells was indicated as yellow color. Tumor tissues from control groups of GBM bearing mice treated with rabbit anti-rat IgG and ET are visualized with freely distributed rabbit IgG and background images, respectively. We have added more details in the Fig. legend and the experimental procedures.

10. The CD11b staining intensity shown in fig.8a reflected the myeloid cell infiltration. Both the ET and CD47 mono-treatment seems to mildly, if any, improved the myeloid compartment infiltration. It is surprisingly that with the ET + anti-CD47 combination treatment, suddenly a multiple-fold increase of CD11b compartment infiltration is observed. An interpretation is needed for such much improved immune infiltration.

Response: We appreciate this important suggestion. Although a substantial amount of tumor infiltrating CD11b⁺ cells is observed on sectioned tumor slides, the overall phagocytosis measured by flowcytometry was not proportional to the slides indicating the heterogeneity of macrophage infiltration of tumor tissues . The phagocytic index was calculated by the following formula: Phagocytic index = (number of engulfed cells/total number of macrophages containing engulfed cells) \times (total number of macrophages containing engulfed cells/total number of counted macrophages) \times 100. We have added statement in the revised Results.

11. There seems to be a lack of consistency regarding the tumor volume and survival data, making the in vivo experiments difficult to interpret. Bio-imaging was performed every 7 days before the initiation of the treatment (day 24) but stopped during the treatment period for 14 days, until the completion of the experiment (day 38). Can the authors explain why this experiment is designed this way? In addition, according to the figure legend, the data of Fig. 6d and Fig. 7a are from the same experiment. However, according to fig. 6d, all the mice stayed alive till at least day 38 (day 14 after treatment), but according to fig. 7a, majority of the mice in the IgG group and some of the mice in the CD47/ ET groups died before day 14 after treatment. The authors should explain this discrepancy.

Response: Thanks for pointing out the issue. We have reviewed our experimental procedures and added more details in the Methods (*Inhibition of orthotopic tumors regrown by blocking FAO and CD47*) and Fig legend to in vivo experiments illustrated in **Fig. 6b**. (page 34, lines 768-772).

12. The images in fig. 8 are unclear and not convincing. It's difficult to recognize phagocytosis on these images. The authors should also interpret what criterion they have used to define phagocytosis based on these images.

Response: We have improved the quality of the images and added details in calculating the macrophage-mediated phagocytosis in the revised Methods. To measure the phagocytosis,

macrophages were incubated with anti-CD11b (1:100; Millipore) as the primary antibody and the secondary antibodies were 1:250 diluted anti-mouse-APC for CD11b which marks macrophages in red. Tumor cells were incubated with anti-GFP (1:100; Cell Signaling) as the primary antibody and the secondary antibodies were 1:250 diluted anti-rabbit-Alexa Fluor 488 for GFP. The nuclei were stained with DAPI. Macrophage phagocytosis of tumor cells are indicated in yellow fluorescence. The phagocytotic cells (yellow fluorescence) were quantified by Image Pro Plus 6.0. The phagocytic index was calculated by the following formula: Phagocytic index = (number of engulfed cells/total number of macrophages containing engulfed cells) \times (total number of macrophages containing engulfed cells/total number of counted macrophages) \times 100. These results have been added in the revised Methods (page 35, lines 782-792).

Minor point:

1. It is misleading that while the first sentence (line 33-35) says radiation can boost the efficacy of immune checkpoint blockade, the immediate second sentence (line 35-37) says radioresistant GBM cells are more aggressive and less responsive to immune checkpoint blockade.

Response: We have revised the Abstract and corrected the sentences.

2. The CD47 expression intensity as shown in fig. 2e is not in line with that in fig.3 and fig. S4, in which a majority of the primary GBM specimen expresses high level of CD47.

Response: Data shown in **Fig. 2e and Fig. S4g** were generated from clinical GBM patients with primary and recurrent tumors after radiotherapy, thus more closely reflecting the in vivo tumor CD47 expression levels. Data shown in **Fig. 3** were generated from established radioresistant GBM cells lines and in vivo irradiated mouse GBM regrown tumors, which showed different expression of CD47 than clinical tumors. This difference may reflect the variations between human samples and experimental mouse models.

3. Line 179 and line 180 seems to be redundant.

Response: This sentence has been deleted.

4. Fig.5e-h, it would be easier to interpretate if the x-axis label Ctrl could be replaced by RR or others.

Response: This label has been indicated by IR- or IR+ with 5 Gy.

5. Line 241 to 244. This sentence can be rewritten for better clarity.

Response: This sentence has been revised.

6. For fig. 3i, the authors should give more details regarding how phagocytosis index was quantified. According to the images shown here, comparing to the sham radiated tumors in which more CD11b positive cells were detected but only two of them overlapped with GFP cells, the only red (CD11b) cell shown in regrow is overlapping with GFP cells, suggesting an actually highly phagocytosis intensity.

Response: We have added detailed procedures in similar experiments shown in **Fig. 8** and improved the image in **Fig 3i**.

Reviewer #2 (Remarks to the Author); expert on metabolism and genetic regulation: This study by Jiang and colleagues examines metabolic features of radioresistant glioblastoma cells, linking increased fatty acid oxidation to RelA acetylation and expression of CD47, thereby promoting immune evasion by suppressing macrophage phagocytosis. In general, the observations are interesting, and promising in vivo effects of targeting FAO and CD47 are shown. Thus, the manuscript has potential for high significance. I have some concerns about the rigor of some of the mechanistic data that would need to be addressed, however.

Response: We very appreciate the insightful evaluation and suggestions. We completely agree that the features of rewired FAO metabolism in radioresistant GBM cells are to be rigorously investigated, which is important not only for validating our current findings but also for developing effective combined modality for GBM therapy. During this revision period, although some experiments were delayed due to pandemic, an array of tests including the required radioactive labeled FAO metabolism has been completed or repeated. Data generated from these experiments are consistent with the observations that, compared to the parental untreated wild type GBM cells, tumor metabolism is indeed rewired from glycolysis-dominant to FAO-overriding pathway in the radioresistant GBM cells and the regrown mouse orthotopic tumors following a dominancy after radiation. Not surprisingly, these results are supported by two new reports from other research groups indicating that fatty acid metabolism accelerates brain metastasis of breast cancer (*Nature Cancer*, April, 2021)[1] and that intratumoral glucose pickup is less-essential for tumor cells compared to infiltrating immune cells (*Nature*, April, 2021)[2]. Likewise, our results demonstrate a highly flexible FAO metabolism adoptable by cancer cells surviving genotoxic irradiation. This further supports that inhibition of FAO may affect more tumor cells than infiltrating T cells and monocytes, a potential advantage of ET treatment of radioresistant GBM without affecting the function of infiltrating immune cells required for anti-CD47 therapy. This is a question also raised by Reviewer 1 and our added experiments deonstrated a dose-dependent ET-enhanced macrophage function. We have added the new information in the revised Discussion (page 14, lines 306-311).

1. The presentation of many of the metabolic data in Figure 1 is unclear. More details are needed to understand what is being shown.
 - a. Lactate data in Figure 1a. How is the data normalized? To cell #? Protein content? Additionally, lactate abundance in cells on its own is not sufficient to draw conclusions about glycolysis. Ideally, glucose uptake and lactate production in culture medium over time should be assessed.

Response: We appreciate the insightful comments and suggested experiments. We have added more details on experimental design and data analysis. The lactate level was measured with culture medium and the results were normalized by cell numbers of each dish. We also agree that lactate abundance is not sufficient to reflect the glycolysis in tumor cells. Based on the suggestion, we have conducted experiments of glucose uptake in culture medium generated from parental WT and RR U251, U87, A172, GL261 cells by using Glucose Assay Kit (A-114, BRSC). The results show that glucose uptake rates were reduced in RR GBM compared to parental WT GBM cells (**Response Figure 5a; Revised Figure 2e**). In addition, to reflect the enhanced FAO metabolites in RR GBM cells, we have conducted the experiments of acetyl CoA generation in WT versus counterpart RR U251, U87 cells and compared

to ET mediated FAO inhibition (200 μ M, 48hrs) and CPT1A^{-/-}, CPT2^{-/-}, and ACAD9^{-/-} RR U251 cells. As shown in the **Response Figure 5**, acetyl CoA generation was enhanced in RR GBM cells compared to WT GBM cells (**Response Figure 5b; Revised Figure 2d**), and FAO- enhanced acetyl CoA levels in RR U251 cells were markedly suppressed by ET treatment as well as in RR U251 cells with CRISPR KO FAO genes CPT1A, CPT2 and CACD9 (**Response Figure 5c; SI Figure 7a**). These results provide strong evidence for a metabolic shift acquired by radioresistant GBM cells from glycolysis-dominant to FAO-centered metabolism in radioresistant GBM cells.

a (Fig 2e)

b (Fig 2d)

c (SI Fig 7a)

Response Figure 5. (a) Glucose uptake measured in culture medium of WT and RR U251, U87, A172, GL261 cells (n=3 per cell line). (b) The concentration of Acetyl CoA which is the final production of FAO presented in WT and RR GBM cells (U251, U87, A172, GL261). (c) Acetyl CoA level presented in RR U251 cells compared with ET treated (200 μ M, 48hrs) and CPT1A^{-/-}, CPT2^{-/-}, ACAD9^{-/-} RR U251 cells. n = 3; * P < 0.05, ** P < 0.01, *** P < 0.001, **** P < 0.0001.

b. Why does palmitate increase ATP levels at lower concentrations and suppress ATP at higher concentrations? This may be due to toxicity due to exposing cells to high amounts of a saturated fatty acid. Authors should consider repeating experiments using a mix of palmitic and oleic acids to avoid toxicity.

Response: Thanks for the question. In the revised manuscript, we have added more details of experimental procedures published previously (*Developmental Cell*) [6]. In Fig. 1 (now Fig. 2b in revised version), we showed a different response to palmitate-induced mitochondrial metabolic activation with more enhanced ATP in RR cells compared to WT cells in relative low palmitate concentrations (12.5 μ M, 25 μ M). However, the lipotoxicity to both RR and WT cells was observed when PA concentration was increased by > 50 μ M. These results were not only consistent with our

previous findings that palmitate, the saturated long-chain fatty acid (the bad fat from diet), is beneficial for mitochondrial metabolism and function when normal liver cells were treated with a certain low concentration plus BSA compared to treated with BSA alone[6]. We also observed, agreed with previous reports in the literature that high increased concentration of palmitate (>100 μM) was lipotoxic which was a harsh challenge and beyond not physiological limits. Thus, we suggest that 25 μM is an appropriate concentration for determining the effects of FAO on the metabolism and aggressive growth of glioblastoma cells. We took the liberty to cite the published results on the lipotoxic effects in all normal tissues attached as Response Table 1 at the end of this Response Letter.

c. What is the assay shown in Figure 1e? What lipid is measured?

Response: We have added details for the experiments shown in Figure 1e (**Revised Figure 2i**) which reflect the boosted lipid turnover rate in the cytoplasm of radioresistant GBM cells. The results demonstrated that if lipid digestion rate is low, there will be lipid accumulation which was demonstrated in RR GBM cells compared to WT GBM cells. Following the standard procedure [22, 23], cells were pretreated with 250 μM free FA oleate (oleic acid: palmitate acid=2:1) for 24 hrs followed by Oil Red staining. Oil Red is a fat-soluble azo dye which is a potent fat-soluble reagent and staining agent. It specifically stains neutral lipids in cells or tissues [22, 23]. Following the reported protocol, the oil red dye was diluted with 100 μl DMSO and incubated for 10 min with gently shaking. The lipid accumulation levels were then measured using fluorescence microplate spectrophotometer (Molecular Devices) at 510 nm. Images were obtained under Nikon microscopy (Eclipse, E1000M, Japan). The description has been added to the Methods of revised manuscript (page 27-28, lines 613-623).

d. In figures f and g, some of the statistics reported look questionable to me. For example, visually, in panel f, it does not look like A172 FIR and FIR ET conditions are significantly different, but $p < 0.01$ is reported. I recommend showing individual data points and double checking the statistical analysis.

Response: . We have replotted the data (**Revised Figure 2i, j**), revised the figures and now showed individual data points for each parameter. As some of the data in Figure 1f (**Revised Figure 2i**) showed substantial variations, we repeat the experiments and recalculated OCR as shown in the following figure.

2. Authors never directly demonstrate that RR cells are conducting more FAO, which is a core conclusion of the paper. This should be tested using radioactive or stable isotope tracers.

Response: We agree that FAO rewiring is an important conclusion of this study but more experiments should be conducted as supports. However, we have measured CTP2 enzymatic activity in WT and RR U251 cells by monitoring ^{14}C - palmitoyl-carnitine which demonstrates the higher FAO activity in RR U251 cells (**Response Figure 6a; SI Figure 3a**). On the basis of the suggestion, we have conducted FAO assay using radioactive isotope tracer ([9,10- ^3H (N)]-palmitic acid [0.5 μCi (~9.3 pmol)]) in cultured RR GBM and WT GBM cells. The cellular FAO activity was determined by quantifying the conversion of ^3H palmitic acid to $^3\text{H}_2\text{O}$ over 6 hrs as shown in the following **Response Figure 6b** and as **Revised Fig. 2c added to the revised manuscript**. The FAO rates are significantly raised in RR GBM cells compared to WT GBM cells in each pair of tumor cell line, supporting enhanced FAO in radioresistant GBM cells. The results have been added in the Results of revised manuscript (page 6, line 135-139).

Response Figure 6. (a) CTP2 enzymatic activity in WT and RR U251 cells measured by monitoring ^{14}C - palmitoyl-carnitine (n = 4). **(b)** FAO activity quantified by the conversion of ^3H palmitic acid to $^3\text{H}_2\text{O}$ over 6 hours using radioactive isotope tracers ([9,10- ^3H (N)]-palmitic acid [0.5 μCi (~9.3 pmol)]) (n=3 per cell line) in parental WT and RR GBM cells (U251, U87, A172, GL261). * $P < 0.05$, ** $P < 0.01$, *** $P < 0.001$).

3. The pathway from fatty acids to regulation of RelA acetylation and CD47 expression needs to be more thoroughly tested. Does fatty acid supplementation promote and inhibition of FAO suppress RelA acetylation? Can ChIP experiments be used to examine NF- κB binding to CD47 locus- is this impacted by fatty acid availability? Does NF- κB inhibition block CD47 upregulation? Does acetate supplementation rescue RelA acetylation and CD47 expression upon ACLY inhibition or FAO inhibition?

Response: Thanks for the suggestion. We have conducted experiments to show that the acetylation domain in RelA is required for NF- κB -mediated CD47 promoter activation which is altered by adding FAO product citrate. Our results showed that FAO product citrate induced NF- κB and CD47 promoter-driven luciferase activity in RR U251 and RR U87 cells (**Revised Figure 4g, 4h**), but not in WT GBM cells (**Response Figure 7a; SI Fig 6e**). QRT-PCR showed that CD47 expression is increased once the cells were treated in 1mM citrate for 6hrs, which is reversed by the ACLY inhibitor SB204990 (25uM,24hrs) (**Response Figure 7b; SI Fig 7c**). To explore the underlying mechanism, we used A-485 which inhibits the acetylation of RelA to treat RR U251 and RR U87 cells before citrate incubation. The results demonstrate an increasing trend induced by citrate which can be blocked by A-485 not only in NF- κB promoter-driven luciferase activity (**Response Figure 7c; SI Fig 7d**) but also in acetylation-RelA protein level (Response Figure 7d; **SI Fig 7e**). Furthermore, we transfected the cells with plasmids containing KAT specific acetylation disabled RelA mutants, and co-transfected with CD47 promoter luciferase reporters in radioresistant GBM cells. Compared to KAT acetylation of WT RelA, CD47 transcriptional activity was markedly suppressed in cells transfected with the KAT acetylation Mut RelA or treated with CBP/p300 acetalization inhibitor A-485 [24-26] (**Response Figure 7e; Revised**

Figure 4k). In agreement with the inhibited CD47 transcription, CD47 mRNA levels were also remarkably reduced by transfection of the cells with mut RelA or treatment with A-485 (**Response Figure 7f; Revised Figure 4k**). These results further support the conclusion that NF- κ B/RelA acetylation is enhanced by FAO derived acetyl-CoA.

Response Figure 7. (a) CD47 promoter-controlled luciferase activity with or without NF- κ B motif deletion measured in WT U251 and WT U87 cells treated with citrate (1 mM, 6 hrs). (b) CD47 mRNA level in WT U251 and WT U87 cells treated with citrate (1 mM, 6 hrs) with or without ACLY inhibitor SB204990 (25 μ M, 24 hrs). (c) Luciferase reporter activity driven by NF- κ B in RR U251 and RR U87 cells treated with with citrate (1 mM, 6 hrs) with or without A-485 (20 μ M, 24 hrs). (d) Western blot of RelA-K310 acetylation, CD47 in RR U251 and RR U87 cells treated with citrate (1 mM, 6 h) with or without A-485 (20 μ M, 24 hrs). (e) CD47 promoter-controlled luciferase activity in RR U251 and RR U87 cells treated with A-485 (20 μ M, 24 hrs), CBP/p300 inhibitor, or co-transfected with KAT-specific acetylation WT or Mut RelA. (f) CD47 mRNA level in RR U251 cells treated with A485 (20 μ M, 24 hrs) transfected with WT or mutant RelA mut. (f) Western blot of RelA-K310 acetylation, CD47 in RR U251 and RR U87 cells treated with citrate (1 mM, 6 h) with or without A-485 (20 μ M, 24 h). * P < 0.05, ** P < 0.01, *** P < 0.001, **** P < 0.0001.

4. It should also be tested whether the effects of FAO on downstream phenotypes are dependent on ACLY-dependent acetyl-CoA production and NF κ B. Does inhibition of NF κ B or ACLY impact phenotypes such as phagocytosis or neurosphere formation?

Response: Thanks for the suggestion. We have conducted the phagocytosis tests in vitro to explore whether downstream phenotypes are regulated by ACLY-dependent acetyl-CoA production and NF- κ B. The result showed that the phagocytosis mediated by macrophages was induced by the ACLY inhibitor SB204990 (25 μ M, 24 hrs) or the RelA acetylation inhibitor A-485 (20 μ M, 24 hrs) (**Response Figure 8**).

Minor:

1. Positive and negative controls should be included in the NF- κ B luciferase assay.

Response: The pGL2-NF κ B-luc plasmid is originally constructed by Dr. Li's group (as the sequence shown below) [27-29] with an IL-6 promoter fragment incorporated into pGL2- luciferase reporter plasmid from Promega. IL-6 promoter contains 3 NF- κ B binding sites with sequencing confirmation. This NF- κ B reporter is used for the detection of NF- κ B activity and serves as a positive control.

Red: NF- κ B binding sites

CGGATGCCAGCTTTATTGAGGCTTAGCAGTGGGTTCCCTAGTTAGCCAGAGAGCTCCCA
GGCTCAGATCTGGTCTAACCAGAGAGACCCAGTACAGGCAAAAAGCAGCTGCTTATATG
CAGGATCTGAGGGCTCGCCACTCCCCAGTCCCGCCAGGCCACGCCTCCCTGGAAAGT
CCCAGCGGAAAGTCCCTTGTAGCAAGCTCGGGTTATGTTAGCTCAGTTACAGTACCA
TAAGATACATTGATGAGTTTGGACAAACCACAACCTAGAATGCAGTGAAAAAATGCTTT
A.

2. Fig. 4j: when reporting an increase in RelA and H3 acetylation, total RelA and histone H3 should be shown as controls.

Response: We have added the levels of total RelA and histone H3 in the **Response Figure 9**

3. Manuscript should be edited for grammar/ English language.

Response: Thanks for the suggestion. The final version has been reviewed by senior authors and other qualified esteemed scientists.

Response Figure 8.

Macrophage(THP1) phagocytosis on RR U251 cells treated with ACLY inhibitor SB204990 (25 μ M, 24 hrs) and A-485 (20 μ M, 24 hrs) detected by flow cytometry. *** $P < 0.001$.

Response Figure 9. Protein level of RelA-K310 acetylation, Histone 3 acetylation in RR U251 and RR U87 cells treated with citrate (1 mM, 6 hrs) with or without SB204990 (25 μ M, 24 hrs) detected by Western blot .

Reviewer #3 (Remarks to the Author); expert on glioblastoma:

In this manuscript, Jiang et al present an interesting story linking the re-wired metabolism of radiation resistant GBM cells to decreased immunogenicity. Here, they show that RR cells have an increase of FAO enzymes which allow for the acetylation of NF- κ B subunits, increasing CD47 expression – and that increases of these proteins give rise to a worse OS in GBM patients. Inhibition or genetic manipulation of the FAO enzymes reduced cell growth, and increased phagocytosis – where dual targeting of FAO and CD47 greatly increased mouse OS and phagocytosis. Overall, this is an intriguing story that may assist in developing better immunotherapy combinations in GBM.

Response: We appreciate the insightful comments. We agree ng that linking tumor adaptive metabolic rewiring to an immune checkpoint regulation is novel and has immediate potential for clinical GBM control by radiation or radiation combined with checkpoint immunotherapy. We believe that these findings are highly informative for deciphering a potential dynamical metabolism-associated immunoreaction in a tumor microenvironment. Our results should shed light on the therapy-associated alternation in metabolism and immune checkpoint gene regulation, which may help to define new targets to enhance GBM control by radiation or in combinational therapy of radiation and immunotherapy.

Other comments:

1. Figure 1f should be read “RR” and “RR ET” instead of FIR, and Figure 1d tumor volume is graphed as if they are matched tumors. However, it wasn’t stated in the text that these were primary/recurrent tumors and may be better graphed more clearly. Sup 2h is missing the “RR+ET label”.

Response: We have corrected FIR to RR, and errors in the legend and text.

2. In Sup Figure 5d, the correlation between CD47 and the NF- κ B targets is greater in primary, as compared to recurrent tumors. However, in the primary group there seems to be two sub-groups with lo/lo and hi/hi expression, it would be interesting to see if there was a survival difference between these two subgroups, strengthening the idea that increased CD47/NF- κ B targets leads to worse OS (will recheck the database).

Response: Thank for the suggestion. We have repeated the database analysis using CGGA database and divided 405 primary gliomas in groups with lo/lo and hi/hi expression of NF- κ B subunits/CD47. We found that there was an apparent survival

Response Figure 10. Kaplan-Meier survival of 405 primary glioma categorized by high (red, n = 202) or low (green, n = 203) co-high expression or low expression of NF- κ B subunits, including NF- κ B1 (a), NF- κ B2 (b), RelA (c), Relb (d), RelC (e) and CD47.

benefit for NF- κ B /CD47 lo/lo expression patient group compared with hi/hi expression group patients (**Response Figure 10a-e**). Therefore, NF- κ B and CD47 is connected and NF- κ B /CD47 axis may affect the survival of glioma patients. The new data have been added into **SI Fig. 6b**.

3. ET treatments over 200 μ M have been shown to also have an inhibitory effect on Complex I of the ETC. Do you think this may be an explanation to ET having a greater effect than a CPT1A^{-/-} model (ex. Fig 4f).

Response: We very appreciate your insightful comment. ET has been shown to possess different functions in regulating metabolism such as shown in the treatment of heart disease and diabetes. It may explain the reason for ET to show a greater effect than CPT1A^{-/-}, but there is also concern that ET may inhibit other cell functions including cardiomyocytes by inducing cardiotoxicity. Our current study with relatively a high increased ET concentration may involve FAO unrelated effects but mouse in vivo systematically administration of ET showed an increased macrophage infiltration and phagocytosis, indicating that ET-mediated FAO inhibition suppresses CD47 expression and enhances immune cell attacks.

4. Figure 4g and 4i should also be done in the WT models to show that this pathway is not being affected in the WT endogenously, and further show that the RR cells have been remodeled and this is why they are able to respond to excess citrate.

Response: Thanks for the suggestion. We have repeated experiments presented in Figure 4g and 4i with WT U251 and U87 cells. As shown in Fig 4g and 4i (**Response Figure 11a and 11b**), citrate is a metabolic regulator of CD47 activity and mRNA and the effect is completely reversed by SB204990, a specific inhibitor of ACLY, demonstrating that citrate mediated RelA acetylation is actively involved in CD47 transcription. In WT U251 and U87 cells, citrate may also activate CD47 but with no statistical significance (**Response Figure 11c; SI Figure 7b**). This is consistent with CD47 mRNA levels when the cells were treated by citrate which was also blocked by SB204990 (**Response Figure 11d; SI Figure 7c**).

Response Figure 11. (a) CD47 promoter-controlled luciferase activity with or without NF- κ B motif deletion measured in RR U251 and RR U87 cells treated with citrate (1 mM, 6 hrs). (b) CD47 mRNA level in RR U251 and U87 cells treated with citrate (1 mM, 6 hrs) with or without ACLY inhibitor SB204990 (25 μ M, 24 hrs). (c) CD47 promoter-controlled luciferase activity with or without NF- κ B motif deletion measured in WT U251 and WT U87 cells treated with citrate (1 mM, 6 h). (d) CD47 mRNA level in WT U251 and WT U87 cells treated with citrate (1 mM, 6 h) with or without ACLY inhibitor SB204990 (25 μ M, 24 hrs). * P < 0.05, ** P < 0.01, *** P < 0.001, **** P < 0.0001.

5. Line 241 has a fragmented sentence.

Response: The sentence has been corrected.

6. I think line 259 is supposed to reference Fig. 8a, b – instead of 7.

Response: This has been corrected.

7. Since mitochondrial function is being studied in this manuscript, it would be advantageous to determine if there is a difference between the overall numbers of mitochondria between the WT and RR cells.

Response: We have added new data generated using mitochondria-targeted GFP vector (**Response Figure 12.;** **Revised Figure 2g**).

Response Figure 12. Mitochondria number counts in parental WT and RR GBM cells (U251, GL261; n = 6 per cell line). **** $P < 0.0001$.

8. CPT1A has also been shown to complex with VDAC1, which may also be playing a role in the CPT1A $-/-$ lines, as well as ET treated cells.

Response: Yes, CPT1A/VDAC1 form complex which also affects the mitochondrial FAO metabolism with CPT1A a major rate-limiting factor.

9. Backgrounds on IB blotting images need to be adjusted. Many of them are too dark and some of these blots were adjusted too much, in particular contrast and whiteness. Thickness of the frame of some of these blots are inconsistent. Some of these frames thickness need to be reduce to 0.75 or 0.5 mm.

Response: We have adjusted the WBs in the revised set of Data.

References cited in Response Letter:

- Ferraro GB, Ali A, Luengo A, Kodack DP, Deik A, Abbott KL, Bezwada D, Blanc L, Prideaux B, Jin X, Possada JM, Chen J, Chin CR, Amoozgar Z, Ferreira R, Chen IX, Naxerova K, Ng C, Westermarck AM, Duquette M, Roberge S, Lindeman NI, Lyssiotis CA, Nielsen J, Housman DE, Duda DG, Brachtel E, Golub TR, Cantley LC, Asara JM *et al*: **Fatty acid synthesis is required for breast cancer brain metastasis.** *Nature Cancer* 2021,
- Reinfeld BI, Madden MZ, Wolf MM, Chytil A, Bader JE, Patterson AR, Sugiura A, Cohen AS, Ali A, Do BT, Muir A, Lewis CA, Hongo RA, Young KL, Brown RE, Todd VM, Huffstater T, Abraham A, O'Neil RT, Wilson MH, Xin F, Tantawy MN, Merryman WD, Johnson RW, Williams CS, Mason EF, Mason FM, Beckermann KE, Vander Heiden MG, Manning HC *et al*: **Cell-programmed nutrient partitioning in the tumour microenvironment.** *Nature* 2021,
- Candas-Green D, Xie B, Huang J, Fan M, Wang A, Menaa C, Zhang Y, Zhang L, Jing D, Azghadi S, Zhou W, Liu L, Jiang N, Li T, Gao T, Sweeney C, Shen R, Lin TY, Pan CX,

- Ozpiskin OM, Woloschak G, Grdina DJ, Vaughan AT, Wang JM, Xia S, Monjazez AM, Murphy WJ, Sun LQ, Chen HW, Lam KS *et al*: **Dual blockade of CD47 and HER2 eliminates radioresistant breast cancer cells.** *Nat Commun* 2020, **11**:4591.
4. Wang Z, Fan M, Candas D, Zhang TQ, Qin L, Eldridge A, Wachsmann-Hogiu S, Ahmed KM, Chromy BA, Nantajit D, Duru N, He F, Chen M, Finkel T, Weinstein LS, Li JJ: **Cyclin B1/Cdk1 coordinates mitochondrial respiration for cell-cycle G2/M progression.** *Dev Cell* 2014, **29**:217-232.
 5. Qin L, Fan M, Candas D, Jiang G, Papadopoulos S, Tian L, Woloschak G, Grdina DJ, Li JJ: **CDK1 Enhances Mitochondrial Bioenergetics for Radiation-Induced DNA Repair.** *Cell Rep* 2015, **13**:2056-2063.
 6. Liu L, Xie B, Fan M, Candas-Green D, Jiang JX, Wei R, Wang Y, Chen HW, Hu Y, Li JJ: **Low-Level Saturated Fatty Acid Palmitate Benefits Liver Cells by Boosting Mitochondrial Metabolism via CDK1-SIRT3-CPT2 Cascade.** *Dev Cell* 2020, **52**:196-209 e199.
 7. Duman C, Yaqubi K, Hoffmann A, Acikgoz AA, Korshunov A, Bendszus M, Herold-Mende C, Liu HK, Alfonso J: **Acyl-CoA-Binding Protein Drives Glioblastoma Tumorigenesis by Sustaining Fatty Acid Oxidation.** *Cell Metab* 2019, **30**:274-289 e275.
 8. Kant S, Kesarwani P, Prabhu A, Graham SF, Buelow KL, Nakano I, Chinnaiyan P: **Enhanced fatty acid oxidation provides glioblastoma cells metabolic plasticity to accommodate to its dynamic nutrient microenvironment.** *Cell Death Dis* 2020, **11**:253.
 9. Sperry J, Condro MC, Guo L, Braas D, Vanderveer-Harris N, Kim KKO, Pope WB, Divakaruni AS, Lai A, Christofk H, Castro MG, Lowenstein PR, Le Belle JE, Kornblum HI: **Glioblastoma Utilizes Fatty Acids and Ketone Bodies for Growth Allowing Progression during Ketogenic Diet Therapy.** *iScience* 2020, **23**:101453.
 10. Han S, Wei R, Zhang X, Jiang N, Fan M, Huang JH, Xie B, Zhang L, Miao W, Butler AC, Coleman MA, Vaughan AT, Wang Y, Chen HW, Liu J, Li JJ: **CPT1A/2-Mediated FAO Enhancement-A Metabolic Target in Radioresistant Breast Cancer.** *Front Oncol* 2019, **9**:1201.
 11. Candas D, Lu CL, Fan M, Chuang FY, Sweeney C, Borowsky AD, Li JJ: **Mitochondrial MKP1 is a target for therapy-resistant HER2-positive breast cancer cells.** *Cancer Res* 2014, **74**:7498-7509.
 12. Ren YB, Luo T, Li J, Fu J, Wang Q, Cao GW, Chen Y, Wang HY: **p28(GANK) associates with p300 to attenuate the acetylation of RelA.** *Mol Carcinog* 2015, **54**:1626-1635.
 13. Tsai YJ, Tsai T, Peng PC, Li PT, Chen CT: **Histone acetyltransferase p300 is induced by p38MAPK after photodynamic therapy: the therapeutic response is increased by the p300HAT inhibitor anacardic acid.** *Free Radic Biol Med* 2015, **86**:118-132.
 14. Divakaruni AS, Hsieh WY, Minarrieta L, Duong TN, Kim KKO, Desousa BR, Andreyev AY, Bowman CE, Caradonna K, Dranka BP, Ferrick DA, Liesa M, Stiles L, Rogers GW, Braas D, Ciaraldi TP, Wolfgang MJ, Sparwasser T, Berod L, Bensinger SJ, Murphy AN: **Etomoxir Inhibits Macrophage Polarization by Disrupting CoA Homeostasis.** *Cell Metab* 2018, **28**:490-503 e497.
 15. Nomura M, Liu J, Rovira, II, Gonzalez-Hurtado E, Lee J, Wolfgang MJ, Finkel T: **Fatty acid oxidation in macrophage polarization.** *Nat Immunol* 2016, **17**:216-217.
 16. Weller M, Le Rhun E: **How did lomustine become standard of care in recurrent glioblastoma?** *Cancer Treat Rev* 2020, **87**:102029.
 17. Audureau E, Chivet A, Ursu R, Corns R, Metellus P, Noel G, Zouaoui S, Guyotat J, Le Reste PJ, Faillot T, Litre F, Desse N, Petit A, Emery E, Lechapt-Zalcman E, Peltier J, Duntze J,

- Dezamis E, Voirin J, Menei P, Caire F, Dam Hieu P, Barat JL, Langlois O, Vignes JR, Fabbro-Peray P, Riondel A, Sorbets E, Zanello M, Roux A *et al*: **Prognostic factors for survival in adult patients with recurrent glioblastoma: a decision-tree-based model.** *J Neurooncol* 2018, **136**:565-576.
18. Weller M, Cloughesy T, Perry JR, Wick W: **Standards of care for treatment of recurrent glioblastoma--are we there yet?** *Neuro Oncol* 2013, **15**:4-27.
 19. Gutmann DH, Kettenmann H: **Microglia/Brain Macrophages as Central Drivers of Brain Tumor Pathobiology.** *Neuron* 2019, **104**:442-449.
 20. Lin H, Patel S, Affleck VS, Wilson I, Turnbull DM, Joshi AR, Maxwell R, Stoll EA: **Fatty acid oxidation is required for the respiration and proliferation of malignant glioma cells.** *Neuro Oncol* 2017, **19**:43-54.
 21. von Roemeling CA, Wang Y, Qie Y, Yuan H, Zhao H, Liu X, Yang Z, Yang M, Deng W, Bruno KA, Chan CK, Lee AS, Rosenfeld SS, Yun K, Johnson AJ, Mitchell DA, Jiang W, Kim BYS: **Therapeutic modulation of phagocytosis in glioblastoma can activate both innate and adaptive antitumour immunity.** *Nat Commun* 2020, **11**:1508.
 22. Tan Z, Xiao L, Tang M, Bai F, Li J, Li L, Shi F, Li N, Li Y, Du Q, Lu J, Weng X, Yi W, Zhang H, Fan J, Zhou J, Gao Q, Onuchic JN, Bode AM, Luo X, Cao Y: **Targeting CPT1A-mediated fatty acid oxidation sensitizes nasopharyngeal carcinoma to radiation therapy.** *Theranostics* 2018, **8**:2329-2347.
 23. Wagner IV, Perwitz N, Drenckhan M, Lehnert H, Klein J: **Cannabinoid type 1 receptor mediates depot-specific effects on differentiation, inflammation and oxidative metabolism in inguinal and epididymal white adipocytes.** *Nutr Diabetes* 2011, **1**:e16.
 24. Hsu E, Zemke NR, Berk AJ: **Promoter-specific changes in initiation, elongation, and homeostasis of histone H3 acetylation during CBP/p300 inhibition.** *Elife* 2021, **10**:
 25. Weinert BT, Narita T, Satpathy S, Srinivasan B, Hansen BK, Scholz C, Hamilton WB, Zucconi BE, Wang WW, Liu WR, Brickman JM, Kesicki EA, Lai A, Bromberg KD, Cole PA, Choudhary C: **Time-Resolved Analysis Reveals Rapid Dynamics and Broad Scope of the CBP/p300 Acetylome.** *Cell* 2018, **174**:231-244 e212.
 26. Zhang L, Sheng C, Zhou F, Zhu K, Wang S, Liu Q, Yuan M, Xu Z, Liu Y, Lu J, Liu J, Zhou L, Wang X: **CBP/p300 HAT maintains the gene network critical for beta cell identity and functional maturity.** *Cell Death Dis* 2021, **12**:476.
 27. Ahmed KM, Nantajit D, Fan M, Murley JS, Grdina DJ, Li JJ: **Coactivation of ATM/ERK/NF-kappaB in the low-dose radiation-induced radioadaptive response in human skin keratinocytes.** *Free Radic Biol Med* 2009, **46**:1543-1550.
 28. Wang Z, Cao N, Nantajit D, Fan M, Liu Y, Li JJ: **Mitogen-activated protein kinase phosphatase-1 represses c-Jun NH2-terminal kinase-mediated apoptosis via NF-kappaB regulation.** *J Biol Chem* 2008, **283**:21011-21023.
 29. Li JJ, Westergaard C, Ghosh P, Colburn NH: **Inhibitors of both nuclear factor-kappaB and activator protein-1 activation block the neoplastic transformation response.** *Cancer Res* 1997, **57**:3569-3576.

Data Attached in Response Letter:

1. Response Figures (1-12)
2. Response Table (Table S2)

a (Fig. 4k)

b (Fig.4l)

c (SI Fig 8a)

Human CD47 potential Acetylation sets			
KAT	Site	Sequence	P-value
CBP/p300	166	GQFGIKTLK ^R YRSGGMDE	0.0395
CBP/p300	163	LFWGQFGIK ^R TLKYRSGG	0.5121
CBP/p300	175	YRSGGMDEK ^R TIALLVAG	0.884
p300	290	QLLGLVYMK ^R FVE-----	1

d (SI Fig 8b)

e (SI Fig 8c)

Response Figure 1. (a) Radioresistant GBM cells transfected with CD47 promoter-controlled luciferase activity were either treated with A-485 (20 μ M, 24 hrs), CBP/p300 inhibitor, or co-transfected with KAT-specific acetylation WT or Mut RelA. (b) CD47 mRNA level in RR U251 and RR U87 cells treated with A-485 (20 μ M, 24 hrs) or transfected with KAT-specific acetylation WT or mutant RelA. (c) Predicted KAT-specific acetylation sites in human CD47 protein. (d) Co-immunoprecipitation assays with WT U251 and RR U251 cells transfected with P300 with or without A-485 (20 μ M, 24 hrs). (e) Degradation of CD47 protein in RR U251 cells with or without A-485 treatment on indicated time point. n = 3; ** P < 0.01, **** P < 0.0001.

a (SI Fig.10d)**b** (Fig. 3d)**c** (Fig. 5c)**d** (Fig 5b)**e** (Fig 5e)**f** (Fig 5a)
Response Figure 2. Phagocytosis of mouse macrophage on mouse GBM GL261 cells. **(a)** Macrophage phagocytosis on CPT2^{-/-} RR GL261 cells compared with RR GL261 cells. **(b)** Macrophage phagocytosis on WT and RR GBM cells (U251, U87, GL261). **(c)** Macrophage phagocytosis on WT and RR GL261 cells treated with ET (80uM,24 h). **(d)** Enhanced macrophage phagocytosis on RR U251 cells treated with indicated ET concentrations (24 h). **(e)** Macrophage(THP1) phagocytosis on vector control and CRISPR-KO CPT1A^{-/-}, CPT2^{-/-}, ACAD9^{-/-} RR U251 cells detected by flow cytometry. **(f)** WT U251 and GL261 cells treated with L-carnitine (L-car, 10 mM, 48 h), palmitate 974 (PA, 25 uM, 48 h) or IR (5 Gy, 16 h). n = 3; **P* < 0.05, ***P* < 0.01, *** *P* < 0.001, *****P* < 0.0001

Response Figure 3. Phagocytosis on WT GL261 cells mediated by mice macrophage in presence of indicated ET concentrations for 24 hrs. n=3; * $P < 0.05$, **** $P < 0.0001$; ns= not significant

a (Fig 5d)**b (Fig 5c)**
Response Figure 4. (a) Upper panel: Western blot of CD47 expression in control and ET treated (200 μ M, 48 h) WT U251 cells compared to CRISPR-mediated CPT1A^{-/-}, CPT2^{-/-}, ACAD9^{-/-} of RR U251 cells. Lower panel: Percentage of CD47 expressing cells detected by flow cytometry in CPT1A^{-/-}, CPT2^{-/-}, ACAD9^{-/-} RR U251 populations or ET treated RR U251 cells. (b) Macrophage phagocytosis on GL261 cells with and without ET treatment (80 μ M, 24 h). n=3; **** $P < 0.0001$; ns= not significant

a (Fig 2e)**b (Fig 2d)****c (SI Fig 7a)**
Response Figure 5. (a) Glucose uptake measured in culture medium of WT and RR U251, U87, A172, GL261 cells (n=3 per cell line). **(b)** The concentration of Acetyl CoA which is the final production of FAO presented in WT and RR GBM cells (U251, U87, A172, GL261). **(c)** Acetyl CoA level presented in RR U251 cells compared with ET treated (200µM, 48hrs) and CPT1A^{-/-}, CPT2^{-/-}, ACAD9^{-/-} RR U251 cells. n = 3; *P < 0.05, **P < 0.01, *** P < 0.001, ****P < 0.0001.

a (SI Fig 3a)**b** (Fig 2c)
Response Figure 6. (a) CTP2 enzymatic activity in WT and RR U251 cells measured by monitoring ¹⁴C- palmitoyl-carnitine (n = 4). (b) FAO activity quantified by the conversion of ³H palmitic acid to ³H₂O over 6 hours using radioactive isotope tracers ([9,10-³H(N)]-palmitic acid [0.5 μCi (~9.3 pmol)]) (n=3 per cell line) in parental WT and RR GBM cells (U251, U87, A172, GL261). **P* < 0.05, ***P* < 0.01, *** *P* < 0.001).

a(SI Fig 7b)

b(SI Fig 7c)

c (SI Fig 7d)

d (SI Fig 7e)

e (Fig 4k)

f (Fig 4l)

Response Figure 7. (a) CD47 promoter-controlled luciferase activity with or without NF- κ B motif deletion measured in WT U251 and WT U87 cells treated with citrate (1 mM, 6 hrs). (b) CD47 mRNA level in WT U251 and WT U87 cells treated with citrate (1 mM, 6 hrs) with or without ACLY inhibitor SB204990 (25 μ M, 24 hrs). (c) Luciferase reporter activity driven by NF- κ B in RR U251 and RR U87 cells treated with with citrate (1 mM, 6 hrs) with or without A-485 (20 μ M, 24 hrs). (d) Western blot of RelA-K310 acetylation, CD47 in RR U251 and RR U87 cells treated with citrate (1 mM, 6 h) with or without A-485 (20 μ M, 24 hrs). (e) CD47 promoter-controlled luciferase activity in RR U251 and RR U87 cells treated with A-485 (20 μ M, 24 hrs), CBP/p300 inhibitor, or co-transfected with KAT-specific acetylation WT or Mut RelA. (f) CD47 mRNA level in RR U251 cells treated with A485 (20 μ M, 24 hrs) transfected with WT or mutant RelA mut. (f) Western blot of RelA-K310 acetylation, CD47 in RR U251 and RR U87 cells treated with citrate (1 mM, 6 h) with or without A-485 (20 μ M, 24 h). * P <0.05, ** P < 0.01, *** P < 0.001, **** P <0.0001.

Response Figure 8.

Macrophage(THP1)

phagocytosis on RR U251 cells treated with ACLY inhibitor SB 204990 (25 μ M, 24 hrs) and A-485 (20 μ M, 24 hrs) detected by flow cytometry. *** $P < 0.001$.

Response Figure 9. Protein level of RelA-K310 acetylation, Histone 3 acetylation in RR U251 and RR U87 cells treated with citrate (1 mM, 6 hrs) with or without SB204990 (25 μ M, 24 hrs) detected by Western blot .

(SI Fig 6b)

Response Figure 10. Kaplan-Meier survival of 405 primary glioma categorized by high (red, n = 202) or low (green, n = 203) co-high expression or low expression of NF-κB subunits, including NF-κB1 (**a**), NF-κB2 (**b**), RelA (**c**), Relb (**d**), RelC (**e**) and CD47.

a (Fig 4g)**b (Fig 4i)****c (SI Fig 7b)****d (SI Fig 7c)**
Response Figure 11. (a) CD47 promoter-controlled luciferase activity with or without NF- κ B motif deletion measured in RR U251 and RR U87 cells treated with citrate (1 mM, 6 hrs). (b) CD47 mRNA level in RR U251 and U87 cells treated with citrate (1 mM, 6 hrs) with or without ACLY inhibitor SB204990 (25 μ M, 24 hrs). (c) CD47 promoter-controlled luciferase activity with or without NF- κ B motif deletion measured in WT U251 and WT U87 cells treated with citrate (1 mM, 6 h). (d) CD47 mRNA level in WT U251 and WT U87 cells treated with citrate (1 mM, 6 h) with or without ACLY inhibitor SB204990 (25 μ M, 24 hrs). * P < 0.05, ** P < 0.01, *** P < 0.001, **** P < 0.0001.

Response Figure 12. Mitochondria number counts in parental WT and RR GBM cells (U251, GL261; $n = 6$ per cell line). **** $P < 0.0001$.

Table 1. Palmitate-induced cellular effects on multiple cell types. Related to Results

Concentration (μ M), ≥ 100	Time (h)	Cells	Cellular effects	Associated proteins	Affected pathways	Reference
800	2, 8, 24	Huh-7 cells, human hepatocytes prepared from clinical liver resection specimens, Mouse hepatocytes isolated from C57BL/6 mice	Apoptosis	JNK, c-Jun, PUMA, NOXA, caspase-3, caspase-7	JNK1-dependent PUMA expression; hepatocytes lipoapoptosis	Cazanave et al., 2009
200, 400, 800	8, 18	Human hepatocellular carcinoma Huh-7 and Hep3B cell lines, primary hepatocytes	Lipoapoptosis	DEVDase, CHOP, Phospho-eIF2- α , XBP-1, JNK1/2, PUMA, Bax	Palmitate-induced ER stress; PUMA and Bim expression attenuated by palmitoleate	Akazawa et al., 2010
100, 500, 750	24	HepG2 cells, Primary mouse hepatocytes derived from C57/B6 mice	Increased ROS and cell death	Sirt3	SIRT3 regulated by nutrient excess; hepatic susceptibility to lipotoxicity	Bao et al., 2010
500	12, 24, 48	CHO cells	Oxidative stress and apoptosis	gadd7, CHOP, GRP78, JNK1, JNK2	The Non-coding RNA gadd7 is a regulator of Lipid-induced oxidative and endoplasmic reticulum stress	Brookheart et al., 2009
500	24, 48	Pancreatic ductal adenocarcinoma Capan-2, and epithelioid carcinoma Panc-1 cells, human beta cells	Increased chemokines and cytokines expression	IL-6, IL-8, IL1- β , CXCL1, CCL2, TNF- α and I κ B α	Palmitate induces chemokines, cytokines expression and cell death	Igoillo-Esteve et al., 2010
500	24	CHO, LY-B cells	Apoptosis	Ceramide, caspase-3	de novo ceramide synthesis; oxidative stress	Listenberger et al., 2001
500	1, 5	CHO cells, H9c2 rat cardiomyoblasts	Cell death	Caspases-3, caspases-7, cytochrome c, GRP78, PDI	Palmitate increases ROS, induce ER stress; release of ER calcium stores; mitochondria-mediated cell death	Borradaile et al., 2006b
200, 300, 400, 500	4, 8, 16, 24	H4IIE liver cells, primary hepatocytes isolated from C57BL/6J mice	ER stress and cell death	CHOP, JNK, caspase-3	Palmitate linked ER stress to cell death via JNK activation	Pfaffenbach et al., 2010
100, 200, 300, 400, 500	5, 12, 24	CHO cells, H9c2 rat cardiomyoblasts	ER stress and apoptosis	eEF1A-1	eEF1A-1 required in lipotoxicity and cell death	Borradaile et al., 2006a
125, 250, 500	12, 24	THP-1 cells,	IL-6 and TNF- α production	ERK1/2, Akt, IL-6, TNF- α	Palmitate regulates IL-6 in proinflammatory monocytes	Bunn et al., 2010
100, 150, 250, 375, 500	24	Human neuroblastoma SH-SY5Y cells and N2a cells	Increased ER stress and attenuation in leptin and IGF-1 expression	CHOP, PERK, IRE1, ATF3, ATF4, ATF6, XBP1, Leptin, IGF1	Palmitate increases the risk of degenerative diseases via reducing IGF1 and leptin expression	(Marwarha et al., 2016

Continued **Table 1**. Palmitate-induced cellular effects on multiple cell types

Concentration (μ M), ≥ 100	Time (h)	Cells	Cellular effects	Associated proteins	Affected pathways	Reference
100, 200, 400, 500	24	Bone marrow-derived macrophages and bone marrow-derived dendritic cells	Activation of NLRP3-ASC inflammasome	Caspase-1, IL-1 β , IL-6, IL-18, TNF, AMPK, Akt, IRS1	SFA in HFD promotes AMPK-autophagy-ROS pathway; NLRP3 inflammasome in macrophages	Wen et al., 2011
100, 200, 300, 400, 500	3, 6, 16	H4IIE liver cells	ER stress and apoptosis	CHOP, GADD34, GRP78, ATF4, XBP1, caspase-3, caspase-9	SFA enhances CHOP, GADD34, GRP78 and ATF4; activates caspase-3 and caspase-9	Wei et al., 2007
400	3, 12, 24	H4IIEC3 rat hepatoma cells	ROS accumulation, caspase activation, metabolic dysregulation and loss of cell viability	Caspase-3, caspase-7	Palmitate induces ROS ; mitochondrial oxidative metabolism	Egnatchik et al., 2014
200, 400	24	Pericytes	Oxidative stress and apoptosis	NADPH oxidase, NF- κ B, ceramide, caspase-3, bax, bcl-2,	Palmitate-induced apoptosis; NADPH oxidase-mediated oxidative stress; ceramide accumulation and NF- κ B activation	Cacicedo et al., 2005
200, 300, 400	24	INS 832/13 cells	Apoptosis	Caspase-3	Palmitate synergizes glucose; caspase-3 activation and cell death	El-Assaad et al., 2003
200, 400	6, 16	Huh-7 cells, hepatocytes isolated from C57BL/6 mice	Apoptosis	Bim, FoxO3a, caspase-3, caspase-7, PP2A,	FFA induces lipapoptosis ; PP2A activation; Bim expression; FoxO3a	Barreyro et al., 2007
150, 250, 350	24	HepG2 cells	ROS accumulation in hepatocytes	NOX3, p38, JNK, PI3K, Akt, FOXO1, PEPCK	Palmitate promotes NOX3; ROS; human hepatocytes	Gao et al., 2010
125, 250	20	Primary skeletal muscle cells, HepG2 hepatoma cells	N/A	IL-6 expression	NF- κ B activation; oxidative stress	Weigert et al., 2004
250	2, 4, 6, 16	H4IIE liver cells, rat primary hepatocytes	ER stress and cell death	CHOP, GRP78, GADD34, ATF4, cytochrome c	SFA links calcium disturbance; ER stress; UPR activation; cell death	Wei et al., 2009
200	72	Bovine aortic smooth muscle cells and endothelial cells obtained from calf aorta as described	ROS production	DAG, PKC	ROS generation; PKC-dependent activation of NADPH oxidase	Inoguchi et al., 2000
200	18, 24	HepG2, Huh7 and MRH 7777 cells, hepatocytes isolated from C57/Bl6 mice	Lipoapoptosis	JNK, Bim, Bax	FFA-induced JNK activation; mitochondrial-mediated apoptosis; Bim; Bax activation	Malhi et al., 2006

Continued **Table 1.** Palmitate-induced cellular effects on multiple cell types

Concentration (μM), <100	Time (h)	Cells	Cellular effects	Reference
50	12, 24	THP-1 cells,	In contrasted with a palmitate concentration of 125 μM , 250 μM or 500 μM , 50 μM do not induce IL-6 and TNF- α production.	Bunn et al., 2010
50	24	Bone marrow-derived macrophages and bone marrow-derived dendritic cells	50 μM of palmitate do not activate NLRP3-ASC inflammasome nor cytokines expression compared with concentrations of 100 μM , 200 μM , 400 μM or 500 μM	Wen et al., 2011
25, 50	24	Human neuroblastoma SH-SY5Y cells and N2a cells	Palmitate 25 μM or 50 μM do not affect the expression of leptin and IGF-1; 100 μM , 150 μM , 250 μM , 375 μM and 500 μM attenuate leptin and IGF-1 which associate with ER stress in metabolic disease including obesity, diabetes, atherosclerosis and cardiovascular diseases.	Marwarha et al., 2016

References Cited in Table 1 on PA mediated normal tissue lipotoxicity

- Akazawa, Y., Cazanave, S., Mott, J.L., Elmi, N., Bronk, S.F., Kohno, S., Charlton, M.R., and Gores, G.J. (2010). Palmitoleate attenuates palmitate-induced Bim and PUMA up-regulation and hepatocyte lipoapoptosis. *Journal of hepatology* 52, 586-593.
- Bao, J., Scott, I., Lu, Z., Pang, L., Dimond, C.C., Gius, D., and Sack, M.N. (2010). SIRT3 is regulated by nutrient excess and modulates hepatic susceptibility to lipotoxicity. *Free radical biology & medicine* 49, 1230-1237.
- Barreyro, F.J., Kobayashi, S., Bronk, S.F., Werneburg, N.W., Malhi, H., and Gores, G.J. (2007). Transcriptional regulation of Bim by FoxO3A mediates hepatocyte lipoapoptosis. *The Journal of biological chemistry* 282, 27141-27154.
- Borradaile, N.M., Buhman, K.K., Listenberger, L.L., Magee, C.J., Morimoto, E.T., Ory, D.S., and Schaffer, J.E. (2006a). A critical role for eukaryotic elongation factor 1A-1 in lipotoxic cell death. *Molecular biology of the cell* 17, 770-778.
- Borradaile, N.M., Han, X., Harp, J.D., Gale, S.E., Ory, D.S., and Schaffer, J.E. (2006b). Disruption of endoplasmic reticulum structure and integrity in lipotoxic cell death. *Journal of lipid research* 47, 2726-2737.
- Brookheart, R.T., Michel, C.I., Listenberger, L.L., Ory, D.S., and Schaffer, J.E. (2009). The non-coding RNA gadd7 is a regulator of lipid-induced oxidative and endoplasmic reticulum stress. *The Journal of biological chemistry* 284, 7446-7454.
- Bunn, R.C., Cockrell, G.E., Ou, Y., Thrailkill, K.M., Lumpkin, C.K., Jr., and Fowlkes, J.L. (2010). Palmitate and insulin synergistically induce IL-6 expression in human monocytes. *Cardiovascular diabetology* 9, 73.
- Cacicedo, J.M., Benjachareowong, S., Chou, E., Ruderman, N.B., and Ido, Y. (2005). Palmitate-induced apoptosis in cultured bovine retinal pericytes: roles of NAD(P)H oxidase, oxidant stress, and ceramide. *Diabetes* 54, 1838-1845.
- Cazanave, S.C., Mott, J.L., Elmi, N.A., Bronk, S.F., Werneburg, N.W., Akazawa, Y., Kahraman, A., Garrison, S.P., Zambetti, G.P., Charlton, M.R., et al. (2009). JNK1-dependent PUMA expression contributes to hepatocyte lipoapoptosis. *The Journal of biological chemistry* 284, 26591-26602.
- Egnatchik, R.A., Leamy, A.K., Jacobson, D.A., Shiota, M., and Young, J.D. (2014). ER calcium release promotes mitochondrial dysfunction and hepatic cell lipotoxicity in response to palmitate overload. *Molecular metabolism* 3, 544-553.
- El-Assaad, W., Buteau, J., Peyot, M.L., Nolan, C., Roduit, R., Hardy, S., Joly, E., Dbaibo, G., Rosenberg, L., and Prentki, M. (2003). Saturated fatty acids synergize with elevated glucose to cause pancreatic beta-cell death. *Endocrinology* 144, 4154-4163.
- Gao, D., Nong, S., Huang, X., Lu, Y., Zhao, H., Lin, Y., Man, Y., Wang, S., Yang, J., and Li, J. (2010). The effects of palmitate on hepatic insulin resistance are mediated by NADPH Oxidase 3-derived reactive oxygen species through JNK and p38MAPK pathways. *The Journal of biological chemistry* 285, 29965-29973.
- Igoillo-Esteve, M., Marselli, L., Cunha, D.A., Ladriere, L., Ortis, F., Grieco, F.A., Dotta, F., Weir, G.C., Marchetti, P., Eizirik, D.L., et al. (2010). Palmitate induces a pro-inflammatory response in human pancreatic islets that mimics CCL2 expression by beta cells in type 2 diabetes. *Diabetologia* 53, 1395-1405.
- Inoguchi, T., Li, P., Umeda, F., Yu, H.Y., Kakimoto, M., Imamura, M., Aoki, T., Etoh, T., Hashimoto, T., Naruse, M., et al. (2000). High glucose level and free fatty acid stimulate reactive oxygen species production through protein kinase C--dependent activation of NAD(P)H oxidase in cultured vascular cells. *Diabetes* 49, 1939-1945.

- Listenberger, L.L., Ory, D.S., and Schaffer, J.E. (2001). Palmitate-induced apoptosis can occur through a ceramide-independent pathway. *The Journal of biological chemistry* 276, 14890-14895.
- Malhi, H., Bronk, S.F., Werneburg, N.W., and Gores, G.J. (2006). Free fatty acids induce JNK-dependent hepatocyte lipoapoptosis. *The Journal of biological chemistry* 281, 12093-12101.
- Marwarha, G., Claycombe, K., Schommer, J., Collins, D., and Ghribi, O. (2016). Palmitate-induced Endoplasmic Reticulum stress and subsequent C/EBPalpha Homologous Protein activation attenuates leptin and Insulin-like growth factor 1 expression in the brain. *Cellular signalling* 28, 1789-1805.
- Pfaffenbach, K.T., Gentile, C.L., Nivala, A.M., Wang, D., Wei, Y., and Pagliassotti, M.J. (2010). Linking endoplasmic reticulum stress to cell death in hepatocytes: roles of C/EBP homologous protein and chemical chaperones in palmitate-mediated cell death. *American journal of physiology Endocrinology and metabolism* 298, E1027-1035.
- steatohepatitis in rats is associated with c-Jun NH2-terminal kinase activation and elevated proapoptotic Bax. *The Journal of nutrition* 138, 1866-1871.
- Wei, Y., Wang, D., Gentile, C.L., and Pagliassotti, M.J. (2009). Reduced endoplasmic reticulum luminal calcium links saturated fatty acid-mediated endoplasmic reticulum stress and cell death in liver cells. *Molecular and cellular biochemistry* 331, 31-40.
- Wei, Y., Wang, D., and Pagliassotti, M.J. (2007). Saturated fatty acid-mediated endoplasmic reticulum stress and apoptosis are augmented by trans-10, cis-12-conjugated linoleic acid in liver cells. *Molecular and cellular biochemistry* 303, 105-113.
- Weigert, C., Brodbeck, K., Staiger, H., Kausch, C., Machicao, F., Haring, H.U., and Schleicher, E.D. (2004). Palmitate, but not unsaturated fatty acids, induces the expression of interleukin-6 in human myotubes through proteasome-dependent activation of nuclear factor-kappaB. *The Journal of biological chemistry* 279, 23942-23952.
- Wen, H., Gris, D., Lei, Y., Jha, S., Zhang, L., Huang, M.T., Brickey, W.J., and Ting, J.P. (2011). Fatty acid-induced NLRP3-ASC inflammasome activation interferes with insulin signaling. *Nature immunology* 12, 408-415.

REVIEWERS' COMMENTS

Reviewer #2 (Remarks to the Author):

Thank you to the authors for their comprehensive efforts to address the reviewer concerns. My concerns have been addressed and I recommend publication of this excellent study.

Reviewer #3 (Remarks to the Author):

In the revised manuscript, the authors has addressed all of my comments with new data and necessary revisions. This is an improved study with high significance. The revised manuscript is sufficient for its consideration of its publication in NCOMMS.